# Swarm Learning for decentralized and confidential clinical machine learning

Stefanie Warnat-Herresthal[1,2,127], Hartmut Schultze[3,127], Krishnaprasad Lingadahalli Shastry[3,127], Sathyanarayanan Manamohan[3,127], Saikat Mukherjee[3,127], Vishesh Garg[3,4,127], Ravi Sarveswara[3,127], Kristian Händler[1,5,127], Peter Pickkers[6,127], N. Ahmad Aziz[7,8,127], Sofia Ktena[9,127], Florian Tran[10,11], Michael Bitzer[12], Stephan Ossowski[13,14], Nicolas Casadei[13,14], Christian Herr[15], Daniel Petersheim[16], Uta Behrends[17], Fabian Kern[18], Tobias Fehlmann[18], Philipp Schommers[19], Clara Lehmann[19,20,21], Max Augustin[19,20,21], Jan Rybniker[19,20,21], Janine Altmüller[22], Neha Mishra[11], Joana P. Bernardes[11], Benjamin Krämer[23], Lorenzo Bonaguro[1,2], Jonas Schulte-Schrepping[1,2], Elena De Domenico[1,5], Christian Siever[3], Michael Kraut[1,5], Milind Desai[3], Bruno Monnet[3], Maria Saridaki[9], Charles Martin Siegel[3], Anna Drews[1,5], Melanie Nuesch-Germano[1,2], Heidi Theis[1,5], Jan Heyckendorf[23], Stefan Schreiber[10], Sarah Kim-Hellmuth[16], COVID-19 Aachen Study (COVAS)*, Jacob Nattermann[24,25], Dirk Skowasch[26], Ingo Kurth[27], Andreas Keller[18,28], Robert Bals[15], Peter Nürnberg[22], Olaf Rieß[13,14], Philip Rosenstiel[11], Mihai G. Netea[29,30], Fabian Theis[31], Sach Mukherjee[32], Michael Backes[33], Anna C. Aschenbrenner[1,2,5,29], Thomas Ulas[1,2], Deutsche COVID-19 Omics Initiative (DeCOI)*, Monique M. B. Breteler[7,34,128], Evangelos J. Giamarellos-Bourboulis[9,128], Matthijs Kox[6,128], Matthias Becker[1,5,128], Sorin Cheran[3,128], Michael S. Woodacre[3,128], Eng Lim Goh[3,128] & Joachim L. Schultze[1,2,5,128✉]

Fast and reliable detection of patients with severe and heterogeneous illnesses is a major goal of precision medicine[1,2]. Patients with leukaemia can be identified using machine learning on the basis of their blood transcriptomes[3]. However, there is an increasing divide between what is technically possible and what is allowed, because of privacy legislation[4,5]. Here, to facilitate the integration of any medical data from any data owner worldwide without violating privacy laws, we introduce Swarm Learning—a decentralized machine-learning approach that unites edge computing, blockchain-based peer-to-peer networking and coordination while maintaining confidentiality without the need for a central coordinator, thereby going beyond federated learning. To illustrate the feasibility of using Swarm Learning to develop disease classifiers using distributed data, we chose four use cases of heterogeneous diseases (COVID-19, tuberculosis, leukaemia and lung pathologies). With more than 16,400 blood transcriptomes derived from 127 clinical studies with non-uniform distributions of cases and controls and substantial study biases, as well as more than 95,000 chest X-ray images, we show that Swarm Learning classifiers outperform those developed at individual sites. In addition, Swarm Learning completely fulfils local confidentiality regulations by design. We believe that this approach will notably accelerate the introduction of precision medicine.

Identification of patients with life-threatening diseases, such as leukaemias, tuberculosis or COVID-19[6,7], is an important goal of precision medicine[2]. The measurement of molecular phenotypes using 'omics' technologies[1] and the application of artificial intelligence (AI) approaches[4,8] will lead to the use of large-scale data for diagnostic purposes. Yet, there is an increasing divide between what is technically possible and what is allowed because of privacy legislation[5,9,10]. Particularly in a global crisis[6,7], reliable, fast, secure, confidentiality- and privacy-preserving AI solutions can facilitate answering important questions in the fight against such threats[11–13]. AI-based concepts range from drug target prediction[14] to diagnostic software[15,16]. At the same time, we need to consider important standards relating to data privacy and protection, such as Convention 108+ of the Council of Europe[17].

AI-based solutions rely intrinsically on appropriate algorithms[18], but even more so on large training datasets[19]. As medicine is inherently decentral, the volume of local data is often insufficient to train reliable classifiers[20,21]. As a consequence, centralization of data is one model that has been used to address the local limitations[22]. While beneficial from an AI perspective, centralized solutions have inherent disadvantages, including increased data traffic and concerns about data ownership, confidentiality, privacy, security and the creation of data monopolies that favour data aggregators[19]. Consequently, solutions

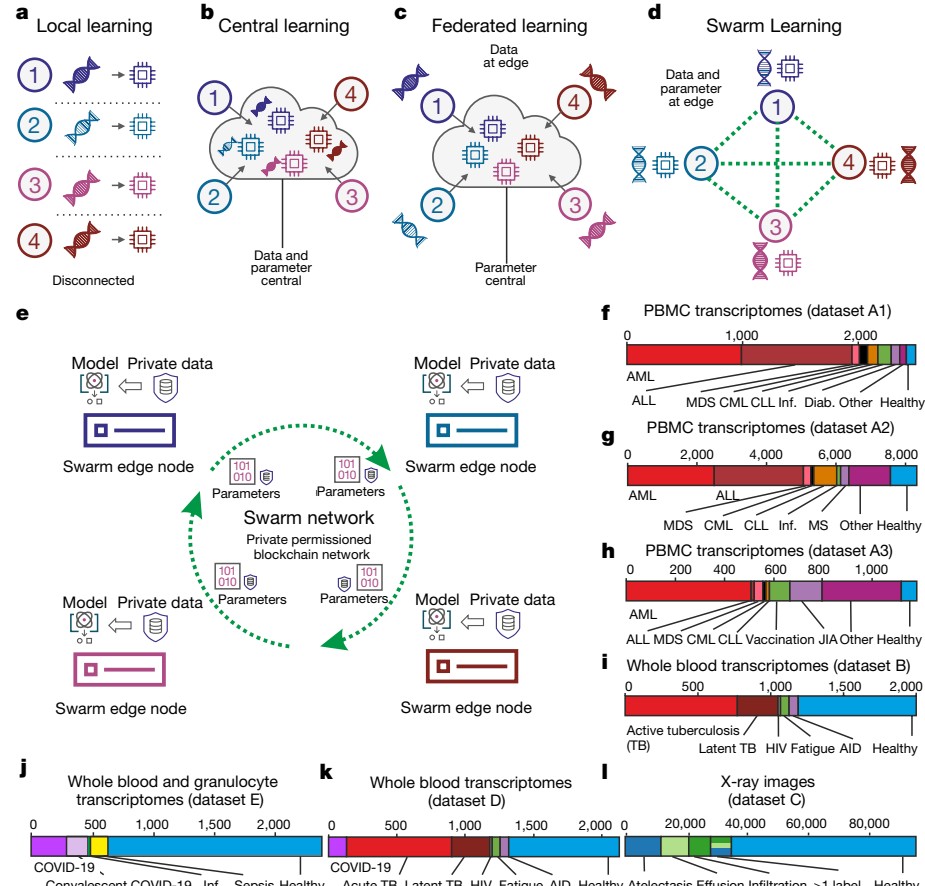

**Fig. 1 | Concept of Swarm Learning. a**, Illustration of the concept of local learning with data and computation at different, disconnected locations. **b**, Principle of cloud-based machine learning. **c**, Federated learning, with data being kept with the data contributor and computing performed at the site of local data storage and availability, but parameter settings orchestrated by a central parameter server. **d**, Principle of SL without the need for a central custodian. **e**, Schematic of the Swarm network, consisting of Swarm edge nodes that exchange parameters for learning, which is implemented using blockchain technology. Private data are used at each node together with the model provided by the Swarm network. **f–l**, Descriptions of the transcriptome datasets used. **f**, **g**, Datasets A1 (**f**; *n* = 2,500) and A2 (**g**; *n* = 8,348): two microarray-based transcriptome datasets of PBMCs. **h**, Dataset A3: 1,181 RNA-seq-based transcriptomes of PBMCs. **i**, Dataset B: 1,999 RNA-seq-based whole blood transcriptomes. **j**, Dataset E: 2,400 RNA-seq-based whole blood and granulocyte transcriptomes. **k**, Dataset D: 2,143 RNA-seq-based whole blood transcriptomes. **l**, Dataset C: 95,831 X-ray images. CML, chronic myeloid leukaemia; CLL, chronic lymphocytic leukaemia; Inf., infections; Diab., type II diabetes; MDS, myelodysplastic syndrome; MS, multiple sclerosis; JIA, juvenile idiopathic arthritis; TB, tuberculosis; HIV, human immunodeficiency virus; AID, autoimmune disease.

to the challenges of central AI models must be effective, accurate and efficient; must preserve confidentiality, privacy and ethics; and must be secure and fault-tolerant by design[23,24]. Federated AI addresses some of these aspects[19,25]. Data are kept locally and local confidentiality issues are addressed[26], but model parameters are still handled by central custodians, which concentrates power. Furthermore, such star-shaped architectures decrease fault tolerance.

We hypothesized that completely decentralized AI solutions would overcome current shortcomings, and accommodate inherently decentral data structures and data privacy and security regulations in medicine. The solution (1) keeps large medical data locally with the data owner; (2) requires no exchange of raw data, thereby also reducing data traffic; (3) provides high-level data security; (4) guarantees secure, transparent and fair onboarding of decentral members of the network without the need for a central custodian; (5) allows parameter merging with equal rights for all members; and (6) protects machine learning models from attacks. Here, we introduce Swarm Learning (SL), which combines decentralized hardware infrastructures, distributed machine learning based on standardized AI engines with a permissioned blockchain to securely onboard members, to dynamically elect the leader among members, and to merge model parameters. Computation is orchestrated by an SL library (SLL) and an iterative AI learning procedure that uses decentral data (Supplementary Information).

## Concept of Swarm Learning

Conceptually, if sufficient data and computer infrastructure are available locally, machine learning can be performed locally (Fig. 1a). In cloud computing, data are moved centrally so that machine learning can be carried out by centralized computing (Fig. 1b), which can substantially increase the amount of data available for training and thereby improve machine learning results[19], but poses disadvantages such as data duplication and increased data traffic as well as challenges for data privacy and security[27]. Federated computing approaches[25] have been developed, wherein dedicated parameter servers are responsible for aggregating and distributing local learning (Fig. 1c); however, a remainder of a central structure is kept.

As an alternative, we introduce SL, which dispenses with a dedicated server (Fig. 1d), shares the parameters via the Swarm network and builds the models independently on private data at the individual sites (short 'nodes' called Swarm edge nodes) (Fig. 1e). SL provides security measures to support data sovereignty, security, and confidentiality

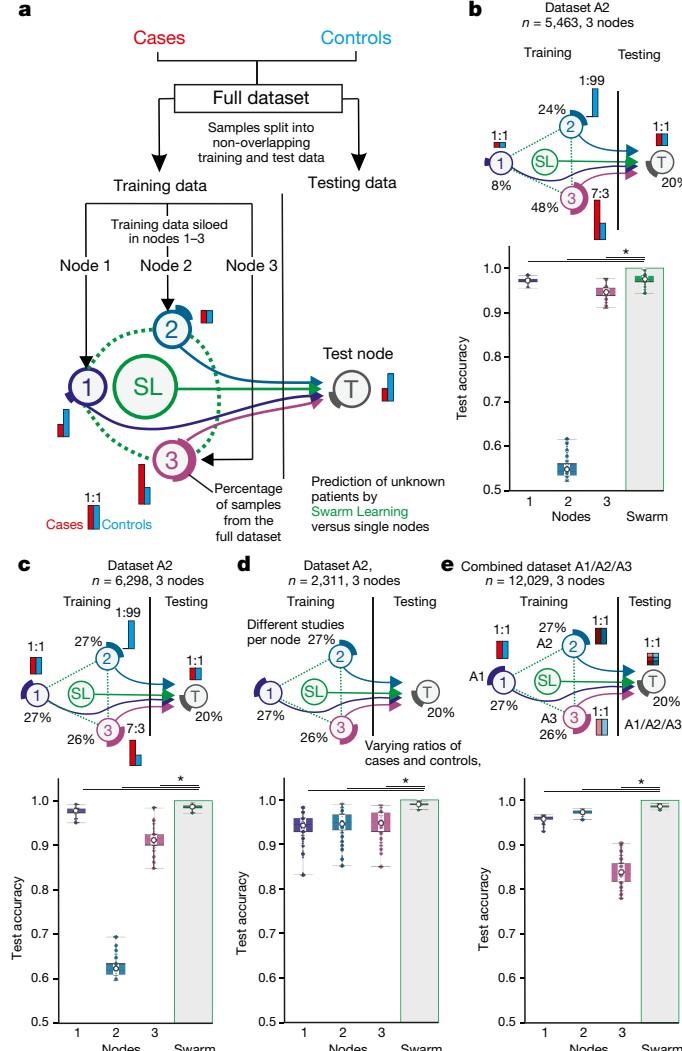

**Fig. 2 | Swarm Learning to predict leukaemias from PBMC data. a**, Overview of the experimental setup. Data consisting of biological replicates are split into non-overlapping training and test sets. Training data are siloed in Swarm edge nodes 1–3 and testing node T is used as independent test set. SL is achieved by integrating nodes 1–3 for training following the procedures described in the Supplementary Information. Red and blue bars illustrate the scenario-specific distribution of cases and controls among the nodes; percentages depict the percentage of samples from the full dataset. **b**, Scenario using dataset A2 with uneven distributions of cases and controls and of samples sizes among nodes. **c**, Scenario with uneven numbers of cases and controls at the different training nodes but similar numbers of samples at each node. **d**, Scenario with samples from independent studies from A2 sampled to different nodes, resulting in varying numbers of cases and controls per node. **e**, Scenario in which each node obtained samples from different transcriptomic technologies (nodes 1–3: datasets A1–A3). The test node obtained samples from each dataset A1–A3. **b**–**e**, Box plots show accuracy of 100 permutations performed for the 3 training nodes individually and for SL. All samples are biological replicates. Centre dot, mean; box limits, 1st and 3rd quartiles; whiskers, minimum and maximum values. Accuracy is defined for the independent fourth node used for testing only. Statistical differences between results derived by SL and all individual nodes including all permutations performed were calculated using one-sided Wilcoxon signed-rank test with continuity correction; *$P < 0.05$, exact $P$ values listed in Supplementary Table 5.

(Extended Data Fig. 1a) realized by private permissioned blockchain technology (Extended Data Fig. 1b). Each participant is well defined and only pre-authorized participants can execute transactions. Onboarding of new nodes is dynamic, with appropriate authorization measures to

recognize network participants. A new node enrolls via a blockchain smart contract, obtains the model, and performs local model training until defined conditions for synchronization are met (Extended Data Fig. 1c). Next, model parameters are exchanged via a Swarm application programming interface (API) and merged to create an updated model with updated parameter settings before starting a new training round (Supplementary Information).

At each node, SL is divided into middleware and an application layer. The application environment contains the machine learning platform, the blockchain, and the SLL (including a containerized Swarm API to execute SL in heterogeneous hardware infrastructures), whereas the application layer contains the models (Extended Data Fig. 1d, Supplementary Information); for example, analysis of blood transcriptome data from patients with leukaemia, tuberculosis and COVID-19 (Fig. 1f–k) or radiograms (Fig. 1l). We selected both heterogeneous and life-threatening diseases to exemplify the immediate medical value of SL.

## Swarm Learning predicts leukaemias

First, we used peripheral blood mononuclear cell (PBMC) transcriptomes from more than 12,000 individuals (Fig. 1f–h) in three datasets (A1–A3, comprising two types of microarray and RNA sequencing (RNA-seq))[3]. If not otherwise stated, we used sequential deep neural networks with default settings[28]. For each real-world scenario, samples were split into non-overlapping training datasets and a global test dataset[29] that was used for testing the models built at individual nodes and by SL (Fig. 2a). Within training data, samples were 'siloed' at each of the Swarm nodes in different distributions, thereby mimicking clinically relevant scenarios (Supplementary Table 1). As cases, we used samples from individuals with acute myeloid leukaemia (AML); all other samples were termed 'controls'. Each node within this simulation could stand for a medical centre, a network of hospitals, a country or any other independent organization that generates such medical data with local privacy requirements.

First, we distributed cases and controls unevenly at and between nodes (dataset A2) (Fig. 2b, Extended Data Fig. 2a, Supplementary Information), and found that SL outperformed each of the nodes (Fig. 2b). The central model performed only slightly better than SL in this scenario (Extended Data Fig. 2b). We obtained very similar results using datasets A1 and A3, which strongly supports the idea that the improvement in performance of SL is independent of data collection (clinical studies) or the technologies (microarray or RNA-seq) used for data generation (Extended Data Fig. 2c–e).

We tested five additional scenarios on datasets A1–A3: (1) using evenly distributed samples at the test nodes with case/control ratios similar to those in the first scenario (Fig. 2c, Extended Data Fig. 2f–j, Supplementary Information); (2) using evenly distributed samples, but siloing samples from particular clinical studies to dedicated training nodes and varying case/control ratios between nodes (Fig. 2d, Extended Data Fig. 3a–h, Supplementary Information); (3) increasing sample size for each training node (Extended Data Fig. 4a–f, Supplementary Information); (4) siloing samples generated with different technologies at dedicated training nodes (Fig. 2e, Extended Data Fig. 4g–i, Supplementary Information); and (5) using different RNA-seq protocols (Extended Data Fig. 4j–k, Supplementary Table 7, Supplementary Information). In all these scenarios, SL outperformed individual nodes and was either close to or equivalent to the central models.

We repeated several of the scenarios with samples from patients with acute lymphoblastic leukaemia (ALL) as cases, extended the prediction to a multi-class problem across four major types of leukaemia, extended the number of nodes to 32, tested onboarding of nodes at a later time point (Extended Data Fig. 5a–j) and replaced the deep neural network with LASSO (Extended Data Fig. 6a–c), and the results echoed the above findings (Supplementary Information).

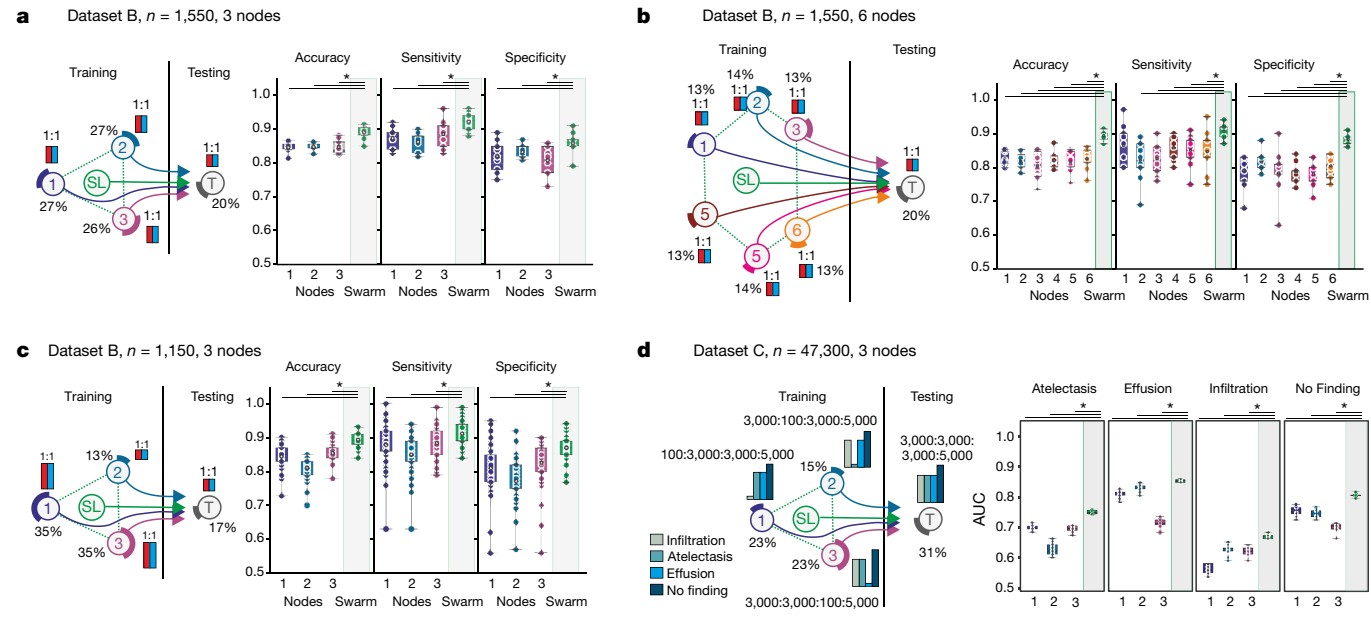

**Fig. 3 | Swarm Learning to identify patients with TB or lung pathologies.**
**a**–**c**, Scenarios for the prediction of TB with experimental setup as in Fig. 2a.
**a**, Scenario with even number of cases at each node; 10 permutations.
**b**, Scenario similar to **a** but with six training nodes; 10 permutations. **c**, Scenario in which the training nodes have evenly distributed numbers of cases and controls at each training node, but node 2 has fewer samples; 50 permutations.
**d**, Scenario for multilabel prediction of dataset C with uneven distribution of diseases at nodes; 10 permutations. **a**–**d**, Box plots show accuracy of all permutations for the training nodes individually and for SL. All samples are biological replicates. Centre dot, mean; box limits, 1st and 3rd quartiles; whiskers, minimum and maximum values. Accuracy is defined for the independent fourth node used for testing only. Statistical differences between results derived by SL and all individual nodes including all permutations performed were calculated with one-sided Wilcoxon signed rank test with continuity correction; *$P < 0.05$, exact $P$ values listed in Supplementary Table 5.

## Swarm Learning to identify tuberculosis

We built a second use case to identify patients with tuberculosis (TB) from blood transcriptomes[30,31] (Fig. 1i, Supplementary Information). First, we used all TB samples (latent and active) as cases and distributed TB cases and controls evenly among the nodes (Extended Data Fig. 7a). SL outperformed individual nodes and performed slightly better than a central model under these conditions (Extended Data Fig. 7b, Supplementary Information). Next, we predicted active TB only. Latently infected TB cases were treated as controls (Extended Data Fig. 7a) and cases and controls were kept even, but the number of training samples was reduced (Fig. 3a). Under these more challenging conditions, overall performance dropped, but SL still performed better than any of the individual nodes. When we further reduced training sample numbers by 50%, SL still outperformed the nodes, but all statistical readouts at nodes and SL showed lower performance; however, SL was still equivalent to a central model (Extended Data Fig. 7c, Supplementary Information), consistent with general observations that AI performs better when training data are increased[19]. Dividing up the training data at three nodes into six smaller nodes reduced the performance of each individual node, whereas the SL results did not deteriorate (Fig. 3b, Supplementary Information).

As TB has endemic characteristics, we used TB to simulate potential outbreak scenarios to identify the benefits and potential limitations of SL and determine how to address them (Fig. 3c, Extended Data Fig. 7d–f, Supplementary Information). The first scenario reflects a situation in which three independent regions (simulated by the nodes) would already have sufficient but different numbers of disease cases (Fig. 3c, Supplementary Information). In this scenario, the results for SL were almost comparable to those in Fig. 3a, whereas the results for node 2 (which had the smallest numbers of cases and controls) dropped noticeably. Reducing prevalence at the test node caused the node results to deteriorate, but the performance of SL was almost unaffected (Extended Data Fig. 7d, Supplementary Information).

We decreased case numbers at node 1 further, which reduced test performance for this node (Extended Data Fig. 7e), without substantially impairing SL performance. When we lowered prevalence at the test node, all performance parameters, including the F1 score (a measure of accuracy), were more resistant for SL than for individual nodes (Extended Data Fig. 7f–j).

We built a third use case for SL that addressed a multi-class prediction problem using a large publicly available dataset of chest X-rays[32] (Figs. 1l, 3d, Supplementary Information, Methods). SL outperformed each node in predicting all radiological findings included (atelectasis, effusion, infiltration and no finding), which suggests that SL is also applicable to non-transcriptomic data spaces.

## Identification of COVID-19

In the fourth use case, we addressed whether SL could be used to detect individuals with COVID-19 (Fig. 1k, Supplementary Table 6). Although COVID-19 is usually detected by using PCR-based assays to detect viral RNA[33], assessing the specific host response in addition to disease prediction might be beneficial in situations for which the pathogen is unknown, specific pathogen tests are not yet possible, existing tests might produce false negative results, and blood transcriptomics can contribute to the understanding of the host's immune response[34–36].

In a first proof-of-principle study, we simulated an outbreak situation node with evenly distributed cases and controls at training nodes and test nodes (Extended Data Fig. 8a, b); this showed very high statistical performance parameters for SL and all nodes. Lowering the prevalence at test nodes reduced performance (Extended Data Fig. 8c), but F1 scores deteriorated only when we reduced prevalence further (1:44 ratio) (Extended Data Fig. 8d); even under these conditions, SL performed best. When we reduced cases at training nodes, all performance measures remained very high at the test node for SL and individual nodes (Extended Data Fig. 8e–j). When we tested outbreak scenarios

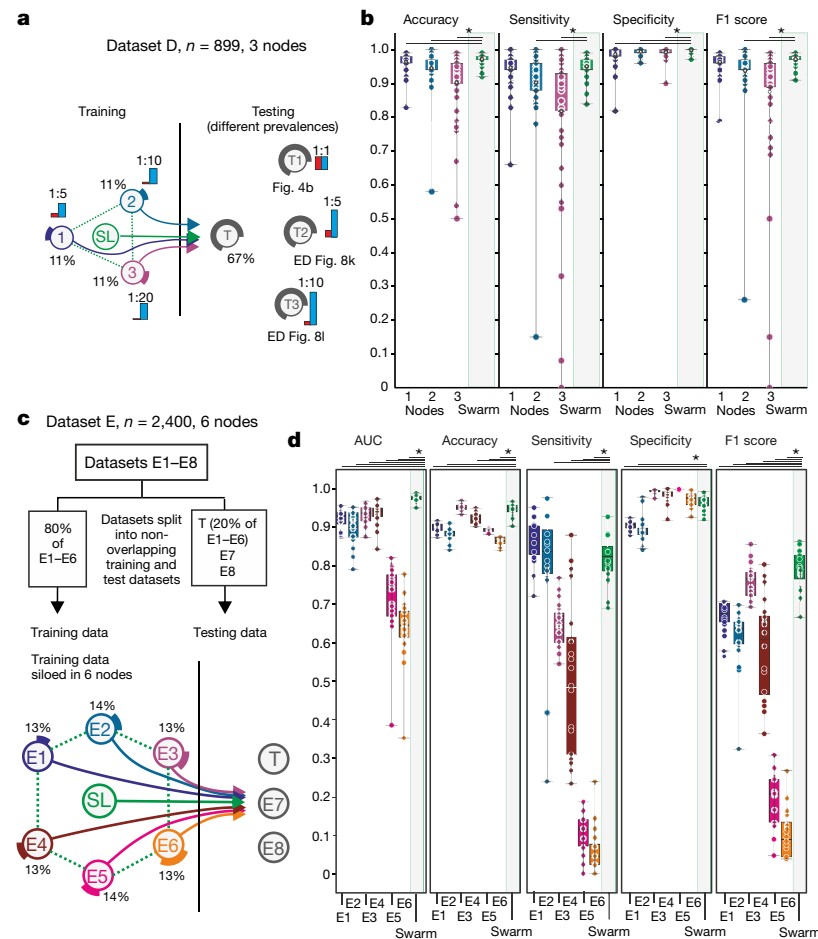

**a**

Dataset D, n = 899, 3 nodes

Training Testing (different prevalences)

**b**

Accuracy Sensitivity Specificity F1 score

**c** Dataset E, n = 2,400, 6 nodes

Datasets E1–E8

80% of E1–E6 Datasets split into non-overlapping training and test datasets T (20% of E1–E6) E7 E8

Training data Testing data

Training data siloed in 6 nodes

**d**

AUC Accuracy Sensitivity Specificity F1 score

**Fig. 4 | Identification of patients with COVID-19 in an outbreak scenario. a**, An outbreak scenario for COVID-19 using dataset D with experimental setup as in Fig. 2a. **b**, Evaluation of **a** with even prevalence showing accuracy, sensitivity, specificity and F1 score of 50 permutations for each training node and SL, on the test node. **c**, An outbreak scenario with dataset E, particularly E1−6 with an 80:20 training:test split. Training data are distributed to six training nodes, independent test data are placed at the test node. **d**, Evaluation of **c** showing AUC, accuracy, sensitivity, specificity and F1 score of 20 permutations. All samples are biological replicates. Centre dot, mean; box limits, 1st and 3rd quartiles; whiskers, minimum and maximum values. Statistical differences between results derived by SL and all individual nodes including all permutations performed were calculated with one-sided Wilcoxon signed-rank test with continuity correction; *P < 0.05, all P values listed in Supplementary Table 5.

with very few cases at test nodes and varying prevalence at the independent test node (Fig. 4a), nodes 2 and 3 showed decreased performance; SL outperformed these nodes (Fig. 4b, Extended Data Fig. 8k, l) and was equivalent to the central model (Extended Data Fig. 8m). The model showed no sign of overfitting (Extended Data Fig. 8n) and comparable results were obtained when we increased the number of training nodes (Extended Data Fig. 9a–d).

We recruited further medical centres in Europe that differed in controls and distributions of age, sex, and disease severity (Supplementary Information), which yielded eight individual centre-specific sub-datasets (E1–8; Extended Data Fig. 9e).

In the first setting, centres E1–E6 teamed up and joined the Swarm network with 80% of their local data; 20% of each centre's dataset was distributed to a test node[29] (Fig. 4c) and the model was also tested on two external datasets, one with convalescent COVID-19 cases (E7) and one of granulocyte-enriched COVID-19 samples (E8). SL outperformed all nodes in terms of area under the curve (AUC) for the prediction of the global test datasets (Fig. 4d, Extended Data Fig. 9f, Supplementary Information). When looking at performance on testing samples split by centre of origin, it became clear that individual centre nodes could not have predicted samples from other centres (Extended Data Fig. 9g). By contrast, SL predicted samples from these nodes successfully. This was similarly true when we reduced the scenario, using E1, E2, and E3 as training nodes and E4 as an independent test node (Extended Data Fig. 9h).

In addition, SL can cope with biases such as sex distribution, age or co-infection bias (Extended Data Fig. 10a–c, Supplementary Information) and SL outperformed individual nodes when distinguishing mild from severe COVID-19 (Extended Data Fig. 10d, e). Collectively, we provide evidence that blood transcriptomes from COVID-19 patients represent a promising feature space for applying SL.

## Discussion

With increasing efforts to enforce data privacy and security[5,9,10] and to reduce data traffic and duplication, a decentralized data model will become the preferred choice for handling, storing, managing, and analysing any kind of large medical dataset[19]. Particularly in oncology, success has been reported in machine-learning-based tumour detection[3,37], subtyping[38], and outcome prediction[39], but progress is hindered by the limited size of datasets[19], with current privacy regulations[5,9,10] making it less appealing to develop centralized AI systems. SL, as a decentralized learning system, replaces the current paradigm of centralized data sharing in cross-institutional medical research. SL's blockchain technology gives robust measures against dishonest participants or adversaries attempting to undermine a Swarm network. SL provides confidentiality-preserving machine learning by design and can inherit new developments in differential privacy algorithms[40], functional encryption[41], or encrypted transfer learning approaches[42] (Supplementary Information).

Global collaboration and data sharing are important quests[13] and both are inherent characteristics of SL, with the further advantage that data sharing is not even required and can be transformed into knowledge sharing, thereby enabling global collaboration with complete data confidentiality, particularly if using medical data. Indeed, statements by lawmakers have emphasized that privacy rules apply fully during a pandemic[43]. Particularly in such crises, AI systems need to comply with ethical principles and respect human rights[12]. Systems such as SL—allowing fair, transparent, and highly regulated shared data analytics while preserving data privacy—are to be favoured. SL should be explored for image-based diagnosis of COVID-19 from patterns in X-ray images or CT scans[15,16], structured health records[12], or data from wearables for disease tracking[12]. Collectively, SL and transcriptomics

(or other medical data) are a very promising approach to democratize the use of AI among the many stakeholders in the domain of medicine, while at the same time resulting in improved data confidentiality, privacy, and data protection, and a decrease in data traffic.

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

¹Systems Medicine, Deutsches Zentrum für Neurodegenerative Erkrankungen (DZNE), Bonn, Germany. ²Genomics and Immunoregulation, Life & Medical Sciences (LIMES) Institute, University of Bonn, Bonn, Germany. ³Hewlett Packard Enterprise, Houston, TX, USA. ⁴Mesh Dynamics, Bangalore, India. ⁵PRECISE Platform for Single Cell Genomics and Epigenomics, Deutsches Zentrum für Neurodegenerative Erkrankungen (DZNE) and the University of Bonn, Bonn, Germany. ⁶Department of Intensive Care Medicine and Radboud Center for Infectious Diseases (RCI), Radboud University Medical Center, Nijmegen, The Netherlands. ⁷Population Health Sciences, Deutsches Zentrum für Neurodegenerative Erkrankungen (DZNE), Bonn, Germany. ⁸Department of Neurology, Faculty of Medicine, University of Bonn, Bonn, Germany. ⁹4th Department of Internal Medicine, National and Kapodistrian University of Athens, Medical School, Athens, Greece. ¹⁰Department of Internal Medicine I, Christian-Albrechts-University and University Hospital Schleswig-Holstein, Kiel, Germany. ¹¹Institute of Clinical Molecular Biology, Christian-Albrechts-University and University Hospital Schleswig-Holstein, Kiel, Germany. ¹²Department of Internal Medicine I, University Hospital, University of Tübingen, Tübingen, Germany. ¹³Institute of Medical Genetics and Applied Genomics, University of Tübingen, Tübingen, Germany. ¹⁴NGS Competence Center Tübingen, Tübingen, Germany. ¹⁵Department of Internal Medicine V, Saarland University Hospital, Homburg, Germany. ¹⁶Department of Pediatrics, Dr. von Hauner Children's Hospital, University Hospital LMU Munich, Munich, Germany. ¹⁷Children's Hospital, Medical Faculty, Technical University Munich, Munich, Germany. ¹⁸Clinical Bioinformatics, Saarland University, Saarbrücken, Germany. ¹⁹Department I of Internal Medicine, Faculty of Medicine and University Hospital of Cologne, University of Cologne, Cologne, Germany. ²⁰Center for Molecular Medicine Cologne (CMMC), University of Cologne, Cologne, Germany. ²¹German Center for Infection Research (DZIF), Partner Site Bonn-Cologne, Cologne, Germany. ²²Cologne Center for Genomics, West German Genome Center, University of Cologne, Cologne, Germany. ²³Clinical Infectious Diseases, Research Center Borstel and German Center for Infection Research (DZIF), Partner Site Hamburg-Lübeck-Borstel-Riems, Borstel, Germany. ²⁴Department of Internal Medicine I, University Hospital Bonn, Bonn, Germany. ²⁵German Center for Infection Research (DZIF), Braunschweig, Germany. ²⁶Department of Internal Medicine II - Cardiology/Pneumology, University of Bonn, Bonn, Germany. ²⁷Institute of Human Genetics, Medical Faculty, RWTH Aachen University, Aachen, Germany. ²⁸Department of Neurology and Neurological Sciences, Stanford University School of Medicine, Stanford, CA, USA. ²⁹Department of Internal Medicine and Radboud Center for Infectious Diseases (RCI), Radboud University Medical Center, Nijmegen, The Netherlands. ³⁰Immunology & Metabolism, Life and Medical Sciences (LIMES) Institute, University of Bonn, Bonn, Germany. ³¹Institute of Computational Biology, Helmholtz Center Munich (HMGU), Neuherberg, Germany. ³²Statistics and Machine Learning, Deutsches Zentrum für Neurodegenerative Erkrankungen (DZNE), Bonn, Germany. ³³CISPA Helmholtz Center for Information Security, Saarbrücken, Germany. ³⁴Institute for Medical Biometry, Informatics and Epidemiology (IMBIE), Faculty of Medicine, University of Bonn, Bonn, Germany. ¹²⁷These authors contributed equally: Stefanie Warnat-Herresthal, Hartmut Schultze, Krishnaprasad Lingadahalli Shastry, Sathyanarayanan Manamohan, Saikat Mukherjee, Vishesh Garg, Ravi Sarveswara, Kristian Händler, Peter Pickkers, N. Ahmad Aziz, Sofia Ktena. ¹²⁸These authors jointly supervised this work: Monique M. B. Breteler, Evangelos J. Giamarellos-Bourboulis, Matthijs Kox, Matthias Becker, Sorin Cheran, Michael S. Woodacre, Eng Lim Goh, Joachim L. Schultze. *Lists of authors and their affiliations appear online. ✉e-mail: joachim.schultze@dzne.de

**COVID-19 Aachen Study (COVAS)**

Paul Balfanz[113], Thomas Eggermann[27], Peter Boor[114], Ralf Hausmann[115], Hannah Kuhn[116], Susanne Isfort[117], Julia Carolin Stingl[118], Günther Schmalzing[118], Christiane K. Kuhl[119], Rainer Röhrig[120], Gernot Marx[121], Stefan Uhlig[122], Edgar Dahl[123,124], Dirk Müller-Wieland[125], Michael Dreher[126] & Nikolaus Marx[125]

[113]Department of Cardiology, Angiology and Intensive Care Medicine, University Hospital RWTH Aachen, Aachen, Germany. [114]Institute of Pathology & Department of Nephrology, University Hospital RWTH Aachen, Aachen, Germany. [115]Institute of Clinical Pharmacology, University Hospital RWTH Aachen, Aachen, Germany. [116]Institute for Biology I, RWTH Aachen University, Aachen, Germany. [117]Department of Hematology, Oncology, Hemostaseology and Stem Cell Transplantation, Medical School, RWTH Aachen University, Aachen, Germany. [118]Institute of Clinical Pharmacology, University Hospital RWTH Aachen, Aachen, Germany. [119]Department of Diagnostic and Interventional Radiology, University Hospital RWTH Aachen, Aachen, Germany. [120]Institute of Medical Informatics, University Hospital RWTH Aachen, Aachen, Germany. [121]Department of Intensive Care, University Hospital RWTH Aachen, Aachen, Germany. [122]Institute of Pharmacology and Toxicology, Medical Faculty Aachen, RWTH Aachen University, Aachen, Germany. [123]Molecular Oncology Group, Institute of Pathology, Medical Faculty, RWTH Aachen University, Aachen, Germany. [124]RWTH centralized Biomaterial Bank (RWTH cBMB) of the Medical Faculty, RWTH Aachen University, Aachen, Germany. [125]Department of Internal Medicine I, University Hospital RWTH Aachen, Aachen, Germany. [126]Department of Pneumology and Intensive Care Medicine, University Hospital RWTH Aachen, Aachen, Germany.

**Deutsche COVID-19 Omics Initiative (DeCOI)**

Janine Altmüller[22], Angel Angelov[14,35], Anna C. Aschenbrenner[1,2,5,29], Robert Bals[15], Alexander Bartholomäus[36], Anke Becker[37], Matthias Becker[1,5,128], Daniela Bezdan[13,14,38], Michael Bitzer[12], Conny Blumert[39], Ezio Bonifacio[40], Peer Bork[41], Bunk Boyke[42], Helmut Blum[43], Nicolas Casadei[13,14], Thomas Clavel[44], Maria Colome-Tatche[31,45,46], Markus Cornberg[47,48,49], Inti Alberto De La Rosa Velázquez[50], Andreas Diefenbach[51], Alexander Dilthey[52], Nicole Fischer[53], Konrad Förstner[54], Sören Franzenburg[11], Julia-Stefanie Frick[14,35], Gisela Gabernet[14,35], Julien Gagneur[56], Tina Ganzenmueller[38], Marie Gauder[14,55], Janina Geißert[14,35], Alexander Goesmann[57], Siri Göpel[12], Adam Grundhoff[23,58], Hajo Grundmann[59], Torsten Hain[60], Frank Hanses[61], Ute Hehr[62], André Heimbach[63], Marius Hoeper[64], Friedemann Horn[39], Daniel Hübschmann[65,66,67], Michael Hummel[68,69], Thomas Iftner[38], Angelika Iftner[38], Thomas Illig[70], Stefan Janssen[71], Jörn Kalinowski[72], René Kallies[73], Birte Kehr[18,28], Andreas Keller[31,45,76], Oliver T. Keppler[75,76], Sarah Kim-Hellmuth[16], Christoph Klein[16], Michael Knop[77,78], Oliver Kohlbacher[79,80], Karl Köhrer[81], Jan Korbel[41], Peter G. Kremsner[82], Denise Kühnert[83], Ingo Kurth[27], Markus Landthaler[84], Yang Li[85], Kerstin U. Ludwig[63], Oliwia Makarewicz[86], Manja Marz[87,88], Alice C. McHardy[89], Christian Mertes[56], Maximilian Münchhoff[75,76], Sven Nahnsen[14,55], Markus Nöthen[63], Francine Ntoumi[90], Peter Nürnberg[22], Stephan Ossowski[13,14], Jörg Overmann[42], Silke Peter[14,35], Klaus Pfeffer[52], Isabell Pink[47], Anna R. Poetsch[91], Ulrike Protzer[92], Alfred Pühler[72], Nikolaus Rajewsky[84], Markus Ralser[93], Kristin Reiche[39], Olaf Rieß[13,14], Stephan Ripke[94], Ulisses Nunes da Rocha[73], Philip Rosenstiel[11], Antoine-Emmanuel Saliba[95], Leif Erik Sander[96], Birgit Sawitzki[97], Simone Scheithauer[98], Philipp Schiffer[99], Jonathan Schmid-Burgk[100], Wulf Schneider[61], Eva-Christina Schulte[101], Joachim L. Schultze[1,2,5,128], Alexander Sczyrba[72], Mariam L. Sharaf[2], Yogesh Singh[13,14], Michael Sonnabend[14,35], Oliver Stegle[41,102], Jens Stoye[103], Fabian Theis[31], Thomas Ulas[1,2], Janne Vehreschild[21,104,105], Thirumalaisamy P. Velavan[90], Jörg Vogel[95], Sonja Volland[70], Max von Kleist[106,107], Andreas Walker[108], Jörn Walter[109], Dagmar Wieczorek[110], Sylke Winkler[111] & John Ziebuhr[112]

[35]Institute of Medical Microbiology and Hygiene, University of Tübingen, Tübingen, Germany. [36]Geomicrobiology, German Research Centre for Geosciences (GFZ), Potsdam, Germany. [37]LOEWE Center for Synthetic Microbiology (SYNMIKRO), Philipps-Universität Marburg, Marburg, Germany. [38]Institute for Medical Virology and Epidemiology of Viral Diseases, University of Tübingen, Tübingen, Germany. [39]Fraunhofer Institute for Cell Therapy and Immunology (IZI), Leipzig, Germany. [40]Centre for Regenerative Therapies Dresden (CRTD), Dresden, Germany. [41]European Molecular Biology Laboratory (EMBL), Heidelberg, Germany. [42]DSMZ - German Collection of Microorganisms and Cell Cultures, Leibniz Institute, Braunschweig, Germany. [43]Gene Center - Functional Genomics Analysis, Ludwig-Maximilians-Universität München, München, Germany. [44]Institute for Medical Microbiology, University Hospital Aachen, RWTH Aachen, Germany. [45]European Research Institute for the Biology of Ageing, University of Groningen, Groningen, The Netherlands. [46]TUM School of Life Sciences Weihenstephan, Technical University of Munich, Freising, Germany. [47]Klinik für Gastroenterologie, Hepatologie und Endokrinologie, Medizinische Hochschule Hannover (MHH), Hannover, Germany. [48]Center for Individualised Infection Medicine (CiiM), Hannover, Germany. [49]German Center for Infection Research (DZIF), Hannover, Germany. [50]Genome Analysis Center, Helmholtz Zentrum München Deutsches Forschungszentrum für Gesundheit und Umwelt, Neuherberg, Germany. [51]Institut für Mikrobiologie und Infektionsimmunologie, Charité – Universitätsmedizin Berlin, Berlin, Germany. [52]Institut für Medizinische Mikrobiologie und Krankenhaushygiene, Universitätsklinikum Düsseldorf, Heinrich-Heine-Universität Düsseldorf, Düsseldorf, Germany. [53]Institut für Medizinische Mikrobiologie, Virologie und Hygiene, Universitätsklinikum Hamburg- Eppendorf (UKE), Hamburg, Germany. [54]German Information Centre for Life Sciences (ZB MED), Cologne, Germany. [55]Quantitative Biology Center, University of Tübingen, Tübingen, Germany. [56]Informatik 29 - Computational Molecular Medicine, Technische Universität München, München, Germany. [57]Bioinformatics and Systems Biology, Justus Liebig University Giessen, Giessen, Germany. [58]Leibniz Institut für Experimentelle Virologie, Hamburg, Germany. [59]Institute for Infection Prevention and Hospital Hygiene, Universitätsklinikum Freiburg, Freiburg, Germany. [60]Institute of Medical Microbiology, Justus Liebig University Giessen, Giessen, Germany. [61]Krankenhaushygiene und Infektiologie, Universitätsklinikum Regensburg, Regensburg, Germany. [62]Zentrum für Humangenetik Regensburg, Regensburg, Germany. [63]Institute of Human Genetics, University of Bonn, School of Medicine & University Hospital Bonn, Bonn, Germany. [64]Klinik für Pneumologie, Medizinische Hochschule Hannover (MHH), Hannover, Germany. [65]Computational Oncology, Molecular Diagnostics Program, National Center for Tumor Diseases (NCT) Heidelberg and German Cancer Research Center (DKFZ), Heidelberg, Germany. [66]Heidelberg Institute for Stem Cell Technology and Experimental Medicine (HI-STEM), Heidelberg, Germany. [67]German Cancer Consortium (DKTK), Heidelberg, Germany. [68]Institute for Pathology, Molecular Pathology, Charité – Universitätsmedizin Berlin, Berlin, Germany. [69]German Biobank Node (bbmri.de), Berlin, Germany. [70]Medizinische Hochschule Hannover (MHH), Hannover Unified Biobank and Institute of Human Genetics, Hannover, Germany. [71]Algorithmic Bioinformatics, Justus Liebig University Giessen, Giessen, Germany. [72]Center for Biotechnology (CeBiTec), Bielefeld University, Bielefeld, Germany. [73]Department of Environmental Microbiology, Helmholtz-Zentrum für Umweltforschung (UFZ), Leipzig, Germany. [74]Algorithmische Bioinformatik, RCI Regensburger Centrum für Interventionelle Immunologie, Universitätsklinikum Regensburg, Regensburg, Germany. [75]Max von Pettenkofer Institute & Gene Center, Virology, National Reference Center for Retroviruses, LMU München, Munich, Germany. [76]German Center for Infection Research (DZIF), partner site Munich, München, Germany. [77]Center for Molecular Biology (ZMBH), Heidelberg University, Heidelberg, Germany. [78]Cell Morphogenesis and Signal Transduction, German Cancer Research Center (DKFZ), Heidelberg, Germany. [79]Applied Bioinformatics, University of Tübingen, Tübingen, Germany. [80]Translational Bioinformatics, University Hospital, University of Tübingen, Tübingen, Germany. [81]Genomics & Transcriptomics Labor (GTL), Universitätsklinikum Düsseldorf, Heinrich-Heine-Universität Düsseldorf, Düsseldorf, Germany. [82]Medical Clinic Internal Medicine VII, University Hospital, University of Tübingen, Tübingen, Germany. [83]Transmission, Infection, Diversification and Evolution Group, Max Planck Institute for the Science of Human History, Jena, Germany. [84]Berlin Institute for Medical Systems Biology, Max Delbrück Center for Molecular Medicine in the Helmholtz Association (MDC), Berlin, Germany. [85]Centre for Individualized Infection Medicine (CiiM) & TWINCORE, joint ventures between the Helmholtz-Centre for Infection Research (HZI) and the Hannover Medical School (MHH), Hannover, Germany. [86]Institute for Infection Medicine and Hospital Hygiene (IIMK), Uniklinikum Jena, Jena, Germany. [87]Michael Stifel Center Jena, Jena, Germany. [88]Bioinformatics/High-Throughput Analysis, Faculty of Mathematics and Computer Science, Friedrich-Schiller-Universität Jena, Jena, Germany. [89]Computational Biology for Infection Research, Helmholtz Centre for Infection Research (HZI), Brunswick, Germany. [90]Institute for Tropical Medicine, University Hospital, University of Tübingen, Tübingen, Germany. [91]Biotechnology Center (BIOTEC) TU Dresden, National Center for Tumor Diseases, Dresden, Germany. [92]Institute of Virology, Technical University of Munich, Munich, Germany. [93]Institute of Biochemistry, Charité – Universitätsmedizin Berlin, Berlin, Germany. [94]Department of Psychiatry and Neurosciences, Charité – Universitätsmedizin Berlin, Berlin, Germany. [95]Helmholtz Institute for RNA-based Infection Research (HIRI), Helmholtz-Center for Infection Research, Würzburg, Germany. [96]Department of Internal Medicine with emphasis on Infectiology, Respiratory-, and Critical-Care-Medicine, Charité – Universitätsmedizin Berlin, Berlin, Germany. [97]Institute of Medical Immunology, Charité – Universitätsmedizin Berlin, Berlin, Germany. [98]Institute of Infection Control and Infectious Diseases, University Medical Center, Georg August University, Göttingen, Germany. [99]Institute of Zoology, University of Cologne, Cologne, Germany. [100]Institute of Clinical Chemistry and Clinical Pharmacology, University Hospital, University of Bonn, Bonn, Germany. [101]Klinik für Psychiatrie und Psychotherapie and Institut für Psychiatrische Phänomik und Genomik, LMU München, Munich, Germany. [102]Division of Computational Genomics and Systems Genetics, German Cancer Research Center (DKFZ), Heidelberg, Germany. [103]Genome Informatics, University of Bielefeld, Bielefeld, Germany. [104]Department I of Internal Medicine, University Hospital of Cologne, University of Cologne, Cologne, Germany. [105]University Hospital Frankfurt, Frankfurt am Main, Germany. [106]Institute for Bioinformatics, Freie Universität Berlin, Berlin, Germany. [107]Robert Koch Institute, Berlin, Germany. [108]Institut für Virologie, Universitätsklinikum Düsseldorf, Heinrich-Heine-Universität Düsseldorf, Düsseldorf, Germany. [109]Genetics and Epigenetics, Saarland University, Saarbrücken, Germany. [110]Institut für Humangenetik, Universitätsklinikum Düsseldorf, Heinrich-Heine-Universität Düsseldorf, Düsseldorf, Germany. [111]Max Planck Institute of Molecular Cell Biology and Genetics, Dresden, Germany and DRESDEN concept Genome Center, TU Dresden, Dresden, Germany. [112]Institute of Medical Virology, Justus Liebig University Giessen, Giessen, Germany.

## Methods

### Pre-processing

**PBMC transcriptome dataset (dataset A).** We used a previously published dataset compiled for predicting AML in blood transcriptomes derived from PBMCs (Supplementary Information)[3]. In brief, all raw data files were downloaded from GEO (https://www.ncbi.nlm.nih.gov/geo/) and the RNA-seq data were preprocessed using the kallisto v0.43.1 aligner against the human reference genome gencode v27 (GRCh38.p10). For normalization, we considered all platforms independently, meaning that normalization was performed separately for the samples in datasets A1, A2 and A3. Microarray data (datasets A1 and A2) were normalized using the robust multichip average (RMA) expression measures, as implemented in the R package affy v.1.60.0. The RNA-seq data (dataset A3) were normalized using the R package DESeq2 (v 1.22.2) with standard parameters. To keep the datasets comparable, data were filtered for genes annotated in all three datasets, which resulted in 12,708 genes. No filtering of low-expressed genes was performed. All scripts used in this study for pre-processing are provided as a docker container on Docker Hub (v 0.1, https://hub.docker.com/r/schultzelab/aml_classifier).

**Whole-blood-derived transcriptome datasets (datasets B, D and E).** As alignment of whole blood transcriptome data can be performed in many ways, we re-aligned all downloaded and collected datasets (Supplementary Information; these were 30.6 terabytes in size and comprised a total of 63.4 terabases) to the human reference genome gencode v33 (GRCh38.p13) and quantified transcript counts using STAR, an ultrafast universal RNA-seq aligner (v.2.7.3a). For all samples in datasets B, D, and E, raw counts were imported using DESeq (v.1.22.2, DESeqDataSetFromMatrix function) and size factors for normalization were calculated using the DESeq function with standard parameters. This was done separately for datasets B, D, and E. As some of the samples were prepared with poly-A selection to enrich for protein-coding mRNAs, we filtered the complete dataset for protein-coding genes to ensure greater comparability across library preparation protocols. Furthermore, we excluded all ribosomal protein-coding genes, as well as mitochondrial genes and genes coding for haemoglobins, which resulted in 18,135 transcripts as the feature space in dataset B, 19,358 in dataset D and 19,399 in dataset E. Furthermore, transcripts with overall expression <100 were excluded from further analysis. Other than that, no filtering of transcripts was performed. Before using the data in machine learning, we performed a rank transformation to normality on datasets B, D and E. In brief, transcript expression values were transformed from RNA-seq counts to their ranks. This was done transcript-wise, meaning that all transcript expression values per sample were given a rank based on ordering them from lowest to highest value. The rankings were then turned into quantiles and transformed using the inverse cumulative distribution function of the normal distribution. This leads to all transcripts following the exact same distribution (that is, a standard normal with a mean of 0 and a standard deviation of 1 across all samples). All scripts used in this study for pre-processing are provided on Github (https://github.com/schultzelab/swarm_learning) and normalized and rank-transformed count matrices used for predictions are provided via FASTGenomics at https://beta.fastgenomics.org/p/swarm-learning.

**X-ray dataset (dataset C).** The National Institutes of Health (NIH) chest X-Ray dataset (Supplementary Information) was downloaded from https://www.kaggle.com/nih-chest-xrays/data[32]. To preprocess the data, we used Keras (v.2.3.1) real-time data augmentation and generation APIs (keras.preprocessing.image.ImageDataGenerator and flow_from_dataframe). The following pre-processing arguments were used: height or width shift range (about 5%), random rotation range (about 5°), random zoom range (about 0.15), sample-wise centre and standard normalization. In addition, all images were resized to 128 × 128 pixels from their original size of 1,024 × 1,024 pixels and 32 images per batch were used for model training.

### The Swarm Learning framework

SL builds on two proven technologies, distributed machine learning and blockchain (Supplementary Information). The SLL is a framework to enable decentralized training of machine learning models without sharing the data. It is designed to make it possible for a set of nodes—each node possessing some training data locally—to train a common machine learning model collaboratively without sharing the training data. This can be achieved by individual nodes sharing parameters (weights) derived from training the model on the local data. This allows local measures at the nodes to maintain the confidentiality and privacy of the raw data. Notably, in contrast to many existing federated learning models, a central parameter server is omitted in SL. Detailed descriptions of the SLL, the architecture principles, the SL process, implementation, and the environment can be found in the Supplementary Information.

### Hardware architecture used for simulations

For all simulations provided in this project we used two HPE Apollo 6500 Gen 10 servers, each with four Intel(R) Xeon(R) CPU E5-2698 v4 @ 2.20 GHz, a 3.2-terabyte hard disk drive, 256 GB RAM, eight Tesla P100 GPUs, a 1-GB network interface card for LAN access and an InfiniBand FDR for high speed interconnection and networked storage access. The Swarm network is created with a minimum of 3 up to a maximum of 32 training nodes, and each node is a docker container with access to GPU resources. Multiple experiments were run in parallel using this configuration.

Overall, we performed 16,694 analyses including 26 scenarios for AML, four scenarios for ALL, 13 scenarios for TB, one scenario for detection of atelectasis, effusion, and/or infiltration in chest X-rays, and 18 scenarios for COVID-19 (Supplementary Information). We performed 5–100 permutations per scenario and each permutation took approximately 30 min, which resulted in a total of 8,347 computer hours.

### Computation and algorithms

**Neural network algorithm.** We leveraged a deep neural network with a sequential architecture as implemented in Keras (v 2.3.1)[28]. Keras is an open source software library that provides a Python interface to neural networks. The Keras API was developed with a focus on fast experimentation and is standard for deep learning researchers. The model, which was already available in Keras for R from the previous study[3], has been translated from R to Python to make it compatible with the SLL (Supplementary Information). In brief, the neural network consists of one input layer, eight hidden layers and one output layer. The input layer is densely connected and consists of 256 nodes, a rectified linear unit activation function and a dropout rate of 40%. From the first to the eighth hidden layer, nodes are reduced from 1,024 to 64 nodes, and all layers contain a rectified linear unit activation function, a kernel regularization with an L2 regularization factor of 0.005 and a dropout rate of 30%. The output layer is densely connected and consists of one node and a sigmoid activation function. The model is configured for training with Adam optimization and to compute the binary cross-entropy loss between true labels and predicted labels.

The model is used for training both the individual nodes and SL. The model is trained over 100 epochs, with varying batch sizes. Batch sizes of 8, 16, 32, 64 and 128 are used, depending on the number of training samples. The full code for the model is provided on Github (https://github.com/schultzelab/swarm_learning/)

**Least absolute shrinkage and selection operator (LASSO).** SL is not restricted to any particular classification algorithm. We therefore adapted the l1-penalized logistic regression[3] to be used with the SLL in the form of a Keras single dense layer with linear activation. The regularization

parameter lambda was set to 0.01. The full code for the model is provided on Github (https://github.com/schultzelab/swarm_learning/)

**Parameter tuning.** For most scenarios, default settings were used without parameter tuning. For some of the scenarios we tuned model hyperparameters. For some scenarios we also tuned SL parameters to get better performance (for example, higher sensitivity) (Supplementary Table 8). For example, for AML (Fig. 2e, f, Extended Data Fig. 2), the dropout rate was reduced to 10% to get better performance. For AML (Fig. 2b), the dropout rate was reduced to 10% and the epochs increased to 300 to get better performance. We also used the adaptive_rv parameter in the SL API to adjust the merge frequency dynamically on the basis of model convergence, to improve the training time. For TB and COVID-19, the test dropout rate was reduced to 10% for all scenarios. For the TB scenarios (Extended Data Fig. 7f, g), the node_weightage parameter of the SL callback API was used to give more weight to nodes that had more case samples. Supplementary Table 8 provides a complete overview of all tuning parameters used.

**Parameter merging.** Different functions are available for parameter merging as a configuration of the Swarm API, which are then applied by the leader at every synchronization interval. The parameters can be merged as average, weighted average, minimum, maximum, or median functions.

In this Article, we used the weighted average, which is defined as

$$P_M = \frac{\sum_{k=1}^{n}(W_k \times P_k)}{n \times \sum_{k=1}^{n} W_k}$$

in which $P_M$ is merged parameters, $P_k$ is parameters from the $k$th node, $W_k$ is the weight of the $k$th node, and $n$ is the number of nodes participating in the merge process.

Unless stated otherwise, we used a simple average without weights to merge the parameter for neural networks and for the LASSO algorithm.

## Quantification and statistical analysis

We evaluated binary classification model performance with sensitivity, specificity, accuracy, F1 score, and AUC metrics, which were determined for every test run. The 95% confidence intervals of all performance metrics were estimated using bootstrapping. For AML and ALL, 100 permutations per scenario were run for each scenario. For TB, the performance metrics were collected by running 10 to 50 permutations. For the X-ray images, 10 permutations were performed. For COVID-19 the performance metrics were collected by running 10 to 20 permutations for each scenario. All metrics are listed in Supplementary Tables 3, 4.

Differences in performance metrics were tested using the one-sided Wilcoxon signed rank test with continuity correction. All test results are provided in Supplementary Table 5.

To run the experiments, we used Python version 3.6.9 with Keras version 2.3.1 and TensorFlow version 2.2.0-rc2. We used scikit-learn library version 0.23.1 to calculate values for the metrics. Summary statistics and hypothesis tests were calculated using R version 3.5.2. Calculation of each metric was done as follows:

$$\text{Sensitivity} = \frac{TP}{TP + FN}$$

$$\text{Specificity} = \frac{TN}{TN + FP}$$

$$\text{Accuracy} = \frac{TP + TN}{TP + FP + TN + FN}$$

$$\text{F1score} = \frac{2TP}{FP + FN + 2TP}$$

where TP is true positive, FP is false positive, TN is true negative and FN is false negative. The area under the ROC curve was calculated using the R package ROCR version 1.0-11.

No statistical methods were used to predetermine sample size. The experiments were not randomized, but permutations were performed. Investigators were not blinded to allocation during experiments and outcome assessment.

## Reporting summary

Further information on research design is available in the Nature Research Reporting Summary linked to this paper.

## Data availability

Processed data from datasets A1–A3 can be accessed from GEO via the superseries GSE122517 or the individual subseries GSE122505 (dataset A1), GSE122511 (dataset A2) and GSE122515 (dataset A3). Dataset B consists of the following series, which can be accessed at GEO: GSE101705, GSE107104, GSE112087, GSE128078, GSE66573, GSE79362, GSE84076, and GSE89403. Furthermore, it contains the data from the Rhineland Study. The Rhineland Study dataset falls under current General Data Protection Regulations (GDPR). Access to these data can be provided to scientists in accordance with the Rhineland Study's Data Use and Access Policy. Requests to access the Rhineland Study's dataset should be directed to RS-DUAC@dzne.de. New samples generated for datasets D and E have been deposited at the European Genome-Phenome Archive (EGA), which is hosted by the EBI and the CRG, under accession number EGAS00001004502. The healthy RNA-seq data included from Saarbrücken are available on application from PPMI through the LONI data archive at https://www.ppmi-info.org/data. Samples received from other public repositories are listed in Supplementary Table 2. Dataset C (NIH chest X-ray dataset) is available on Kaggle (https://www.kaggle.com/nih-chest-xrays/data). Normalized log-transformed and rank transformed expressions as used for the predictions are available via FASTGenomics at https://beta.fastgenomics.org/p/swarm-learning.

## Code availability

The code for preprocessing and for predictions can be found at GitHub (https://github.com/schultzelab/swarm_learning). The Swarm Learning software can be downloaded from https://myenterpriselicense.hpe.com/.

**Acknowledgements** We thank the Michael J. Fox Foundation and the Parkinson's Progression Markers Initiative (PPMI) for contributing RNA-seq data; the CORSAAR study group for additional blood transcriptome samples; the collaborators who contributed to the collection of COVID-19 samples (B. Schlegelberger, I. Bernemann, J. C. Hellmuth, L. Jocham, F. Hanses, U. Hehr, Y. Khodamoradi, L. Kaldjob, R. Fendel, L. T. K. Linh, P. Rosenberger, H. Häberle and J. Böhne); and the NGS Competence Center Tübingen (NCCT), who contributed to the generation of data and the data sharing (in particular, J. Frick, M. Sonnabend, J. Geissert, A. Angelov, M. Pogoda, Y. Singh, S. Poths, S. Nahnsen and M. Gauder). This work was supported in part by the German Research Foundation (DFG) to J.L.S., O.R., P.R., P.N. (INST 37/1049-1, INST 216/981-1, INST 257/605-1, INST 269/768-1, INST 217/988-1, INST 217/577-1, INST 217/1011-1, INST 217/1017-1 and INST 217/1029-1); under Germany's Excellence Strategy (DFG – EXC2151 – 390873048); by the HGF Incubator grant sparse2big (ZT-I-O007); by EU projects SYSCID (grant 733100, P.R.) and ImmunoSep (grant 84722, J.L.S.); and by HPE to the DZNE for generating whole blood transcriptome data from patients with COVID-19. J.L.S. was further supported by the BMBF-funded excellence project Diet–Body–Brain (DietBB) (grant 01EA1809A), and J.L.S. and J.R. by NaFoUniMedCovid19 (FKZ: 01KX2021, project acronym COVIM). S.K. is supported by the Hellenic Institute for the Study of Sepsis. The clinical study in Greece was supported by the Hellenic Institute for the Study of Sepsis. E.J.G.-B. received funding from the FrameWork 7 programme HemoSpec (granted to the National and Kapodistrian University of Athens), the Horizon2020 Marie-Curie Project European Sepsis Academy (676129, granted to the National and Kapodistrian University of Athens), and the Horizon 2020 European Grant ImmunoSep (granted to the Hellenic Institute for the Study of Sepsis). P.R. was supported by DFG ExC2167, a stimulus fund from Schleswig-Holstein and the DFG NGS Centre CCGA. The clinical study in Munich was supported by the Care-for-Rare Foundation. S.K.-H. is a scholar of the Reinhard-Frank Stiftung. D.P. is funded by the Hector Fellow Academy. The work was additionally supported by the Michael J. Fox Foundation for Parkinson' Research under grant 14446. M.G.N. was supported by an ERC Advanced Grant (833247) and a Spinoza Grant of the Netherlands Organization for Scientific Research. R.B. and A.K. were

supported by Dr. Rolf M. Schwiete Stiftung, Staatskanzlei des Saarlandes and Saarland University. J.N. is supported by the DFG (SFB TR47, SPP1937) and the Hector Foundation (M88). M.A. is supported by COVIM: NaFoUniMedCovid19 (FKZ: 01KX2021). M. Becker is supported by the HGF  Helmholtz AI grant Pro-Gene-Gen (ZT-I-PF-5-23).

**Author contributions** The idea was conceived by H.S., K.L.S., E.L.G., and J.L.S. Subprojects and clinical studies were directed by H.S., K.L.S., K.H., M. Bitzer, J.R., S.K.-H., J.N., I.K., A.K., R.B., P.N., O.R., P.R., M.M.B.B., M. Becker, and J.L.S. The conceptualization was performed by S.W.-H., H.S., K.L.S., M. Becker, S.C., M.S.W., E.L.G., and J.L.S. Direction of the clinical programs, collection of clinical information and patient diagnostics were done by P.P., N.A.A., S.K., F.T., M. Bitzer, C.H., D.P., U.B., F.K., T.F., P.S., C.L., M.A., J.R., B.K., M.S., J.H., S.S., S.K.-H., J.N., D.S., I.K., A.K., R.B., M.G.N., M.M.B.B., E.J.G.-B, and M.K. Patient samples were provided by P.P., N.A.A., S.K., F.T., M. Bitzer, S.O., N.C., C.H., D.P., U.B., F.K., T.F., P.S, C.L., M.A., J.R., B.K., M.S., J.H., S.S., S.K.-H., J.N., D.S., I.K., A.K., R.B., M.G.N., M.M.B.B., E.J.G-B, and M.K. Laboratory experiments were performed by K.H., S.O., N.C., J.A., L.B., J.S.-S., E.D.D., M.K., and H.T. Primary data analysis and data QC were provided by S.W.-H., K.H., S.O., N.C., J.A., N.M., J.P.B., L.B., J.S.-S., E.D.D., M.N.-G., A.K., P.N., O.R., P.R., T.U., M. Becker, and J.L.S. Programming and coding for the current project were done by S.W.-H., Saikat Mukherjee, V.G., R.S., C.S., M.D., C.M.S., and M. Becker. The Swarm Learning environment was developed by S. Manamohan, Saikat Mukherjee, V.G., R.S., M.D., B.M., S.C., M.S.W., and E.L.G. Statistics and machine learning were done by S.W.-H., Saikat Mukherjee, V.G., R.S., M.D., F.T., Sach Mukherjee, S.C., E.L.G., and J.L.S. Data privacy and confidentiality concepts were developed by H.S., K.L.S., M. Backes, E.L.G., and J.L.S. Data interpretation was done by S.W.-H., H.S., Saikat Mukherjee, A.C.A., M. Becker, and J.L.S. Data were visualized by S.W.-H., H.S., M. Becker, and J.L.S. The original draft was written by S.W.-H., H.S., K.L.S., A.C.A., M. Becker, and J.L.S. Writing, reviewing and editing of revisions was done by S.W.-H., H.S., K.L.S., A.C.A., M.M.B.B., M. Becker, E.L.G., and J.L.S. Project management and administration were performed by H.S., K.L.S., A.D., A.C.A., M. Becker, and J.L.S. Funding was acquired by H.S., S.K., D.P., M.A., J.R., S.K.-H., J.N., A.K., R.B., P.N., O.R., P.R., M.G.N., F.T., E.J.G.-B, M.B., S.C., and J.L.S. All authors commented on the manuscript.

**Funding** Open access funding provided by Deutsches Zentrum für Neurodegenerative Erkrankungen e.V. (DZNE) in der Helmholtz-Gemeinschaft.

**Competing interests** H.S., K.L.S., S. Manamohan, Saikat Mukherjee, V.G., R.S., C.S., M.D., B.M, C.M.S., S.C., M.S.W. and E.L.G. are employees of Hewlett Packard Enterprise. Hewlett Packard Enterprise developed the SLL in its entirety as described in this work and has submitted multiple associated patent applications. E.J.G.-B. received honoraria from AbbVie USA, Abbott CH, InflaRx GmbH, MSD Greece, XBiotech Inc. and Angelini Italy and independent educational grants from AbbVie, Abbott, Astellas Pharma Europe, AxisShield, bioMérieux Inc, InflaRx GmbH, and XBiotech Inc. All other authors declare no competing interests.

## Additional information
**Correspondence and requests for materials** should be addressed to J.L.S.

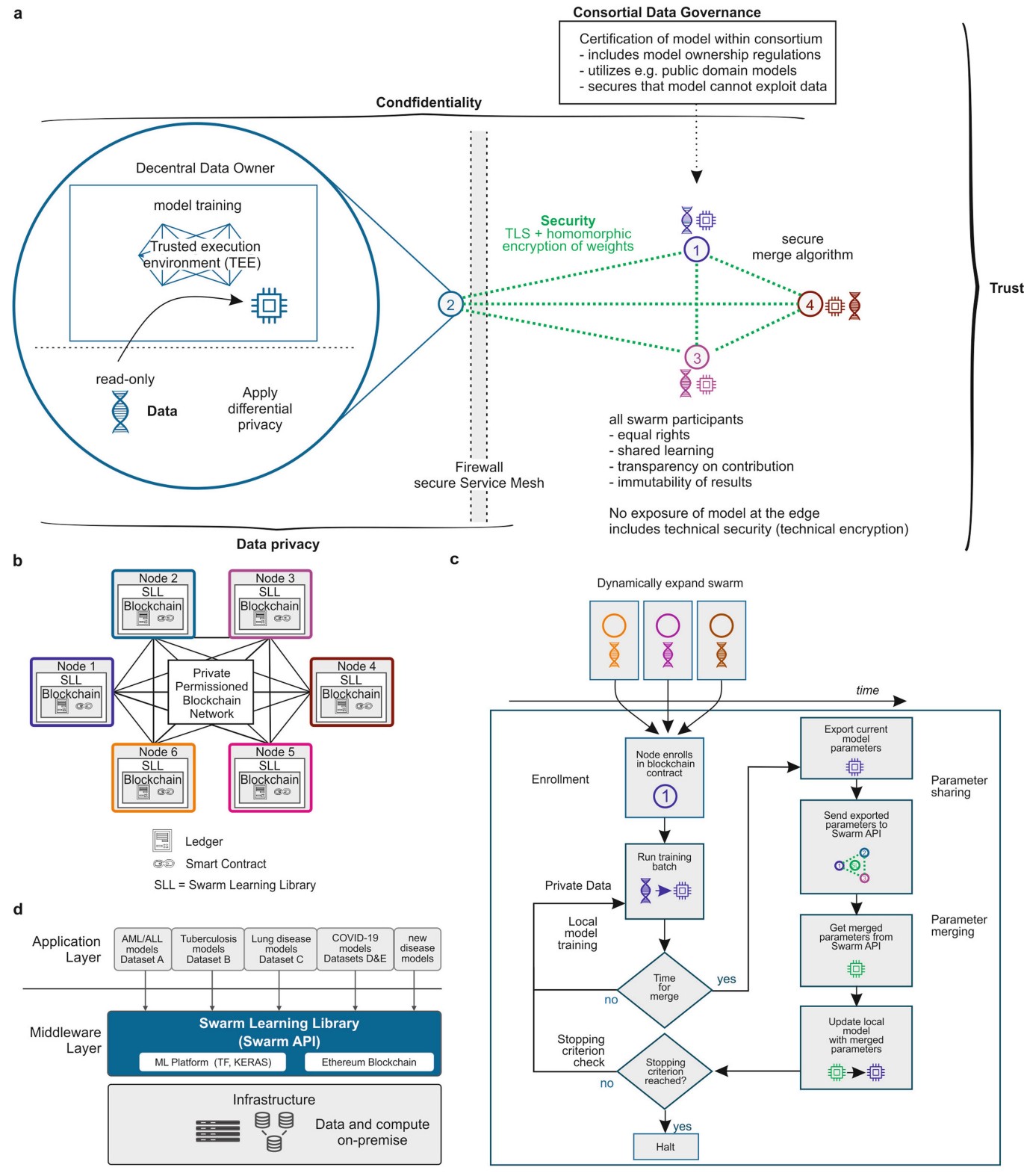

**Extended Data Fig. 1 | Corresponding to Fig. 1. a**, Overview of SL and the relationship to data privacy, confidentiality and trust. **b**, Concept and outline of the private permissioned blockchain network as a layer of the SL network. Each node consists of the blockchain, including the ledger and smart contract, as well as the SLL with the API to interact with other nodes within the network. **c**, The principles of the SL workflow once the nodes have been enrolled within the Swarm network via private permissioned blockchain contract and dynamic onboarding of new Swarm nodes. **d**, Application and middleware layer as part of the SL concept.

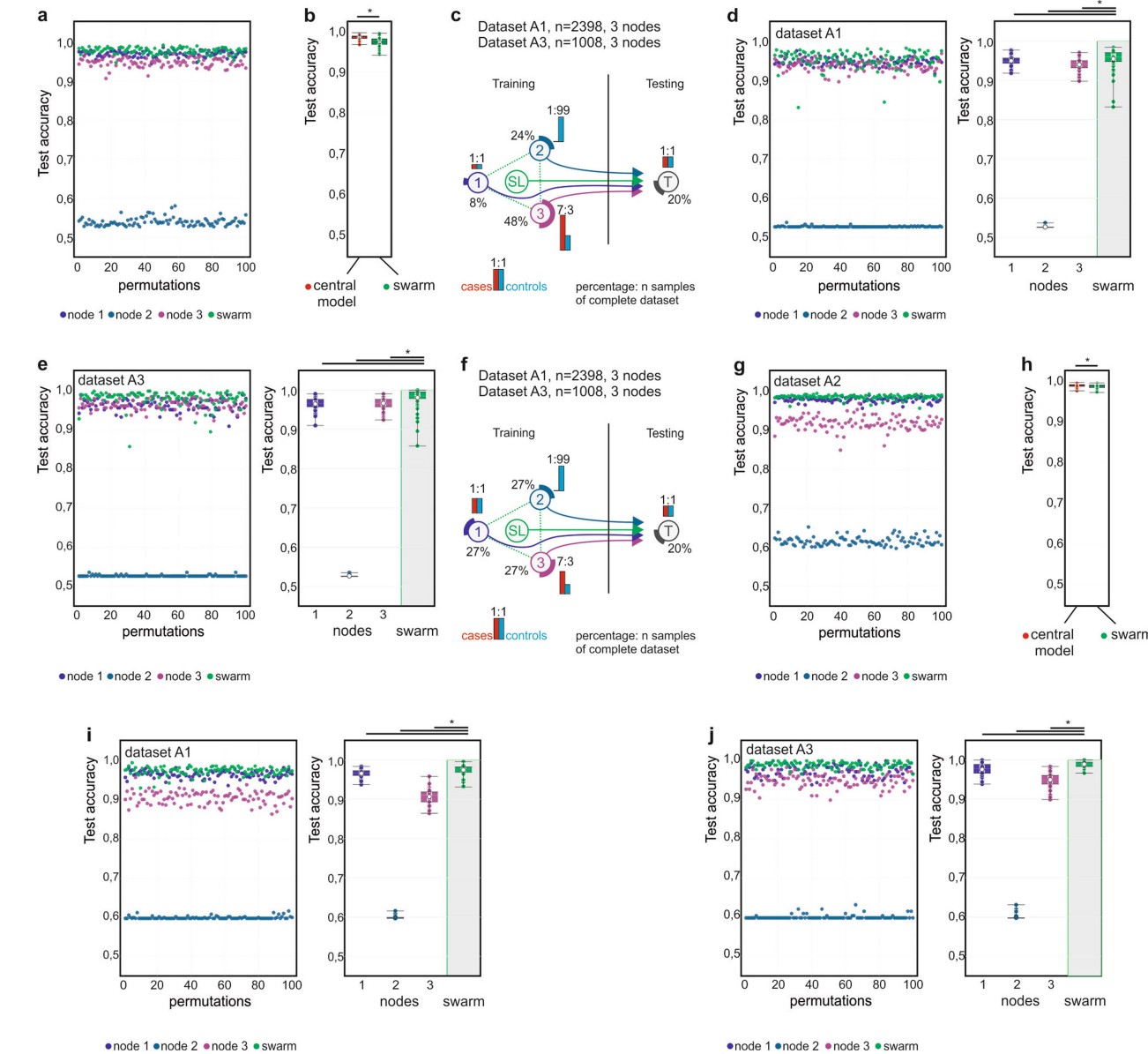

**Extended Data Fig. 2 | Scenario corresponding to Fig. 2b, c in datasets A1 and A3.** Main settings and representation of schema and data visualization as described in Fig. 2a. **a**, Evaluation of test accuracy for 100 permutations of the scenario shown in Fig. 2b. **b**, Evaluation of SL versus central model for the scenario shown in Fig. 2b for 100 permutations. **c**, Scenario with different prevalences of AML and numbers of samples at each training node. The test dataset has an even distribution. **d**, Evaluation of test accuracy for 100 permutations of dataset A1 per node and SL. **e**, Evaluation using dataset A3 for 100 permutations. **f**, Scenario with similar training set sizes per node but decreasing prevalence. The test dataset ratio is 1:1. **g**, Evaluation of test accuracy for 100 permutations of the scenario shown in Fig. 2c. **h**, Evaluation of SL versus central model of the scenario shown in Fig. 2c for 100 permutations.

**i**, Evaluation of test accuracy over 100 permutations for dataset A1 with the scenario shown in **f**. **j**, Evaluation of test accuracy over 100 permutations for dataset A3 with the scenario shown in **f**. **b**, **d**, **e**, **h**–**j**, Box plots show representation of accuracy of 100 permutations performed for the 3 training nodes individually as well as the results obtained by SL. All samples are biological replicates. Centre dot, mean; box limits, 1st and 3rd quartiles; whiskers, minimum and maximum values. Accuracy is defined for the independent fourth node used for testing only. Statistical differences between results derived by SL and all individual nodes including all permutations performed were calculated with one-sided Wilcoxon signed rank test with continuity correction; *$P < 0.05$, exact $P$ values listed in Supplementary Table 5.

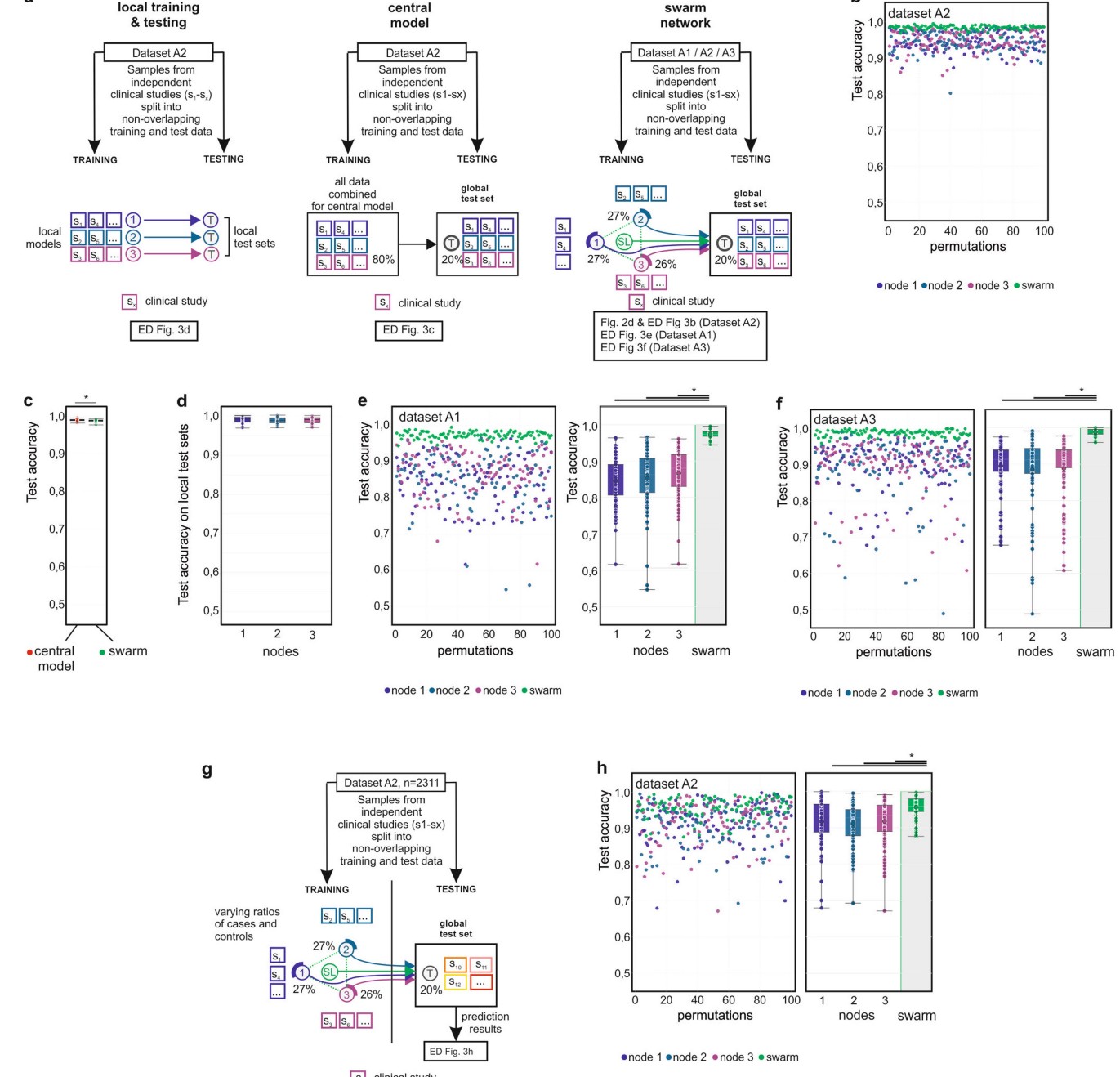

**Extended Data Fig. 3 | Scenario to test for batch effects of siloed studies in datasets A1–A3 and scenario with multiple consortia.** Main settings and representation of schema and data visualization are as in Fig. 2a. **a**, Scenario with training nodes coming from independent clinical studies for local models (left), central model (middle) and the Swarm network (right) and testing on a non-overlapping global test with samples from the same studies. **b**, Evaluation of test accuracy over 100 permutations for dataset A2 with the scenario shown in **a** (right) and Fig. 2d. **c**, Comparison of test accuracy between central model (**a**, middle) and SL (**a**, right). **d**, Comparison of test accuracy on the local test datasets (**a**, left) for 100 permutations. **e**, Evaluation of test accuracy of individual nodes versus SL over 100 permutations for dataset A1 when training nodes have data from independent clinical studies. **f**, Evaluation of test accuracy of individual nodes versus SL over 100 permutations for dataset A3

when training nodes have data from independent clinical studies. **g**, Scenario with three consortia contributing training nodes and a fourth one providing the testing node. **h**, Evaluation of test accuracy for scenario shown in **g** over 100 permutations for dataset A2. **d**–**f**, **h**, Box plots show representation of accuracy of all permutations performed for the 3 training nodes individually as well as the results obtained by SL (**d** only for local models). All samples are biological replicates. Centre dot, mean; box limits, 1st and 3rd quartiles; whiskers, minimum and maximum values. Performance measures are defined for the independent fourth node used for testing only. Statistical differences between results derived by SL and all individual nodes including all permutations performed were calculated with one-sided Wilcoxon signed rank test with continuity correction; *$P < 0.05$, exact $P$ values are listed in Supplementary Table 5.

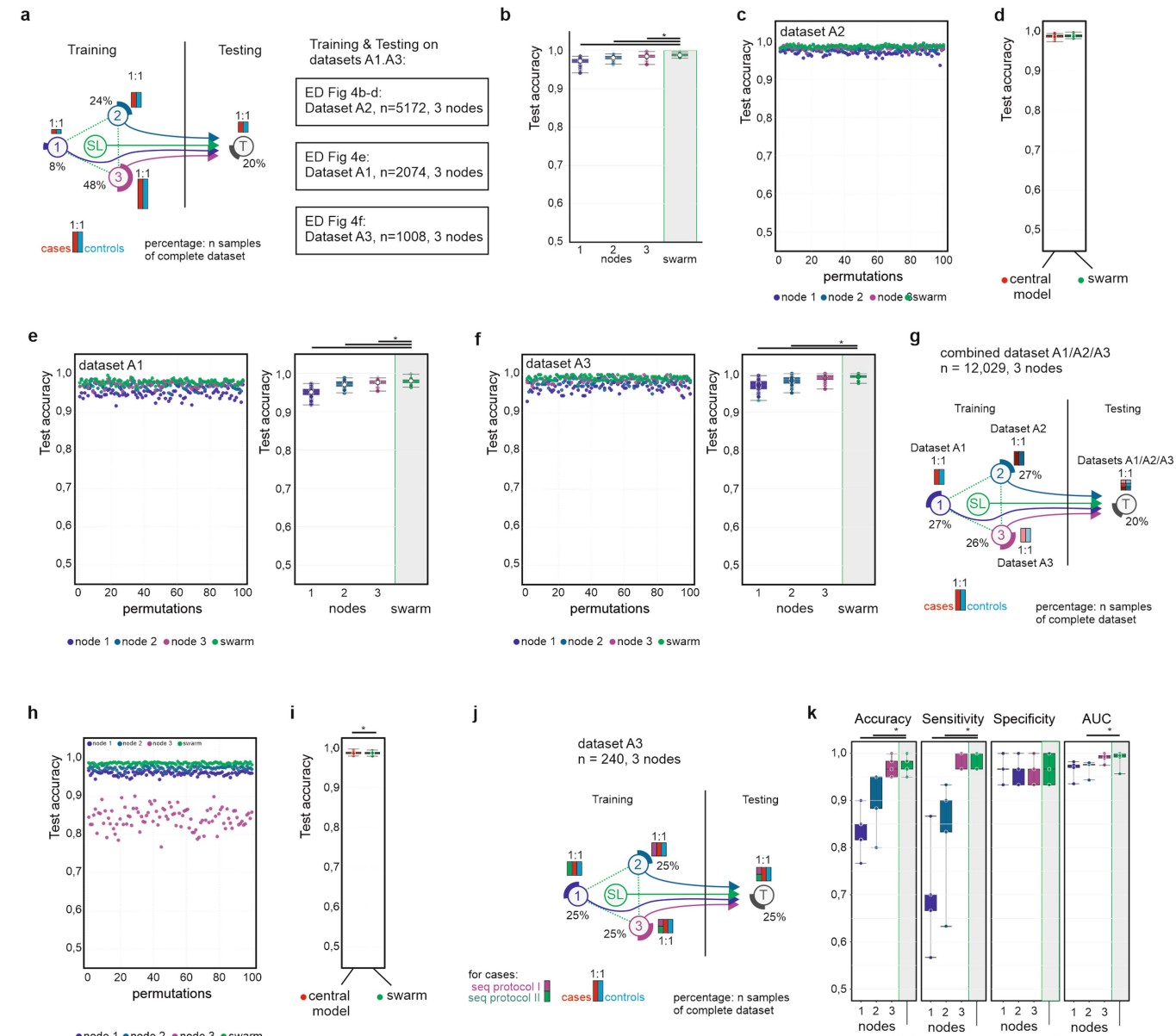

**Extended Data Fig. 4 | Scenario corresponding to Fig. 2e in datasets A1 and A3 and scenario using different data generation methods in each training node.** Main settings and representation of schema and data visualization are as in Fig. 2a. **a**, Scenario with even distribution of cases and controls at each training node and the test node, but different numbers of samples at each node and overall increase in numbers of samples. **b**, **c**, Test accuracy for evaluation of dataset A2 over 100 permutations. **d**, Comparison of central model with SL over 100 permutations. **e**, Test accuracy for evaluation of dataset A1 over 99 permutations. **f**, Test accuracy for evaluation of dataset A3 over 100 permutations. **g**, Scenario where datasets A1, A2, and A3 are assigned to a single training node each. **h**, Evaluation of test accuracy over 100 permutations. **i**, Comparison of the test accuracy of central model and SL over 98

permutations. **j**, Scenario similar to **g** but where the nodes use datasets from different RNA-seq protocols. **k**, Evaluation of results for accuracy, AUC, sensitivity, and specificity over five permutations. **d**–**f**, **i**, **k**, Box plots show predictive performance over all permutations performed for the three training nodes individually as well as the results obtained by SL. All samples are biological replicates. Centre dot, mean; box limits, 1st and 3rd quartiles; whiskers, minimum and maximum values. Performance measures are defined for the independent fourth node used for testing only. Statistical differences between results derived by SL and all individual nodes including all permutations performed were calculated with one-sided Wilcoxon signed rank test with continuity correction; *$P < 0.05$, exact $P$ values listed in Supplementary Table 5.

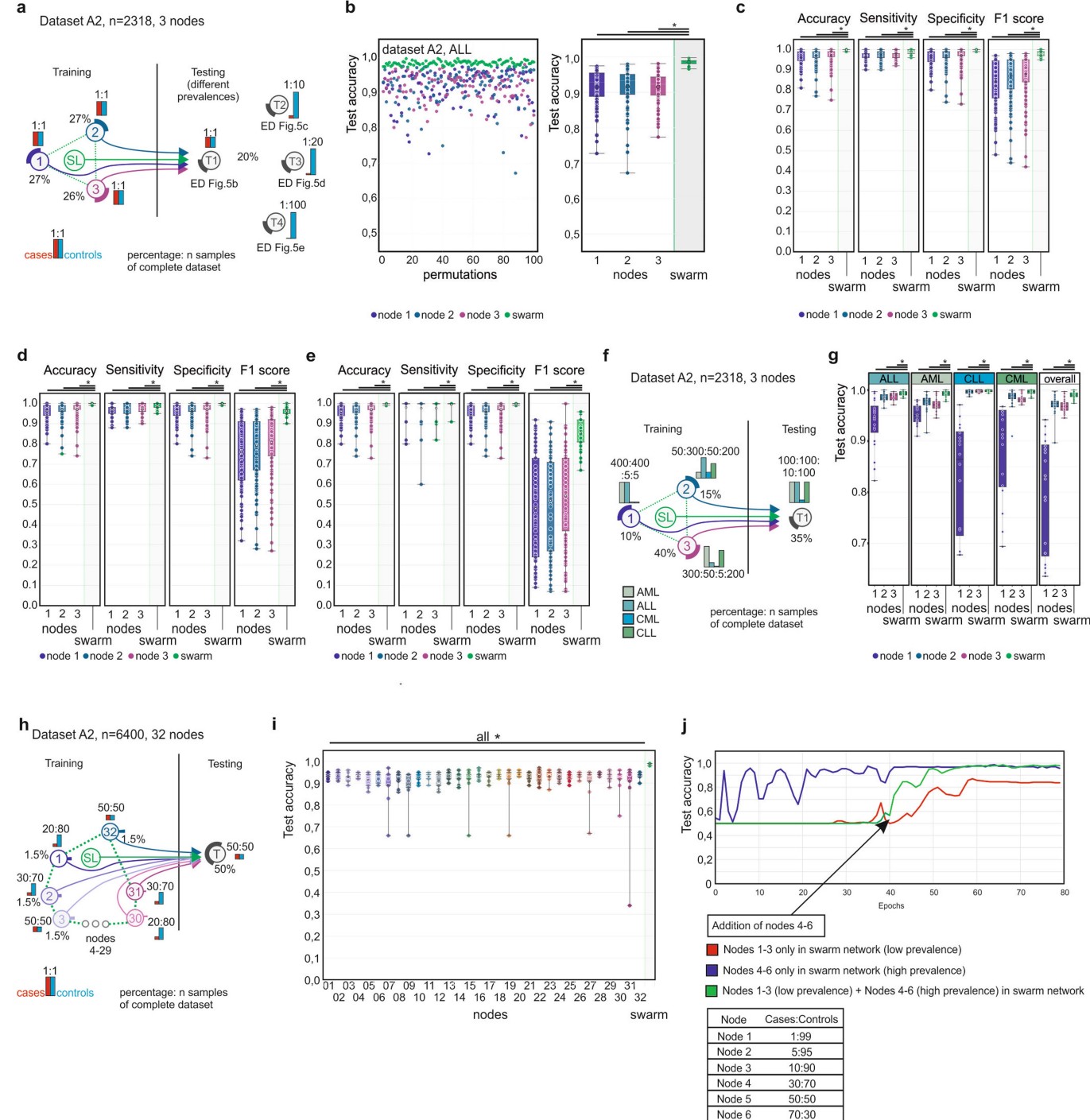

**Extended Data Fig. 5 | Scenario for ALL in dataset 2 and multi-class prediction and expansion of SL.** Main settings are identical to what is described in Fig. 2a. Here cases are samples derived from patients with ALL, while all other samples are controls (including AML). **a**, Scenario for the detection of ALL in dataset A2. The training sets are evenly distributed among the nodes with varying prevalence at the testing node. Data from independent clinical studies are samples to each node, as described for AML in Fig. 2d. **b**, Evaluation of scenario in **a** for test accuracy over 100 permutations with a prevalence ratio of 1:1. **c**, Evaluation using a test dataset with prevalence ratio of 10:100 over 100 permutations. **d**, Evaluation using a test dataset with prevalence ratio of 5:100 over 100 permutations. **e**, Evaluation using a test dataset with prevalence ratio of 1:100. **f**, Scenario for multi-class prediction of different types of leukaemia in dataset A2. Each node has a different

prevalence. **g**, Test accuracy for the different types of leukaemia over 20 permutations. **h**, Scenario that simulates 32 small Swarm nodes. **i**, Evaluation of test accuracy for the 32 nodes and the Swarm over 10 permutations. **j**, Development of accuracy over training epochs with addition of new nodes. **b**–**e**, **g**, **i**, Box plots show performance of all permutations performed for the training nodes individually as well as the results obtained by SL. All samples are biological replicates. Centre dot, mean; box limits, 1st and 3rd quartiles; whiskers, minimum and maximum values. Performance measures are defined for the independent test node used for testing only. Statistical differences between results derived by SL and all individual nodes including all permutations performed were calculated with one-sided Wilcoxon signed rank test with continuity correction; *$P$ < 0.05, exact $P$ values listed in Supplementary Table 5.

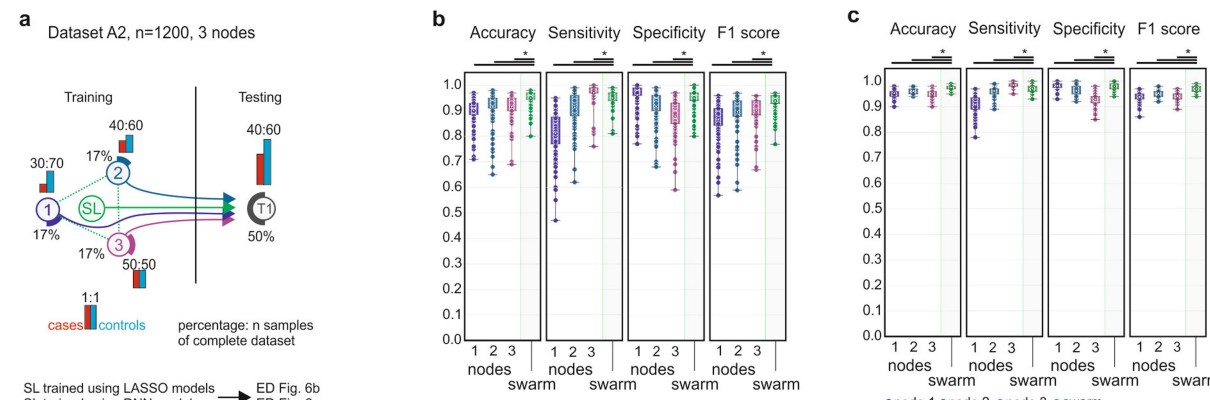

**Extended Data Fig. 6 | Comparison of LASSO and neural networks.**
**a**, Scenario for training different models in the Swarm. **b**, Evaluation of a LASSO model for accuracy, sensitivity, specificity and F1 score over 100 permutations. **c**, Evaluation of a Neural Network model for accuracy, sensitivity, specificity and F1 score over 100 permutations. **b**, **c**, Box plots show performance of all permutations performed for the training nodes individually as well as the results obtained by SL. All samples are biological replicates. Centre dot, mean; box limits, 1st and 3rd quartiles; whiskers, minimum and maximum values. Performance measures are defined for the independent fourth node used for testing only. Statistical differences between results derived by SL and all individual nodes including all permutations performed were calculated with one-sided Wilcoxon signed rank test with continuity correction; *$P < 0.05$, exact $P$ values listed in Supplementary Table 5.

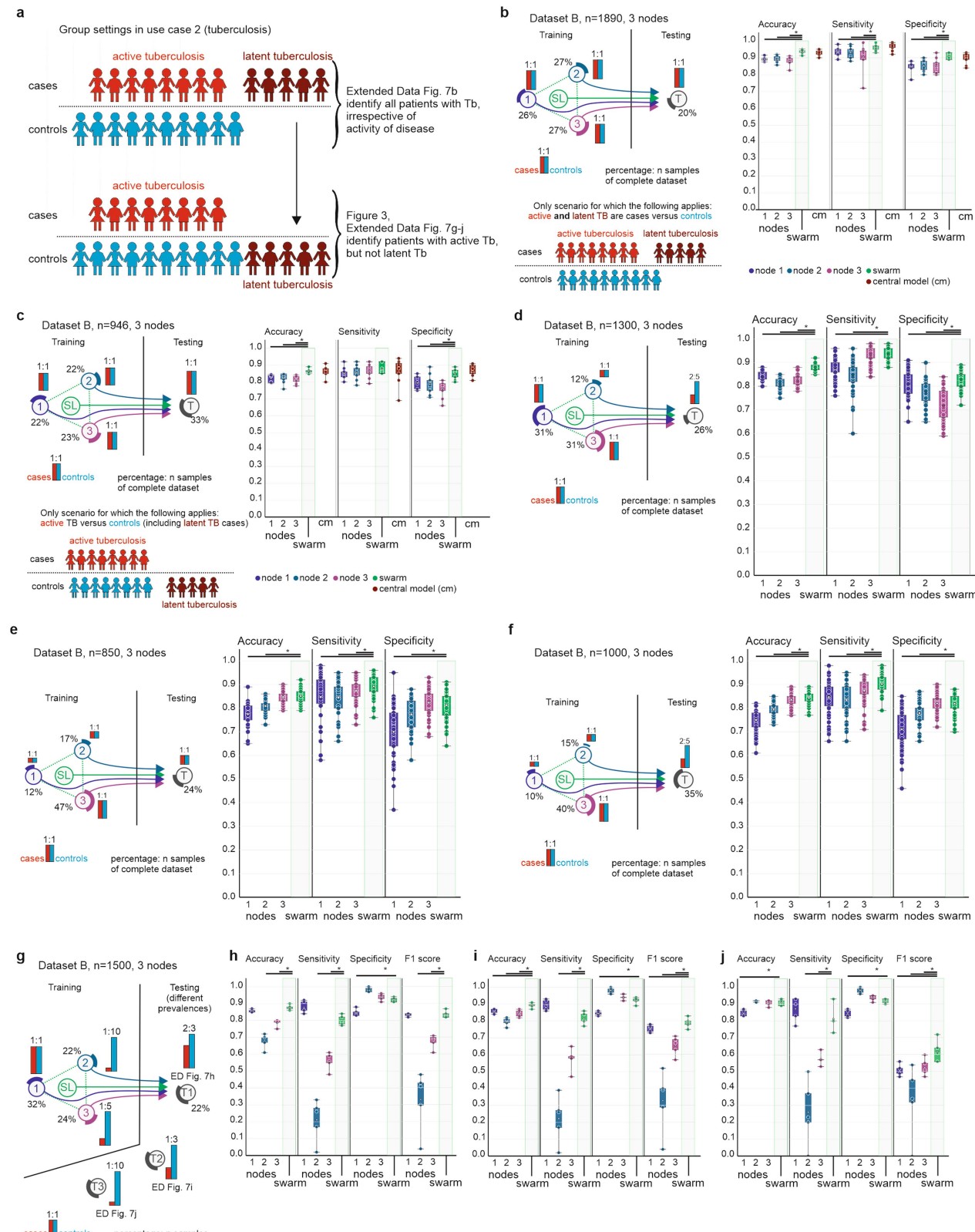

**Extended Data Fig. 7** | See next page for caption.

**Extended Data Fig. 7 | Scenarios for detecting all TB versus controls and for detecting active TB with low prevalence at training nodes.** Main settings are as in Fig. 2a. **a**, Different group settings used with assignment of latent TB to control or case. **b**, Left, evaluation of a scenario where active and latent TB are cases. The data are evenly distributed among the training nodes. Right, test accuracy, sensitivity and specificity for nodes, Swarm and a central model over 10 permutations. **c**, Left, scenario similar to **b** but with latent TB as control. Right, test accuracy, sensitivity and specificity for nodes, Swarm and a central model over 10 permutations. **d**, Left, scenario with reduced prevalence at the test node. Right, test accuracy, sensitivity and specificity for nodes and Swarm over 10 permutations. **e**, Scenario with even distribution of cases and controls at each training node, where node 1 has a very small training set. The test dataset is evenly distributed. Right, test accuracy, sensitivity and specificity over 50 permutations. **f**, Left, scenario similar to **e** but with uneven distribution in the test node. Right, test accuracy, sensitivity and specificity over 50

permutations. **g**, Scenario with each training node having a different prevalence. Three prevalence scenarios were used in the test dataset. **h**, Accuracy, sensitivity, specificity and F1 score over five permutations for testing set T1 as shown in **g**. **i**, As in **h** but with prevalence changed to 1:3 cases:controls in the training set. **j**, As in **h** but with prevalence changed to 1:10 cases:controls in the training set. **b**–**f**, **h**–**j**, Box plots show performance of all permutations performed for the training nodes individually as well as the results obtained by SL. All samples are biological replicates. Centre dot, mean; box limits, 1st and 3rd quartiles; whiskers, minimum and maximum values. Performance measures are defined for the independent fourth node used for testing only. Statistical differences between results derived by SL and all individual nodes including all permutations performed were calculated with one-sided Wilcoxon signed rank test with continuity correction; *$P < 0.05$, exact $P$ values listed in Supplementary Table 5.

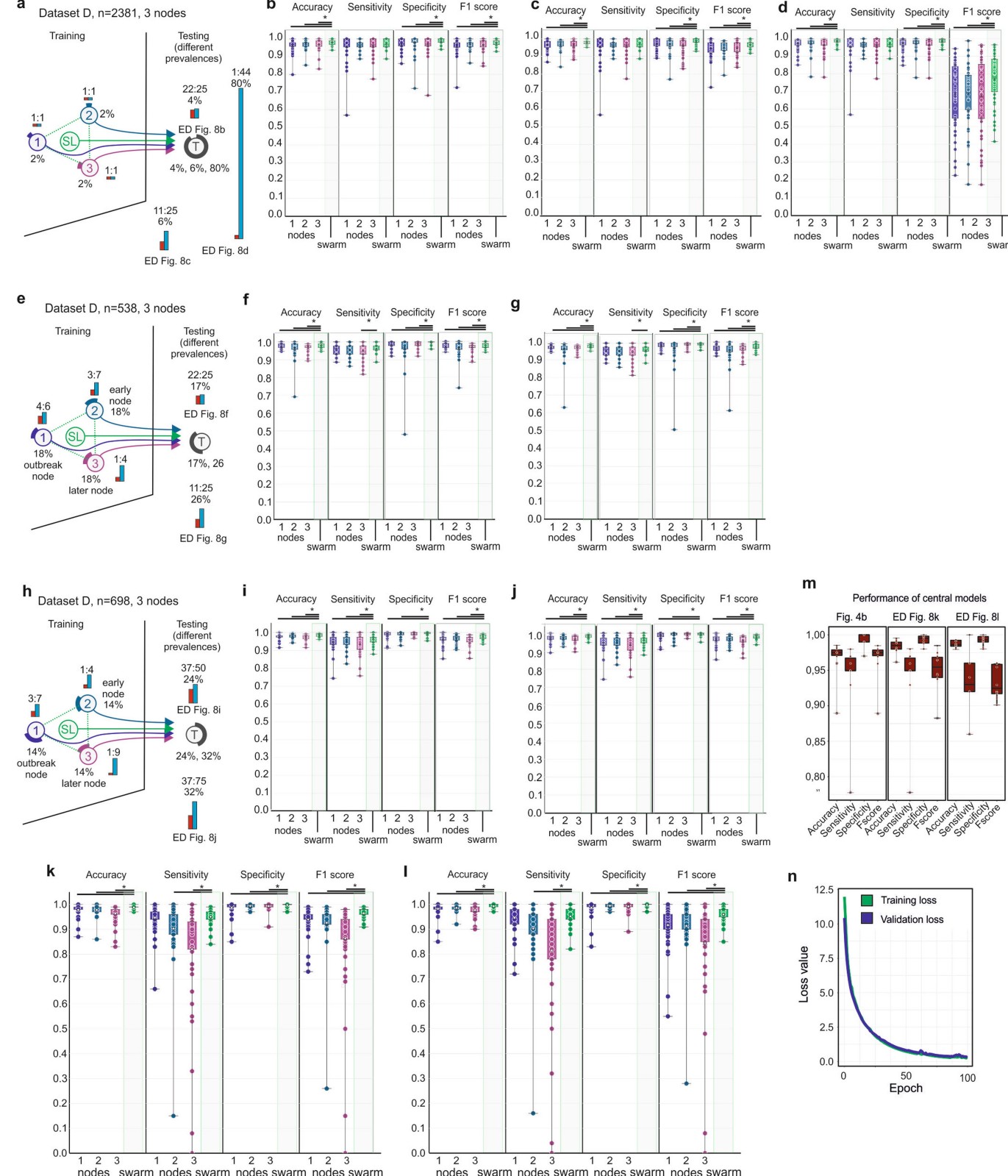

**Extended Data Fig. 8** | See next page for caption.

**Extended Data Fig. 8 | Baseline scenario for detecting patients with COVID-19 and scenario with reduced prevalence at training nodes.** Main settings are as in Fig. 2a. **a**, Scenario for detecting COVID-19 with even training set distribution among nodes 1–3. Three testing sets with different prevalences were simulated. **b**, Accuracy, sensitivity, specificity and F1 score over 50 permutations for scenario in **a** with a 22:25 case:control ratio. **c**, As in **b** for an 11:25 ratio. **d**, As in **b** for a 1:44 ratio. **e**, Scenario with the same sample size at each training node, but prevalence decreasing from node 1 to node 3. There are two test datasets (**f**, **g**). **f**, Evaluation of scenario in **e** with 22:25 ratio at the test node over 50 permutations. **g**, Evaluation of scenario in **e** with reduced prevalence over 50 permutations. **h**, Scenario similar to **e** but with a steeper decrease in prevalence between nodes 1 and 3. **i**, Evaluation of scenario in **h** with a ratio of 37:50 at the test node over 50 permutations. **j**, Evaluation of scenario in **h** with a reduced prevalence compared to **i** over 50 permutations. **k**, Scenario as in Fig. 4a using a 1:5 ratio for cases and controls in the test dataset evaluated over 50 permutations. **l**, Scenario as in Fig. 4a using a 1:10 ratio in the test dataset to simulate detection in regions with new infections, evaluated over 50 permutations. **m**, Performance of central models for **k**, **l** and Fig. 4b. **n**, Loss function of training and validation loss over 100 training epochs. **b**–**d**, **f**, **g**, **i**–**m**, Box plots show performance of all permutations performed for the training nodes individually as well as the results obtained by SL. All samples are biological replicates. Centre dot, mean; box limits, 1st and 3rd quartiles; whiskers, minimum and maximum values. Performance measures are defined for the independent fourth node used for testing only. Statistical differences between results derived by SL and all individual nodes including all permutations performed were calculated with one-sided Wilcoxon signed rank test with continuity correction; *$P < 0.05$, exact $P$ values listed in Supplementary Table 5.

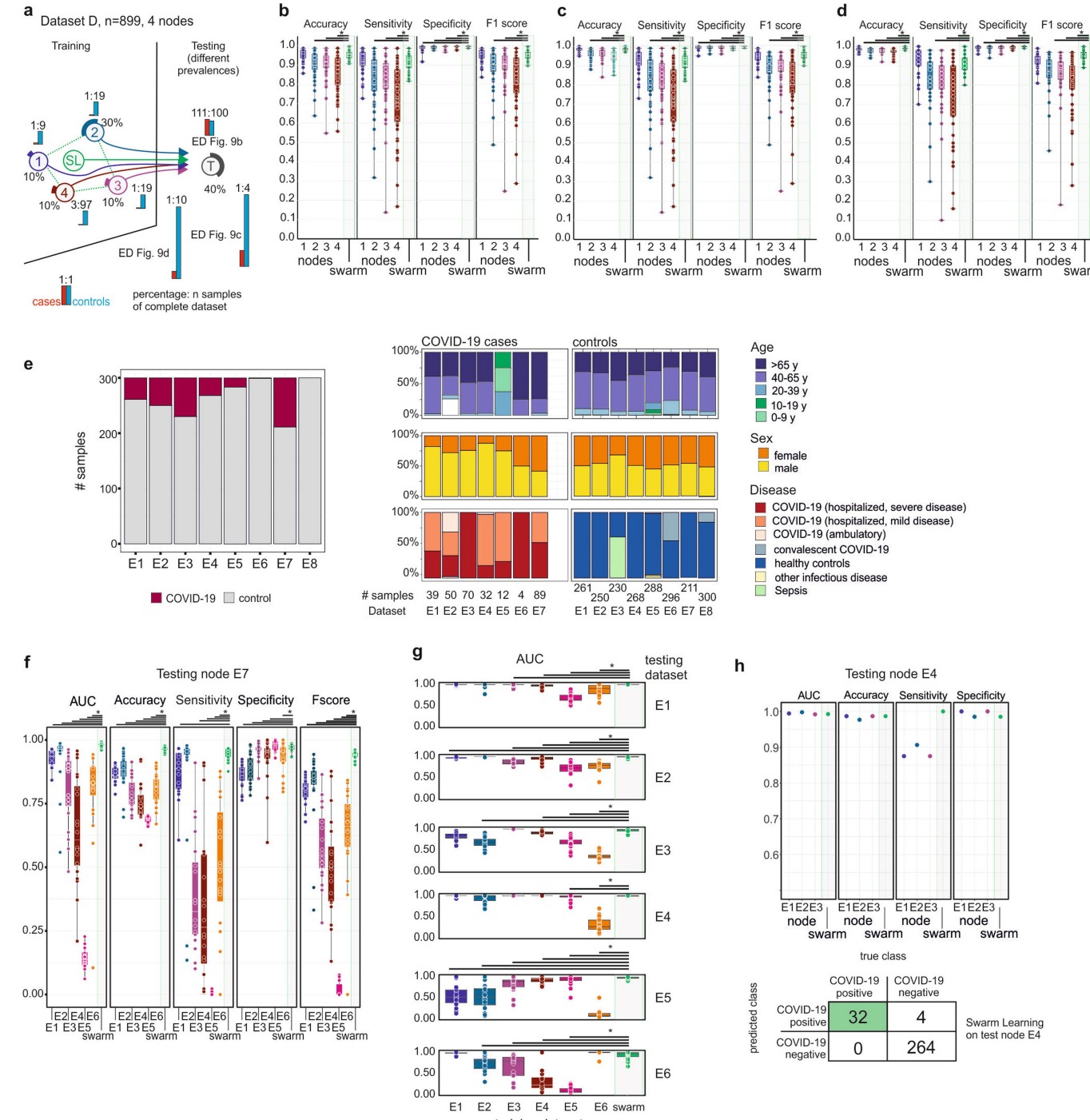

**Extended Data Fig. 9 | Scenario with reduced prevalence in training and test datasets and multi-centre scenario at a four-node setting.** Main settings as in Fig. 2a. **a**, Scenario with prevalences from 10% at node 1 to 3% at node 4. There are three test datasets (**b**–**d**) with decreasing prevalence and increasing total sample size. **b**, Evaluation of scenario in **a** with 111:100 ratio over 50 permutations. **c**, Evaluation of scenario in **a** with 1:4 ratio and increased sample number of the test dataset over 50 permutations. **d**, Evaluation of scenario in **a** with 1:10 prevalence and increased sample number of the test dataset over 50 permutations. **e**, Dataset properties for the participating cities E1–E8, indicating case:control ratio and demographic properties. **f**, AUC, accuracy, sensitivity, specificity and F1 score over 20 permutations for scenario that uses E1–E6 as training nodes and E7 as external test node. **g**, Evaluation of a multi-city scenario where a medical centre (in each row)

serves as a test node. The AUC for each training node and the SL is shown for 20 permutations. **h**, Multi-city scenario. Only three nodes (E1–E3) are used for training and the external test node E4 uses data from a different sequencing facility. AUC, accuracy, sensitivity and specificity as well as the confusion matrix for one prediction. **b**–**d**, **f**, **g**, Box plots show performance of all permutations performed for the training nodes individually as well as the results obtained by SL. All samples are biological replicates. Centre dot, mean; box limits, 1st and 3rd quartiles; whiskers, minimum and maximum values. Performance measures are defined for the independent fourth node used for testing only. Statistical differences between results derived by SL and all individual nodes including all permutations performed were calculated with one-sided Wilcoxon signed rank test with continuity correction; *$P < 0.05$, exact $P$ values listed in Supplementary Table 5.

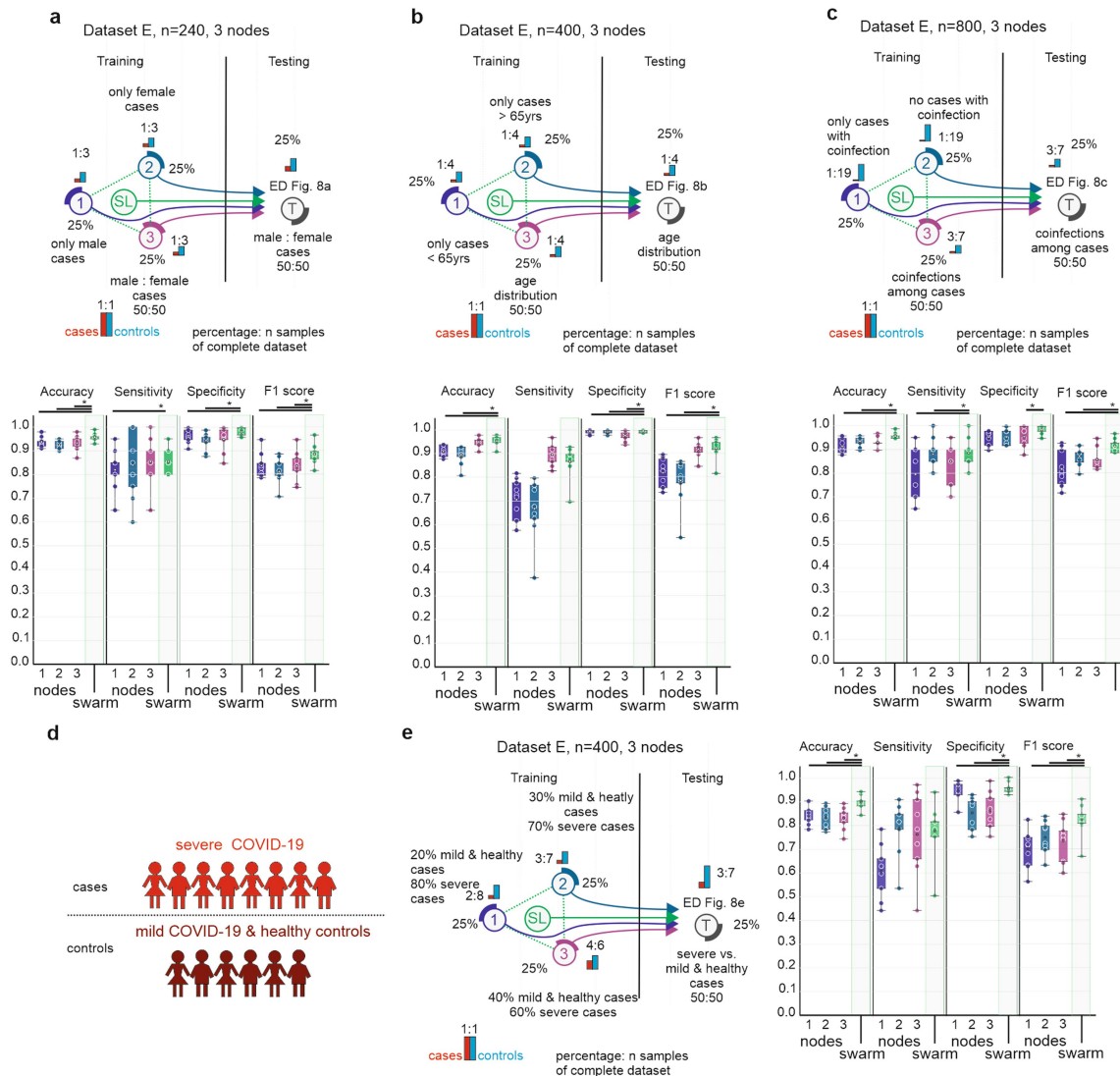

**Extended Data Fig. 10 | Scenarios for testing different factors and scenario for testing disease severity.** Main settings as in Fig. 2a. **a**, Top, scenario to test influence of sex with three training nodes. Training node 1 has only male cases, node 2 has only female cases. Training node 3 and the test node have a 50%/50% split. Bottom, accuracy, sensitivity, specificity and F1 score for each training node and the Swarm in 10 permutations. **b**, Top, scenario to test influence of age with three training nodes. Training node 1 only has cases younger than 65 years, node 2 only has cases older than 65 years. Training node 3 and the test node have a 50%/50% split of cases above and below 65 years. Bottom, accuracy, sensitivity, specificity and F1 score for each training node and the Swarm in 10 permutations. **c**, Top, scenario to test influence of co-infections with three training nodes. Training node 1 has only cases with co-infections, node 2 has no cases with co-infections. Training node 3 and the test node have a 50%/50% split. Bottom, accuracy, sensitivity, specificity and F1 score for each training node and the Swarm in 10 permutations. **d**, Prediction setting. Severe cases of

COVID-19 are cases, mild cases of COVID-19 and healthy donors are controls. **e**, Left, scenario to test influence of disease severity with three training nodes. Training node 1 has 20% mild or healthy and 80% severe cases, node 3 has 40% mild or healthy and 60% severe cases. Training node 2 and the test node have 30% mild or healthy and 70% severe cases. Right, accuracy, sensitivity, specificity and F1 score for each training node and the Swarm for 10 permutations. **a**–**c**, **e**, Box plots show performance all permutations performed for the training nodes individually as well as the results obtained by SL. All samples are biological replicates. Centre dot, mean; box limits, 1st and 3rd quartiles; whiskers, minimum and maximum values. Performance measures are defined for the independent fourth node used for testing only. Statistical differences between results derived by SL and all individual nodes including all permutations performed were calculated with one-sided Wilcoxon signed rank test with continuity correction; *$P < 0.05$, exact $P$ values listed in Supplementary Table 5.

# nature research

# Reporting Summary

Nature Research wishes to improve the reproducibility of the work that we publish. This form provides structure for consistency and transparency in reporting. For further information on Nature Research policies, see our Editorial Policies and the Editorial Policy Checklist.

## Statistics

For all statistical analyses, confirm that the following items are present in the figure legend, table legend, main text, or Methods section.

| n/a | Confirmed | |
|---|---|---|
| ☐ | ☒ | The exact sample size (*n*) for each experimental group/condition, given as a discrete number and unit of measurement |
| ☐ | ☒ | A statement on whether measurements were taken from distinct samples or whether the same sample was measured repeatedly |
| ☐ | ☒ | The statistical test(s) used AND whether they are one- or two-sided<br>*Only common tests should be described solely by name; describe more complex techniques in the Methods section.* |
| ☐ | ☒ | A description of all covariates tested |
| ☒ | ☐ | A description of any assumptions or corrections, such as tests of normality and adjustment for multiple comparisons |
| ☐ | ☒ | A full description of the statistical parameters including central tendency (e.g. means) or other basic estimates (e.g. regression coefficient) AND variation (e.g. standard deviation) or associated estimates of uncertainty (e.g. confidence intervals) |
| ☐ | ☒ | For null hypothesis testing, the test statistic (e.g. *F*, *t*, *r*) with confidence intervals, effect sizes, degrees of freedom and *P* value noted<br>*Give P values as exact values whenever suitable.* |
| ☒ | ☐ | For Bayesian analysis, information on the choice of priors and Markov chain Monte Carlo settings |
| ☒ | ☐ | For hierarchical and complex designs, identification of the appropriate level for tests and full reporting of outcomes |
| ☒ | ☐ | Estimates of effect sizes (e.g. Cohen's *d*, Pearson's *r*), indicating how they were calculated |

*Our web collection on statistics for biologists contains articles on many of the points above.*

## Software and code

Policy information about availability of computer code

Data collection
Dataset A: All raw data files were downloaded from GEO and the RNA-seq data was preprocessed using the kallisto aligner v.0.43.1 against the human reference genome gencode v27 (GRCh38.p10). For normalization, we considered all platforms independently, meaning that normalization was performed separately for the samples in Dataset A1, A2 and A3, respectively. Microarray data (Datasets A1 and A2) was normalized using the robust multichip average (RMA) expression measures, as implemented in the R package affy (version 1.60.0). RNA-seq data (Dataset A3) was normalized with the R package DESeq2 (version 1.22.2) using standard parameters. In order to keep the datasets comparable, data was filtered for genes annotated in all three datasets, which resulted in 12,708 genes. No filtering of low-expressed genes was performed. All scripts used in this study for pre-processing are provided as a docker container on Docker Hub (version 0.1, https://hub.docker.com/r/schultzlab /aml_classifier) and GitHub (https://github.com/schultzlab/swarm_learning).
Dataset B,D,E: All raw data file were downloaded from GEO or collected at the partner hospitals and aligned to the human reference genome gencode v33 (GRCh38.p13) and quantified transcript counts using STAR v 2.7.3a. For all samples in Datasets B and D,E, raw counts were imported using the R package DESeq2 (version 1.22.2, DESeqDataSetFromMatrix function) and size factors for normalization were calculated using the DESeq function using standard parameters.
Dataset C: The NIH Chest X-Ray dataset was downloaded from https://www.kaggle.com/nih-chest-xrays/data. In order to preprocess the data, we used Python (version 3.6.9) and Keras (version 2.3.1) real-time data augmentation and generation APIs (keras.preprocessing.image.ImageDataGenerator and flow_from_dataframe). The following pre-processing arguments were used: height or width shift range (~ 5%), random rotation range (~ 5 degree), random zoom range (~ 0.15), sample-wise center and standard normalization. Additionally, all images are resized to (128 * 128) from their original size of (1024 * 1024).

Data analysis
All models for the experiments have been implemented using Python (version 3.6.9), Keras (version 2.3.1), Tensorflow (2.2.0-rc2) and scikit-learn (version 0.23.1). The LASSO algorithm has been implemented using Keras (version 2.3.1). All code is available on GitHub (https://github.com/schultzlab/swarm_learning).
Measurements of sensitivity, specificity, accuracy and F1 score of each permutation run was read into a table in Excel (Microsoft Excel for Microsoft 365 MSO: Version: 2008 13127.21348 (16.0.13127_21336 64-bit)) using Power Query (Microsoft Excel for Microsoft 365 MSO: Version: 2008 13127.21348 (16.0.13127_21336 64-bit)) and used for visualization for the different scenarios in Power BI [Version:

2.81.5831.821 64-bit (Mai 2020)] with Box and Whisker chart by MAQ Software (https://appsource.microsoft.com/en-us/product/power-bi-visuals/WA104381351, version 3.2.1).
AUC, positive predictive value, all confidence intervals and statistical tests were calculated using R (version 3.5.2) and the R packages MKmisc (version 1.6) and ROCR (version 1.0.7).

For manuscripts utilizing custom algorithms or software that are central to the research but not yet described in published literature, software must be made available to editors and reviewers. We strongly encourage code deposition in a community repository (e.g. GitHub). See the Nature Research guidelines for submitting code & software for further information.

## Data

Policy information about availability of data

All manuscripts must include a data availability statement. This statement should provide the following information, where applicable:
- Accession codes, unique identifiers, or web links for publicly available datasets
- A list of figures that have associated raw data
- A description of any restrictions on data availability

Processed data can be accessed via the SuperSeries GSE122517 or via the individual SubSeries GSE122505 (dataset A1), GSE122511 (dataset A2) and GSE122515 (dataset A3). Dataset B consists of the following series which can be accessed at GEO: GSE101705, GSE107104, GSE112087, GSE128078, GSE66573, GSE79362, GSE84076, and GSE89403. Furthermore, it contains the Rhineland study. This dataset is not publicly available because of data protection regulations. Access to data can be provided to scientists in accordance with the Rhineland Study's Data Use and Access Policy. Requests for further information or to access the Rhineland Study's dataset should be directed to RS-DUAC@dzne.de. Dataset D and E contain dataset B and additional samples for COVID-19. These datasets are made available at the European Genome-Phenome Archive (EGA) under accession number EGAS00001004502 , which is hosted by the EBI and the CRG. The healthy RNA-seq data included from Saarbrücken is available from PPMI through the LONI data archive, https://www.ppmi-info.org/data. The NIH CC Chest X-Ray (Dataset C) can be downloaded from https://www.kaggle.com/nih-chest-xrays/data. Normalized log transformed expression matrices of datasets A1, A2, A3, B, D and E as used for the predictions are made available via FASTGenomics at https://beta.fastgenomics.org/p/swarm-learning.

# Field-specific reporting

Please select the one below that is the best fit for your research. If you are not sure, read the appropriate sections before making your selection.

☒ Life sciences    ☐ Behavioural & social sciences    ☐ Ecological, evolutionary & environmental sciences

For a reference copy of the document with all sections, see nature.com/documents/nr-reporting-summary-flat.pdf

# Life sciences study design

All studies must disclose on these points even when the disclosure is negative.

| | |
|---|---|
| Sample size | For the 12029 samples from data set A (AML), we followed work of Warnat-Herresthal et al, 2020, (doi: 10.1016/j.isci.2019.100780). Dataset B (Tb, 1999 samples) is a collection of all available PAX-based high-quality Tb datasets and controls on GEO. For COVID-19 in dataset D, the collection of 134 samples and 9 controls was driven by availability of consenting patients. For dataset E, the collection of 2400 samples was driven by availability of consenting patients. Dataset C has been compiled and published by the NIH CC and contains 112120 X-ray images. It is one of the largest community data sets and has been used in many studies. |
| Data exclusions | We used a minimum of five million aligned reads per samples to exclude low-quality samples from the Covid samples. This number is recommended as a minimum for bulk RNA sequencing, as e.g. stated by Illumina (https://support.illumina.com/bulletins/2017/04/considerations-for-rna-seq-read-length-and-coverage-.html) |
| Replication | The swarm learning approach has been successfully replicated in five data sets (A,B,C,D,E) with multiple permutations. |
| Randomization | The allocation into experimental group was determined by disease/condition and no other covariates were used. An additional experiment tested the impact of age, sex and COVID-19 diseases severity. |
| Blinding | Blinding was not applicable, since we collected pre-existing data sets. Additionally to guarantee independent sampling, we performed random permutations of training and test data sets. |

# Reporting for specific materials, systems and methods

We require information from authors about some types of materials, experimental systems and methods used in many studies. Here, indicate whether each material, system or method listed is relevant to your study. If you are not sure if a list item applies to your research, read the appropriate section before selecting a response.

## Materials & experimental systems

| n/a | Involved in the study |
|-----|----------------------|
| ☒ | Antibodies |
| ☒ | Eukaryotic cell lines |
| ☒ | Palaeontology and archaeology |
| ☒ | Animals and other organisms |
| ☐ | Human research participants ☒ |
| ☒ | Clinical data |
| ☒ | Dual use research of concern |

## Methods

| n/a | Involved in the study |
|-----|----------------------|
| ☒ | ChIP-seq |
| ☒ | Flow cytometry |
| ☒ | MRI-based neuroimaging |

# Human research participants

Policy information about studies involving human research participants

**Population characteristics**

The Rhineland Study participants stem from an ongoing community-based cohort study in which all inhabitants of two geographically defined areas in the city of Bonn, Germany aged 30–100 years are being invited to participate. Persons living in these areas are predominantly German with Caucasian ethnicity. Participation in the study is possible by invitation only. The only exclusion criterion is insufficient German language skills to give informed consent.
The COVID-19 samples are described in Supplementary Table 6.

**Recruitment**

The Rhineland Study is an ongoing community-based cohort study in which all inhabitants of two geographically defined areas in the city of Bonn, Germany, aged 30 years and above are being invited to participate. Persons living in these areas are predominantly German from Caucasian descent. Participation in the study is possible by invitation only. The only exclusion criterion is insufficient command of the German language to give informed consent. Therefore, given that participation in the Rhineland Study does not depend on any health-related outcome (e.g. the presence or absence of any particular lifestyle, disease or therapy), the potential risk of any selection bias impacting our results is, in all likelihood, very low.
COVID-19 samples were collected based on availability. For all COVID-19 patients, the study was carried out in accordance with the applicable rules concerning the review of research ethics committees and informed consent. All patients or legal representatives were informed about the study details and could decline to participate. COVID-19 was diagnosed by a positive SARS-CoV-2 RT-PCR test in nasopharyngeal or throat swabs and/or by typical chest CT-scan finding.

**Ethics oversight**

Approval to undertake the Rhineland Study was obtained from the ethics committee of the University of Bonn, Medical Faculty. Collection of Covid19 samples was overseen by the research ethics committees at Radboud University Medical Centre in Nijmegen, the Netherlands (local ethics committee CMO Arnhem-Nijmegen, registration no. 2016-2923), and the Sotiria Athens General Hospital (Ethics Committee of Sotiria Athens General Hospital, IRB 23/12.08.2019) or the ATTIKON University General Hospital ((Ethics Committee of ATTIKON University General Hospital, IRB 26.02.2019) in Athens, Greece as well as the respective committees at the other sites: Kiel, Germany (COVIDOM, Ethics Committee of the University of Kiel, IRB D466/20), Saarbrücken, Germany (CORSAAR, Ethics Committee Medical Association of the Saarland, IRB 62/20, IRB 20200597), Munich, Germany (Ethics Committee of the LMU Munich, IRB 286/2020B01), Tübingen Germany (DeCOI Host Genomes, Ethics Committee of the Medical Faculty of the University of Tübingen, IRB 286/2020B01), Aachen, Germany (COVAS, Ethics Committee of the Medical Faculty of the Technical University Aachen, IRB 20-085), Cologne, Germany (Ethics Committee of the University of Cologne, IRB 20-1187_1) and Bonn, Germany (Ethics Committee of the Medical Faculty of the University of Bonn, IRB 073/19, 134/20). Dataset C is IRB approved (personal communication by Dr. Summers  Senior Investigator,  Clinical Image Processing Service,  NIH CC).

Note that full information on the approval of the study protocol must also be provided in the manuscript.

