## [Peer Review File · Nature]

Manuscript Title: Swarm Learning for decentralized and confidential clinical machine learning

Editorial Notes:

Figure labels in the published article may not match figure labels in this file, as the figures have been revised and reorganized during the peer review process.

Reviewer Comments & Author Rebuttals

Reviewer Reports on the Initial Version:

Referee #1 (Remarks to the Author):

Summary of the key results

The paper introduces Swarm Learning (SL), a framework that allows for distributed machine learning in a safe and secure environment. The authors claim that the approach guarantees privacy through de-centralization and blockchain mechanisms.

The authors show that on datasets with high levels of signal, their approach systematically gets higher aggregate accuracy than on any individual node (e.g. edge computation). The authors also show that as they try to make the simulation more and more realistic the performance deteriorates, and yet SL still outperforms each individual node.

Originality and significance: if not novel, please include reference

The approach is novel and would be of high significance, unfortunately the authors stop short from evaluating their model in realistic scenarios reducing the significance of the contribution.

Also, to be precise, SL does not provide privacy per se, as is claimed by the authors. SL is still a novel approach in this reviewer's eyes. It's just the portrayal is somewhat inaccurate. As presented, SL appears to be federated learning with extra privacy guarantees due to blockchain and computational shifting between leaders. Federated learning does not provide privacy, it provides confidentiality (the data is not shared with the central party) but private information is still contained in the gradients. To obtain privacy, one needs to combine federated learning with a differentially private algorithm like DP-SGD. The differential privacy literature shows that access to gradients/parameters of the model reduces data privacy. Adding noise to model gradients, which this paper does not use, is one method of adding additional privacy guarantees (see Abadi et al 2016 and Dwork et al 2006).

Overall, the approach is a novel and exciting direction and if the authors are able to demonstrate high performance in appropriately realistic scenarios, it will be of broad interest.

Data & methodology: validity of approach, quality of data, quality of presentation

Tested scenarios are not realistic in many ways. Results, i.e. performance comparisons, are also not done appropriately. See the list of reasons supporting these statements below.

a) The performance of the models is often in the high 90s. This indicates a very high signal to noise ratio, something that is very rare in realistic healthcare scenarios. The provided examples should vary both in type (not all blood transcriptomes) and signal-to-noise ratio.

b) The test set consists of IID draws from each node, and then each local (or edge) model has its accuracy recorded on its respective test patients. However, the SL model makes predictions on all the nodes test set patients. It would be more appropriate to compare the pairwise accuracy between Node j 's test set patients on the local model and the SL model, i.e. reported performance should be done on the same set of patients. There will then be as many comparisons as their nodes and appropriate multiple corrections will have to be made.

c) A more realistic setting would be to have test set patients which are from a different dataset/source, not drawn IID from existing patients. This would more properly show the ability of the SL model to generalize. It also would be nice to see the comparison to the best achievable performance, i.e. by the model that is trained on all the data.

d) The authors assign the number of nodes independent of the data sources, whereas the nodes should always be composed of the same source - again, to set the realistic scenario.

e) It would be useful to see how the algorithm scales with many nodes (the most shown in the paper was six), but in many multisite studies there could be hundreds of nodes. How many nodes can you add with noisy data before SL breaks down?

f) Authors do not provide illustration/means for onboarding a new node with new data and using a trained SL to make predictions. Perhaps they suggest to retrain/re-ensemble SL every time? If that is the case, how do they guarantee safety and efficacy of the approach over time (something that health authorities normally require to be used in practice/on patients).

g) The authors show deterioration of performance for unbalanced datasets. Reporting balanced accuracy is misleading at all times - one can train on balanced datasets, but should always strive to test using realistic case/control ratio. ALL is not very common at the hospital, presumably that is the scenario where such a tool would be used, so a realistic simulated test should reflect that case/control ratio.

h) It is not clear how the parameter for ensembling various models is picked - this is a foundational part of the performance of the SL and should be brought up in the main text and clearly defined

i) It is also not clear what methods in which scenarios were used - were deep learning methods necessary or did linear regression perform comparably in every scenario? It would be nice to have an illustration of performance in a variety of scenarios, where linear function is sufficient to capture the dependency of outcome on input and where non-linearity is essential.

Additionally,

Line 108: It is not correct that federated learning provides privacy. Federated learning provides confidentiality (the data is not shared with the central party) but private information is still contained in the gradients shared with the central party. To obtain privacy, one needs to combine federated learning with a differentially private algorithm like DP-SGD.

Line 209: The reference to Keras for a deep neural network is surprising: does this mean the network was implemented in Keras or was one of the tutorial models used to analyze the data? If the latter, what process was used to select the model as being appropriate for the data being analyzed here?

Line 232-233: The gain in accuracy compared to a model trained on node 1 only is not very significant, which would make it unlikely that node 1 gains from participating in the SL protocol. How does the proposed approach differ from selecting the node with the most balanced dataset and sharing its parameters with other nodes? Is this addressed by the fourth scenario?

Lines 427-428: The authors need to provide more details here on the inherent privacy properties of swarm learning. From what I understand, it only provides confidentiality. This would not satisfy the GDPR because individuals whose records are included in the datasets collected by participating nodes would still be at risk of identification and private attribute inference.

Appropriate use of statistics and treatment of uncertainties

As mentioned above, there are issues with the way accuracy is compared. There should not be three decision boundaries for the node-based models compared to one decision boundary for the SL approach.

Accuracy is not the ideal metric to be used, and should be contrasted with AUROC/AUPRC (which is used for some simulations but not all)

The authors do not present much detail as to how the SL was tuned, other than to say For some scenarios we also tuned Swarm Learning parameters to get better performance, for example higher sensitivity. This suggests that the SL algorithm might have overfit the data, as it does not appear that these hyperparameters were adjusted on a validation set.

Conclusions: robustness, validity, reliability

Datasets all use blood transcriptomics and have a level of signal that is non-representative of most machine learning datasets, thus limiting the generalizability of findings.

The privacy guarantees of the model are not convincing. The use of a block-chain and non-centralized learning is not sufficient.

Many details about how SL would be used and what its limitations are not covered in sufficient detail in the paper, omitting compelling evidence needed for deployment. More work needs to be shown on a theoretical as well as experimental level.

Suggested improvements: experiments, data for possible revision

Obtain additional datasets that are not only blood transcriptomics and have baseline accuracy/AUC below 90%.

Simulate realistic distributions of data to nodes.

Report results in standard measures, i.e. other than balanced accuracy, in the main paper.

Contrast performance of SL to federated data approach or full-data approach. Answer the question: how much worse is performance due to privacy guarantees from SL approach?

Other suggestions can be found in the Data & Methodology section.

References: appropriate credit to previous work?

There would seem to be much work in differential privacy and distributed learning literature that would help the authors improve their results and cite limitations to their approach. For example, non-IID data seems to decrease the performance of federated learning systems (see Federated Learning with Non-IID Data [Zhao et al 2018]). Appropriate comparisons in both related literature section and experimental section would improve the paper substantially.

Clarity and context: lucidity of abstract/summary, appropriateness of abstract, introduction and conclusions

Many important details of the paper are found in the appendix (normalization, for example). Fundamental description of how SL works should be in the main paper to help set expectations for the presented results.

A lot of detail is spent on explaining the dataset details in contrast to the SL algorithm's theoretical properties and practical implementation.

Much remains to be clarified, see suggestions above.

Referee #2 (Remarks to the Author):

The author developed a swarm learning method aiming at solving the problem of data segmentation and disconnection in COVID-19, tuberculosis or leukemia classification. The key innovation of this article is the usage of swarm learning, including a blockchain to keep track of all activity and a dynamic leader for model aggregation. The author conducted experiments to show performance of SL in predicting different diseases using simulated scenario. I have reviewed many manuscript recently in the field of federated/decentralized machine learning in healthcare, this is one of the most well written manuscripts with clear logic and presentation. In addition, the experiment design of simulated different scenario in using SL to predict diseases such as Leukemia is the most complete and thoughtful design I have seen in distributed learning in healthcare.

As this is a submission to Nature which need to meet the highest standards, I have the following comments

1. The experimental design for the five scenarios of data distribution is definitely good. In the field of federated learning, training on data with non-IID distribution has long been a challenge. In the current experiment design, the non-IID distribution come from factors such as data generation technologies (arrays vs. RNA-seq) I would suggest, if the data are available, include an 6th experiment that divide data into different side according age, gender or other demographic features will help further prove the robustness of the method.
2. Though I am really happy to see researchers in the field are conducting sophisticated simulating experiments, the biggest limitation of this study is actually that the experiments were simulated. What is very much needed in real practice is to show that SL or federated learning can be implemented in real systems where data are siloed in their original forms and is able train models with high accuracy. This is the main thing lack from this study.
3. I may have missed this point but the author used model trained from each data site as the "baseline" to compare with performance of SL. While models trained from each side can be used as the "lower boundary" for performance, a upper bound of "centralized learning" where model trained on aggregated data need to be calculated. Reduction of performance of SL compared with centralized learning need to be shown.
4. The author simulated 3 data sites for SL in many of the experiments. However, in reality, it is

unlikely that SL is needed if data are only distributed in 3 silos (eg. only 2 data transfer agreements needs to be down to aggregate the data)

Therefore, I would suggest the author at least increase the number to 10 or 20, to show that the strategy work in a robust manner.

5. In addition to point 4, one beauty of blockchain is that the records on blockchain can hardly be maliciously modified as many nodes has a cope / a part of the logs. If there are too few nodes in the blockchain , a majority attack will be easy. In this sense, three nodes is far too few.

6. How much safer a dynamic center is compared with a fixed center need to be further clarified.

7. Inclusion of COVID-19 data into this study is a plus give the current urgently needs for robust COVID-19 related models. However, the explanation of motivation of using COVID-19 data is not so convincing.

One big problem for most COVID-19 nucleic acid tests is the high false negative rate. The transcriptome based prediction method showed in this study could be used as a supplementary to nucleic acid test and serological tests. I think the value is not reflected in the manuscript

8. The texts in figure 3 and 4 are hardly readable and the boxplot size need to be adjusted such that the dot does not confust the boxes.

Referee #4 (Remarks to the Author):

In this paper, "Swarm Learning as a privacy-preserving machine learning approach for disease classification," Warnat-Herresthal, Schultze, and Shastry et al. use swarm learning (SL), a relatively new kind of machine-learning to examine over 14,000 blood transcriptomes derived from 100 individual studies with non-uniform distribution of cases and controls and significant study biases, to show the ability of SL to develop disease classifiers based on distributed data for COVID-19, tuberculosis, or leukemias that outperform those developed at individual sites. This method notably protects the privacy at distinct sites and is proposed as a new means by which people could apply machine learning to large-scale data. They also applied these ideas to a set of 132 COVID-19 patient blood profiles.

Overall the work is promising and could lead to broader use of these methods.

1) The algorithm describes cases and controls for their analysis of features and the selection of parameters, but how would this model work on more granular or continuous variable data?

2) Their figures are illustrative in concept but unclear on some details. For example, Figure 2a shows studies A-E, but the text and legend refer to A1, A2, A3, and B, and C, which is what I presume they meant. They just need to be consistent.

3) They permuted 100 times for most tests, but what would happen if they did this 500 or 1,000 times? How did they pick this number? Usually there is a parameter sweep that shows how a classifier threshold was chosen, and the authors may be able to improve their methods with a feature selection threshold sweep.

4) The COVID-19 data seems out of place here. With so few samples, the algorithms are almost certainly overfitting, but I don't see anything in the methods that shows how this is corrected? Can the authors clarify?

5) Also for the COVID-19 data, they show strong signals in the blood, but co-infection can and does happen (7-20% depending on the study) and the authors should examine if there is any impact of this on their metrics (especially if they can get this from the clinical metadata, which would be cool). Co-infection summaries are here: <https://www.ncbi.nlm.nih.gov/pmc/articles/PMC7255350>.

6) For the RNA-seq data, there are many features that can alter the accuracy and quantification dynamics, as well as dramatically different kinds of library preparations, but this does not seem to be addressed. The authors themselves noted that they were surprised that the RNA-seq and array data could be better merged here, but what about other features like read length or platform? Could they make a compelling case that it also does not matter for those features as well (for example, shown <https://genomebiology.biomedcentral.com/articles/10.1186/s13059-015-0697-y>)? This is one of the most compelling cases for these methods, is that even with the noise, you can make it through and find significant clinical features.

7) For the access to computational resources (e.g. XSEDE) and code base, can the authors describe how they deploy these methods and help others to use? This is a critical feature of data access, and their GitHub page (https://github.com/schultzelab/swarm_learning) only hosts a Jupyter notebook and a few scripts, but very limited annotation or descriptions of their methods. This can and should be expanded and made more explicit.

8) For the discussion, the authors should examine a bit more of the details of what cannot be seen with these methods. Related to #1, the authors claim to detect AML and ALL with high accuracy (96-96%), but what about the sub-classes of AML? There are some types of AML that show hypermethylator phenotypes or much more aggressive disease based on genetics/epigenetic differences, and even entirely novel classifiers based on genetic and epigenetic heterogeneity (e.g. <https://europepmc.org/article/pmc/pmc4938719>). They do approach this with acute vs. latent Tb, but it would be good to think about other parameters in other diseases, and if/how this SL approached could apply.

Author Rebuttals to Initial Comments (author comments are in blue):

Referee #1 (Remarks to the Author):

Summary of the key results

The paper introduces Swarm Learning (SL), a framework that allows for distributed machine learning in a safe and secure environment. The authors claim that the approach guarantees privacy through de-centralization and blockchain mechanisms.

The authors show that on datasets with high levels of signal, their approach systematically gets higher aggregate accuracy than on any individual node (e.g. edge computation). **The authors also show that as they try to make the simulation more and more realistic the performance deteriorates, and yet SL still outperforms each individual node.**

Originality and significance: if not novel, please include reference

Comment/Question 1

The approach is novel and would be of high significance, unfortunately the authors ***stop short from evaluating their model in realistic scenarios reducing the significance of the contribution.***

We thank the Reviewer for this overall very positive evaluation and providing us with a clear path and suggestions how we could optimize the manuscript and thereby our findings. Concerning more realistic scenarios, we have undertaken considerable effort, we generated new data for COVID-19 and included more medical centers (see lines 537-570), we also applied Swarm Learning to radiograms (lines 482-498) and increased the number of nodes in the leukemia example (lines 386-394). Overall, these additions have clearly improved the message of our manuscript.

Comment/Question 2

Also, to be precise, SL does not provide privacy per se, as is claimed by the authors. SL is still a novel approach in this Reviewer's eyes. It's just the portrayal is somewhat inaccurate. As presented, SL appears to be federated learning with extra privacy guarantees due to blockchain and computational shifting between leaders. Federated learning does not provide privacy, it provides confidentiality (the data is not shared with the central party) but private information is still contained in the gradients. To obtain privacy, one needs to combine federated learning with a differentially private algorithm like DP-SGD. The differential privacy literature shows that access to gradients/parameters of the model reduces data privacy. Adding noise to model gradients, which this paper does not use, is one method of adding additional privacy guarantees (see Abadi et al 2016 and Dwork et al 2006).

We thank the Reviewer for these very insightful comments. We have addressed them in several ways. First, we have asked our long-standing collaborator, Professor Michael Backes, Scientific Director of CISA (Helmholtz Center for Information Security) in Saarbrücken to join the team and to evaluate solutions. Together with him we came to the conclusion that while Swarm Learning is providing major and strongly elevated measures to enable privacy-preserving machine learning by design (as outlined by the Reviewer with extra privacy guarantees due to blockchain and computational shifting between leaders), similar to all other currently applied approaches (central cloud-based machine learning, federated cloud-based machine learning, local machine learning), it will need to address data reconstruction and membership inference attacks, which for example can be reached by differentially private algorithms like DP-SGD. We can follow the Reviewer's chain of arguments, but we also concluded that this problem is orthogonal to Swarm Learning as a new architecture for joined machine learning and as such not specific to Swarm Learning but Machine Learning itself. Consequently, we opted to tone down the context of privacy-preserving throughout the manuscript (see title (line 1f), abstract (line 103f, 110f), introduction (line 123f, 144ff, 156ff), results (line 222-224, 242-244, 264), and discussion section (line 592, 603, 668, 685ff, 695f)). Further, we added a paragraph in the Supplementary Information section to further elude to this point (lines 1389ff). Moreover, we make the reader aware of the issue raised by the Reviewer, namely, to distinguish between confidentiality and privacy-preserving (see line 156ff). Nevertheless, we see the avoidance of a central leader and the usage of block chain technology as major and conceptual improvements over federated machine learning approaches and therefore would like to keep the phrase "enabling privacy-preserving machine learning". If the Reviewer still feels that we should change towards "providing confidentiality" only, we will follow, but hope the solution provided is also practicable.

The text added to the Supplementary Information (lines 1389ff) is as follows:

The inherent trade-off between the effectiveness of a machine learning system and the privacy guarantees it can offer is subject to lively ongoing research. In this paper, we therefore decided not to integrate specific privacy mechanisms in the design of our Swarm Learning framework, but to make it compatible with previously used underlying machine learning models, including models that rely on privacy-enhancing technologies such as obfuscation and anonymization. The goal was to enable their seamless integration in the Swarm Learning framework. The framework itself does not rely on a fixed central aggregator, but instead implements a consensus-based selection of changing aggregators for every merge cycle by means of smart contracts. The framework moreover implements state-of-the-art security technologies (certificate-based access control, secure containment, network encryption) to protect data from direct unauthorized access.

Comment/Question 3

Overall, the approach is a novel and exciting direction and if the authors are able to demonstrate high performance in appropriately realistic scenarios, it will be of broad interest.

We thank the Reviewer for this overall very positive evaluation.

Data & methodology: validity of approach, quality of data, quality of presentation

Comment/Question 4

Tested scenarios are not realistic in many ways. Results, i.e. performance comparisons, are also not done appropriately. See the list of reasons supporting these statements below.

As outlined in the main section and in more detail below, we have taken several routes and actions to address all the suggestions by the Reviewer.

Comment/Question 5

a) The performance of the models is often in the high 90s. This indicates a very high signal to noise ratio, something that is very rare in realistic healthcare scenarios. The provided examples should vary both in type (not all blood transcriptomes) and signal-to-noise ratio.

We agree with the Reviewer, the performance of blood transcriptomics for leukemia and for COVID-19 is sometimes in the high 90s, for tuberculosis it is often below the 90s. In previous work, we demonstrated for lung cancer that the performance of blood transcriptomics was more variable also showing performances in the low 80s (Zander et al, Clin Cancer Res, 2011). We also agree with the Reviewer that certain feature spaces (such as blood transcriptomes) and certain diseases (such as the leukemias and COVID-19) can have a very high signal-to-

noise ratio. In fact, when studying the biology of bulk blood transcriptomes of COVID-19, we saw enormous reprogramming of immune cells, which explains the strong signal. We see this as a very valuable data space for machine learning approaches. However, we also agree with the Reviewer that blood transcriptomics is not yet used widely in clinical practice and that other feature spaces currently available in healthcare do have bad signal-to-noise ratios and therefore, it is much more difficult to establish classifiers with high performance in those data spaces. Therefore, this is one more reason to assess blood transcriptomics for clinical questions.

To address the points raised by the Reviewer, we now have included Swarm Learning on X-rays in the manuscript and demonstrate that a feature space other than blood transcriptomics and with lower signal-to-noise ratio also performs well when Swarm Learning is applied (see lines 482ff, **Fig. 3m-n**). Further, we extended the COVID-19 analyses and included samples from SARS-CoV-2 negative patients with sepsis, which show an overlap in biology with COVID-19, therefore harbor overlapping changes, we provide unbalanced scenarios only training e.g., on males and testing on females or using test nodes with very different disease severity distributions. All these aspects introduce additional scenarios (lines 537ff, **Fig. 4e-f, Extended Data Fig. 9-10**). All together, we illustrate that Swarm Learning under these conditions still performs well.

Comment/Question 6

b) The test set consists of IID draws from each node, and then each local (or edge) model has its accuracy recorded on its respective test patients. However, the SL model makes predictions on all the nodes test set patients. It would be more appropriate to compare the pairwise accuracy between Node j 's test set patients on the local model and the SL model, i.e. reported performance should be done on the same set of patients. There will then be as many comparisons as their nodes and appropriate multiple corrections will have to be made.

We thank the Reviewer for the comment and apologize that we have not been clear enough about our experimental setup. First, training and test samples were always independent, we have clarified this in the Result Section (lines 276ff) and in the Supplementary Information Section (lines 1382f) and we have also added this more clearly now to the workflows (**Extended Data Fig. 3a, 3g**). Second, indeed, for testing we always used an independent test set (we refer to the “global test set” in the revised manuscript, line 280f), which was used for evaluating statistical performance for each individual node as well as the Swarm. In other words, we did not use node-specific test sets but always used the same set of patients as a global test set for all nodes and the swarm prediction in all scenarios, as suggested by the Reviewer.

We entirely agree with the Reviewer a node-specific “local test set” would share sample

characteristics (distribution, sample batch) with the training samples derived from the same

node and would likely be an overfitting scenario for the nodes. We agree with the Reviewer that choosing the right test set for prediction is crucial. To make this point even more clear, we added the predictive performances of node-specific test sets (“local test sets”) to a scenario where the nodes resemble a potentially very high batch effect (Fig. 2d) and report these in a new supplementary figure (**Extended Data Fig. 3d**). The prediction results are at par with the performance of SL on the global test set, which combined all batches across nodes. We added a workflow which summarizes the overall setup for this scenario (**Extended Data Fig. 3a**). A second use case, for which we now can report this effect is COVID-19 (**Extended Data Fig. 9b**) for which it is similarly apparent that locally testing would overestimate classifier performance.

Comment/Question 7

c) A more realistic setting would be to have test set patients which are from a different dataset/source, not drawn IID from existing patients. This would more properly show the ability of the SL model to generalize. It also would be nice to see the comparison to the best achievable performance, i.e. by the model that is trained on all the data.

Again, we sincerely apologize to the Reviewer that we did not make our experimental setup clear enough. As mentioned above, training and test data were always independent for all scenarios tested, addressing different factors that might influence performance of SL. Considering the leukemia use case, for **Fig. 2b, c and e**, samples were randomly selected from each clinical study for the global test set. In other words, while the global test dataset contained samples from the clinical studies (distributed to the training nodes), these were independent from the training samples. We always used the newly formed global test dataset and termed this independent dataset as the test node. Here the goal was to determine the effect of sample size and distribution of cases and controls (**Fig 2b, c and e**). In another scenario (**Extended Data Fig. 3g-h**), we distributed a set of studies reported in the literature to independent nodes (here we did not allow samples from one study to be distributed to several nodes) and the test node (the “global test dataset”) contained yet another independent set of studies. This scenario increases batch effects and we wanted to assess how SL would perform under this realistic scenario. One could imagine the following situation in real life: Independent research consortia (each one node) might have collected their study data in a smaller central data space to apply further machine learning on the samples from the individual studies. Each of the consortia, however, would not be able to share data with others, e.g., due to legal or data privacy constraints, something that often happens in medicine. In other words, a further centralization would not be possible. Here SL can be applied by connecting the different consortia. In lack of the application of a completely centralized model, SL is the best model to utilize even more data and we can actually demonstrate that SL outperforms the individual nodes (e.g., consortia of sets of studies). We

also think that this scenario is one part of the answer to the valuable point of the Reviewer concerning generalizability.

The Reviewer further suggested a central model (train on all data). We provide this information for the leukemia case in **Extended Data Fig. 2b, 2h, 3c, 4c and 4h** and for the tuberculosis cases in **Extended Data Fig. 7b, c**. We would like to take the liberty to mention that a completely centralized model is probably not a realistic scenario considering the many constraints that preclude sharing data around the world. In other words, even if data are centralized to certain data spaces, it is unlikely that data at different central data spaces would be all centralized at one unified central data space. However, SL could connect even data placed at such centralized data spaces. Here each central data space would become a node of the Swarm. Considering this scenario, and considering the performance we observed for SL, we postulate that SL of centralized data spaces (“clouds”) will outperform each individual cloud.

We further addressed this valuable point by the Reviewer in the part concerning prediction of COVID-19. By including training data sets based on samples derived from additional medical centers with very different patient scenarios (only children vs. only ICU patients vs. ambulatory + hospitalized mild to severe patients, vs mainly convalescence patients). Here training was performed on individual medical centers and tested on independent other medical centers. Again, SL outperformed individual nodes (**Extended Data Fig. 9h-i**).

Comment/Question 8

d) The authors assign the number of nodes independent of the data sources, whereas the nodes should always be composed of the same source - again, to set the realistic scenario.

Again, we apologize that we have not been clear enough in the primary submission. To better understand the performance of SL we assessed the influence of different aspects that occur in real-life scenarios. For each of these scenarios, the samples at a given node were fixed to the node, in other words, under any given scenario, each node was always composed of the defined set of samples. It is however also true, that when addressing different questions, the composition of the samples (e.g., prevalence between cases and controls) at any given node in the scenarios was changed.

Taking the leukemia case as the example, two extreme but realistic scenarios were included. In one scenario (**Extended Data Fig. 3g-h**), we simulated the situation, where three larger consortia (here the nodes) are independently addressing whether leukemia can be detected by AI using their respective set of individual clinical studies. A fourth, yet independent consortium (the test node with the global test dataset) agreed on benchmarking the individual results from the three consortia. Here comes SL in. By learning on all data from the

three test nodes (the three independent consortia), we illustrate on the global test dataset that SL outperforms each individual consortium without sharing the data.

In another realistic scenario (**Figure 2f**), we addressed the influence of different experimental technologies to obtain the data, here the generation of blood transcriptomes by different microarrays and RNA-seq technology. It is well known in the field that use of these different technologies has a significant impact on the data itself. However, we postulated that SL should be able to extract a feature space that would perform better than if a node is trained only on one technology. Indeed, this is the case. Even, if technologies are highly standardized, we still see differences between different centers. This is also true when comparing studies that used different RNA-seq procedures, as we now assessed with the new Extended Data Figures 7d-e. Here, SL outperforms the nodes that have been trained only on one RNA-seq sequencing protocol. In the newly provided COVID-19 case (**Fig. 4e-f** and **Extended Fig. 9-10**), we also see an influence of the sequencing procedure on the data quality. Yet, SL can clearly cope with these existing real-world issues rather well. We have re-written the text in the result section to a large extent (see yellow marked text) and added additional supplementary information (lines 1275ff) to better explain the intention of the scenarios and why these reflect situations that are happening in current medical research and practice.

Comment/Question 9

e) It would be useful to see how the algorithm scales with many nodes (the most shown in the paper was six), but in many multisite studies there could be hundreds of nodes. How many nodes can you add with noisy data before SL breaks down?

We agree with the Reviewer, in the long run, SL will be of particular interest for situations, where data are available at many places, yet data sharing and transport is not possible. We had shown that increasing from three to six nodes in the TB case did not change SL performance but rather individual node performance. We now added several additional settings with larger numbers of nodes. First, we increased to 32 nodes in the leukemia case and illustrate that swarm outperforms the nodes also in this case (**Extended Data Fig. 5h-i**). We have not yet seen a breakdown of SL, when getting towards the current limits of simulating larger numbers of nodes on our current hardware setup.

f) Authors do not provide illustration/means for onboarding a new node with new data and using a trained SL to make predictions. Perhaps they suggest to retrain/re-ensemble SL every time? If that is the case, how do they guarantee safety and efficacy of the approach over time (something that health authorities normally require to be used in practice/on patients).

We agree with the reviewer that this is indeed a very important consideration to take into account and we would like to take the opportunity and explain in greater detail how onboarding of new node members would work:

First, Swarm Learning supports dynamic addition of new nodes into the Swarm network. A node can join the Swarm network any time after proper authorization. The newly added node waits for the next synchronization round (merge process) to get the model with merged weights at this synchronization interval. The node then starts training the model with its local data and participates in the merge process beginning with the next synchronization level. Thus, the newly added node starts its training from the point where all the other nodes in the network are, meaning the full model does not have to be trained again.

Following the suggestion of the reviewer, we conducted an experiment to show this process by generating the accuracy graph for the training process (**Extended Data Fig. 5j**). In the graph we plot the accuracy of the model at every epoch. This is like the graph one can get by integrating with TensorBoard. For the experiment we distributed the training data across 6 nodes with varying case: control ratios.

First, we trained the swarm model with 3 nodes (nodes 1-3 having low prevalence). This model achieved an accuracy of 83% at the end of the 80th epoch (red line in **Extended Data Fig. 5j**). Then, we trained a swarm model with 3 nodes (nodes 4-6 having high prevalence). This model achieves an accuracy of 96% at the end of 80th epoch (blue line in **Extended Data Fig. 5j**). Finally, we conducted a swarm training starting with 3 nodes (node 1-3 having low prevalence), during the training process we on-boarded 3 more nodes (node 4-6 having high prevalence) during the 39th epoch. The newly added nodes started to contribute to swarm learning from the 40th epoch on and improved the training accuracy significantly. This 6 node swarm model achieved an accuracy of 97% at the end of the 80th epoch (green line in **Extended Data Fig. 5j**).

Comment/Question 10

g) The authors show deterioration of performance for unbalanced datasets. Reporting balanced accuracy is misleading at all times - one can train on balanced datasets, but should always strive to test using realistic case/control ratio. ALL is not very common at the hospital, presumably that is the scenario where such a tool would be used, so a realistic simulated test should reflect that case/control ratio.

We agree with the Reviewer, realistic case/control ratios are of importance to judge the overall performance of any given medical test and blood transcriptomics and machine learning are no exception. We also agree, at regular hospital wards, leukemias are not quite common, while the prevalence of leukemias at hematology departments is extremely high. Therefore, incidence and prevalence dramatically depend on circumstance. We had previously addressed this issue for acute myeloid leukemia (Warnat-Herresthal et al. 2020)

illustrating that the positive predictive value is dependent on the prevalence as expected.

Following the suggestion of the Reviewer, we have added a new scenario testing different prevalence for acute lymphatic leukemia (ALL), which is added in **Extended Data Fig. 5a-e**. As expected, performance is reduced in lower prevalence situations, however, SL clearly outperforms individual nodes, and the differences become more prominent the lower the prevalence is. We have also changed the text accordingly in the result section (lines 371ff) and the methods section (lines 1293ff).

Comment/Question 11

h) It is not clear how the parameter for ensembling various models is picked - this is a foundational part of the performance of the SL and should be brought up in the main text and clearly defined

We thank the Reviewer for this helpful comment, and we are more than happy to further elucidate these points here and in the manuscript. SL does not use model ensemble techniques. Like many of the federated learning models, SL does averaging of model parameters. The elected epoch leader is responsible for merging and averaging the model parameters. In the simplest form, the leader contacts every swarm node that has expressed willingness to participate in this current merge round, downloads their model parameters via a secure P2P communication channel. It performs the averaging (by default SL does simple average but has options for weighted average or custom merge algorithms) and redistributes the merged weights. The swarm nodes receive and apply the merged parameters to their respective local model. This completes one merge cycle. The swarm nodes now re-train the model with another batch of local data and restart the merge cycle again. This process continues until the Swarm network collectively decides it is the time to stop. This information is now added to the main text (lines 233ff, lines 249ff) and supplementary information (lines 1513ff).

Comment/Question 12

i) It is also not clear what methods in which scenarios were used - were deep learning methods necessary or did linear regression perform comparably in every scenario? It would be nice to have an illustration of performance in a variety of scenarios, where linear function is sufficient to capture the dependency of outcome on input and where non-linearity is essential.

We agree with the Reviewer, once SL is established as a valid approach to perform machine learning on decentralized data, it will be important to address which methods perform best

on which data. As outlined by the Reviewer, one important aspect will be to benchmark whether different linear regression models can perform as well as more sophisticated models or deep neural networks. We have previously benchmarked linear models on the leukemia data and did not find dramatic differences in performance (Warnat-Herresthal et al. 2020).

Following the Reviewers recommendation, we now tested one of these linear models on the leukemia datasets. As can be seen in **Extended Data Fig. 6b**, the overall performance is not as high as with DNN (**Extended Data Figure 6c**), however, SL again outperformed individual nodes. We added additional text as well (lines 394ff, lines 1271f, lines 1290f, lines 1582f, lines 1608ff).

Comment/Question 13

Line 108: It is not correct that federated learning provides privacy. Federated learning provides confidentiality (the data is not shared with the central party) but private information is still contained in the gradients shared with the central party. To obtain privacy, one needs to combine federated learning with a differentially private algorithm like DP-SGD.

We interpret this statement as a follow up on previous comment/question 2 by this Reviewer and refer to our extended answer on this earlier question. In brief, we have made changes throughout the text to accommodate the Reviewer's comment (see title (line 1), abstract (lines 103, 110), introduction (lines 123, 144ff, 156ff), result (line 222ff, line 242ff, line 264) and discussion sections (line 592, 603, 668, 685ff) as well as Supplementary Information section (line 1389ff).

Comment/Question 14

Line 209: The reference to Keras for a deep neural network is surprising: does this mean the network was implemented in Keras or was one of the tutorial models used to analyze the data? If the latter, what process was used to select the model as being appropriate for the data being analyzed here?

We apologize to the Reviewer if we were not sufficiently explicit about what was done. We used the Keras Framework for our python implementation of our deep learning neural networks. The text has been updated accordingly (lines 274f, 1586ff). Further, we used a sequential model and parameters were specified as detailed in Supplementary Table 8.

Comment/Question 15

Line 232-233: The gain in accuracy compared to a model trained on node 1 only is not very significant, which would make it unlikely that node 1 gains from participating in the SL

protocol. How does the proposed approach differ from selecting the node with the most balanced dataset and sharing its parameters with other nodes? Is this addressed by the fourth scenario?

We thank the Reviewer for very careful evaluation of our results. As also requested by this Reviewer, we now also provide the AUC, which shows that SL outperforms node 1 in this scenario. Furthermore, we indeed intended to address what happens if nodes are already showing high performance. Would SL also outperform such nodes? As correctly mentioned by the Reviewer, this is addressed in **Figure 2f** where we ‘node-optimized’ the scenario. Still SL outperforms all nodes, clearly suggesting that even under such conditions a Swarm would be favored over nodes derived from individual results. We have provided additional information to the text in the Result section (lines 289-323) to better address this point raised by the Reviewer.

Comment/Question 16

Lines 427-428: The authors need to provide more details here on the inherent privacy properties of swarm learning. From what I understand, it only provides confidentiality. This would not satisfy the GDPR because individuals whose records are included in the datasets collected by participating nodes would still be at risk of identification and private attribute inference.

We interpret this statement as a follow up on previous Comment/Question 2 by this Reviewer and refer to our extended answer on this earlier question. In brief, we have made changes throughout the text to accommodate the Reviewer’s comment (see title (line 1), abstract (lines 103, 110), introduction (lines 123, 144ff, 156ff), result (line 222ff, line 242f, line 264) and discussion sections (line 592, 603, 668, 685ff) as well as Supplementary Information section (line 1389ff).

Appropriate use of statistics and treatment of uncertainties

Comment/Question 17

As mentioned above, there are issues with the way accuracy is compared. There should not be three decision boundaries for the node-based models compared to one decision boundary for the SL approach.

We apologize to the Reviewer that we did not make this clear enough. We would also like to refer the Reviewer to our extended answers to Comment/Question 6, 7, and 8. In brief, both nodes, and SL were tested with the same general test dataset. Changes have been made as detailed further above.

Comment/Question 18

Accuracy is not the ideal metric to be used, and should be contrasted with AUROC/AUPRC (which is used for some simulations but not all)

We have followed the Reviewer's suggestion and have added AUC to most use cases. The information has been added throughout the text and is also provided in Supplementary Table 4.

Comment/Question 19

The authors do not present much detail as to how the SL was tuned, other than to say For some scenarios we also tuned Swarm Learning parameters to get better performance, for example higher sensitivity. This suggests that the SL algorithm might have overfit the data, as it does not appear that these hyperparameters were adjusted on a validation set.

We apologize to the Reviewer if the information was not sufficient. Swarm Learning has two tunable parameters – one is to tune the frequency of merges and the other one is to select the merge algorithm. We have used these parameters to tune the performance of Swarm models. Merge frequency is tuned based on the training dataset size and the batch size to improve the efficiency of SL. By default, SL uses a simple average algorithm for merging the parameters. In most scenarios we used default settings. In cases where a node has very few control data points, we have tuned the SL to use a weighted merge algorithm and give less weightage to the parameters from the node which has less control data. Model specific hyperparameters, like learning rate, dropouts, etc. are set the same for both the local node models and the SL model. This information has been provided to the Supplementary Information section (lines 1631ff) and as the Supplementary Table 8.

Concerning overfitting, the signal for COVID-19, but also leukemia is very strong, and the model is able to learn with very few training samples. To address overfitting in our use cases, we cross validated training and validation loss to ensure the models are not overfitting. This information is provided as **Extended Data Figure 8I**.

Conclusions: robustness, validity, reliability

Comment/Question 20

Datasets all use blood transcriptomics and have a level of signal that is non-representative of most machine learning datasets, thus limiting the generalizability of findings.

We agree with the Reviewer, in the initial submission we focused on blood transcriptomics since this is a data space with strong disease signals, a data space that is underestimated in medicine and certainly requires more attention in the future. As such, we feel that the introduction of blood transcriptomics for the identification of patients with a certain disease

brings additional novelty, as also suggested by Reviewer 2. As outlined in an extended answer to Comment/Question 1 and 5, we have also added SL scenarios based on chest X-rays and

demonstrate that SL also performs well with this completely different data space. This information is added in lines 482ff.

Comment/Question 21

The privacy guarantees of the model are not convincing. The use of a block-chain and non-centralized learning is not sufficient.

We interpret this statement as a follow up on previous Comment/Question 2 and 16 by this Reviewer and also refer to our extended answers on these earlier questions. In brief, we have made changes throughout the text to accommodate the Reviewer's comment (see title (line 1), abstract (lines 103, 110), introduction (lines 123, 144ff, 156ff), result (line 222ff, line 242f, line 264) and discussion sections (line 592, 603, 668, 685ff) as well as Supplementary Information section (line 1389ff).

Comment/Question 22

Many details about how SL would be used and what its limitations are not covered in sufficient detail in the paper, omitting compelling evidence needed for deployment. More work needs to be shown on a theoretical as well as experimental level.

We thank the Reviewer for pointing out improvements in our manuscript, particularly concerning evidence for deployment of SL technology. Concerning technical deployment, the SL software is packaged and delivered as docker containers, deployed decentralized using any container orchestration mechanism, including Kubernetes. During initialization Swarm containers are installed and configured to interact with the Swarm network (additional text see lines 249ff). For review we provide a sandbox environment which hosts a 3 node Swarm network with the AML dataset distributed among the nodes and a test node. Shell scripts are provided to run the experiments using this Swarm network and the AML dataset. Reviewers can request access to the system via the Editors, who will be provided with respective access codes. We have undertaken considerable effort to make the setup of a Swarm network as easy as possible, nevertheless additional work is under way to make communication in the network even more secure by applying policy-based mutual authentication by integrating with identity providers. Currently, SL has limitations, e.g., all nodes within a swarm use the same model, every node needs to use the same ML platform and SL works only for models that can be parameterized. This information has now been added to the manuscript (The Swarm Learning architecture principles lines 1450ff).

To better understand limitations and possibilities of Swarm Learning, we have increased the COVID-19 samples and scenarios (lines 499ff) and have introduced a new data type, chest X-ray images (lines 482 ff). Furthermore, we have studied scenarios with different Swarm configurations (**Extended Data Fig. 5h-i**).

We further studied the robustness of SL towards different data production technologies in transcriptomics. We assessed the impact of different RNA-seq protocols on predictive performance and if SL could overcome these limitations. In a sixth scenario, we split the data accordingly (**Extended Data Fig. 4i**). While data used at node 1 was sequenced with 100 bp paired end reads on an Illumina HiSeq 2000 instrument, the AML cases from node 2 were sequenced with 50 bp paired end reads using an Illumina HiSeq 2500 instrument. Library preparation was performed using TruSeq library preparation kit in both cases. (**Extended Data Fig. 4j**, details on all included studies are listed in Supplementary Table 6). We illustrate that SL outperforms the nodes with unequal distribution of the data. (line 296ff).

Suggested improvements: experiments, data for possible revision

Comment/Question 23

Obtain additional datasets that are not only blood transcriptomics and have baseline accuracy/AUC below 90%.

We interpret this statement as a follow up on previous Comment/Question 1, 5 and 20. We refer the Reviewer to the more extended answers above. Briefly, we have added SL based on chest X-rays and demonstrate that SL also performs well with this completely different data space. This information is added in lines 482ff.

Comment/Question 24

Simulate realistic distributions of data to nodes.

We interpret this statement as a follow up on previous Comment/Question 1 and would like to refer the Reviewer to the more extended answers above. Briefly, we have added additional scenarios with more nodes. This information is now provided as new **Extended Data Fig. 5i** and in the Result section (lines 386ff).

Comment/Question 25

Report results in standard measures, i.e. other than balanced accuracy, in the main paper.

If we understand the Reviewer's comment correctly, the Reviewer asks to add additional performance measures to the main paper, e.g., for leukemia where we initially only focused

on accuracy in the main figure and sensitivity and specificity in Extended Data figures. To address this comment, we now also added AUC and F1 scores (see Supplementary Table 4). Due to space constraints and the addition of new data, it was not possible for us to add all information to the main figures. We hope the Reviewer can agree with the way we now provide all the new information.

Comment/Question 26

Contrast performance of SL to federated data approach or full-data approach. Answer the question: how much worse is performance due to privacy guarantees from SL approach?

We thank the Reviewer for this comment. As outlined in more detail in our extended answer to Comment/Question 7 we followed the Reviewer's suggestion and added a central model. We would like to reiterate that a completely centralized model is probably not a realistic scenario considering the many constraints that preclude sharing data around the world. In other words, even if data are centralized to certain central data spaces, it is unlikely that data at different central data spaces would be all centralized at one major central data space. However, SL could connect even data placed at different central data spaces. Here each such data space would become a node of the Swarm. Considering this scenario, and considering the performance we observed for SL, we postulate that SL of central data spaces will outperform each such data space.

Comment/Question 27

Other suggestions can be found in the Data & Methodology section.

We propose the Reviewer is pointing towards the questions concerning data and methodology which we answered at the respective questions.

References: appropriate credit to previous work?

Comment/Question 28

There would seem to be much work in differential privacy and distributed learning literature that would help the authors improve their results and cite limitations to their approach. For example, non-IID data seems to decrease the performance of federated learning systems (see Federated Learning with Non-IID Data [Zhao et al 2018]). Appropriate comparisons in both related literature section and experimental section would improve the paper substantially.

We thank the Reviewer for this interesting comment. Considering the first part of the comment, we interpret this statement as a follow up on previous Comments/Questions 2, 13, 16, 21 and 22 by this Reviewer and refer to our extended answers on these earlier questions. Concerning the mentioned paper by Zhao and the suggestion to address this issue, we now provide additional information in the new COVID-19 use cases (**Fig. 4e-f, Extended Data Fig.**

9c). In **Fig. 4** we follow - in principle - the suggested solution by Zhao. Here we use an independent global test dataset at the test node, which is derived from 20% of the data (independent of the training data) provided by the training nodes. This global test dataset is used to assess individual training node and SL performance. In an alternative approach, we

replace this global test node by a completely independent node (here node 7 (an independent medical center, **Extended Data Fig. 9c**) and show that the Swarm performs similarly well in both scenarios. This is further described in the text in the Result Section (lines 545-556). Here we also cite Zhao et al. 2018.

Clarity and context: lucidity of abstract/summary, appropriateness of abstract, introduction and conclusions

Comment/Question 29

Many important details of the paper are found in the appendix (normalization, for example). Fundamental description of how SL works should be in the main paper to help set expectations for the presented results.

We see the point of this Reviewer. Indeed, we tried to write the manuscript so that potential users in the medical domain would also find the report of interest. Reviewer 2 stated in his/her summary of the manuscript *“I have reviewed many manuscripts recently in the field of federated/decentralized machine learning in healthcare, this is one of the most well written manuscripts with clear logic and presentation.”* To address the Reviewers comment, we have added further information concerning the fundamental description of how SL works in the main text now (see lines 233-266). We hope that the Reviewer can agree with this strategy.

Comment/Question 30

A lot of detail is spent on explaining the dataset details in contrast to the SL algorithm’s theoretical properties and practical implementation.

We see the point of this Reviewer. Indeed, we tried to write the manuscript so that potential users in the medical domain would also find the report of interest. Reviewer 2 stated in his/her summary of the manuscript *“I have reviewed many manuscripts recently in the field of federated/decentralized machine learning in healthcare, this is one of the most well written manuscripts with clear logic and presentation.”* Nevertheless, we agree with the Reviewer to add additional information on theoretical properties and practical implementation, since this is similarly important. We have added this information now in the more technical section of the paper (lines 233-266). Further, we make a clear link to this section in the main text (lines 233-266). We hope that the Reviewer can agree with this strategy.

Comment/Question 31

Much remains to be clarified, see suggestions above.

We followed the Reviewer's suggestions above

Referee #2 (Remarks to the Author):

The author developed a swarm learning method aiming at solving the problem of data segmentation and disconnection in COVID-19, tuberculosis or leukemia classification. The key innovation of this article is the usage of swarm learning, including a blockchain to keep track of all activity and a dynamic leader for model aggregation. The author conducted experiments to show performance of SL in predicting different diseases using simulated scenario. I have reviewed

many manuscript recently in the field of federated/decentralized machine learning in healthcare, this is one of the most well written manuscripts with clear logic and presentation. In addition, the experiment design of simulated different scenarios in using SL to predict diseases such as Leukemia is the **most complete and thoughtful design I have seen in distributed learning in healthcare.**

We thank the Reviewer for the detailed review and the very positive words and the appreciation of our work.

As this is a submission to Nature which need to meet the highest standards, I have the following comments

Comment/Question 1

1. The experimental design for the five scenarios of data distribution is definitely good. In the field of federated learning, training on data with non-IID distribution has long been a challenge. In the current experiment design, the non-IID distribution come from factors such as data generation technologies (arrays vs. RNA-seq) I would suggest, if the data are available, include an 6th experiment that divide data into different side according age, gender or other demographic features will help further prove the robustness of the method.

We thank the Reviewer for the excellent suggestions to further illustrate the strength of Swarm Learning. As suggested, we addressed additional parameters such as age, gender, and other demographic features in adding new data as well as new training and test cases. Age and gender as potential influencing factors have been addressed in the COVID-19 use case by performing two additional scenarios where we trained on sex- and age-biased nodes and tested on a mixed dataset (**Extended Data Fig. 10**). As can be clearly seen, SL is outperforming even under these conditions. Results have been described in lines 571-583.

As an additional demographic factor, we used location in the COVID-19 scenario, for which the COVID-19 cases for training were strictly kept at the swarm node (medical center), at which the patients were actually treated. COVID-19 patients at the included medical centers

were extremely heterogeneous ranging from mainly children and adolescent patients at one center, only ICU patients at another and mainly convalescent patients at yet another medical

center. These new use cases have been added as new **Figure 4e-f** and **Extended Date Fig. 9e-f** and described in lines 537-564.

Comment/Question 2

2. Though I am really happy to see researchers in the field are conducting sophisticated simulating experiments, the biggest limitation of this study is actually that the experiments were simulated. What is very much needed in real practice is to show that SL or federated learning can be implemented in real systems where data are siloed in their original forms and is able train models with high accuracy. This is the main thing lack from this study.

We thank the Reviewer for this insightful comment. In our revised manuscript, we have addressed the Reviewers' suggestions by improving the descriptions of the existing scenarios (see lines 1275-1358), by testing more nodes within the existing datasets (Extended Data Fig. 9d, lines 390ff), by generating new data from additional COVID-19 cases at additional institutions and providing these in even more realistic scenarios (**Fig. 4e-f, Extended Data Fig. 9-10**, lines 537-583) and by extending the model to chest X-rays as a second medical data space (**Figure 3m-n**, lines 482ff). Concerning the description of the initial scenarios we apologize to the reviewer. Even though we introduced simulations, in each scenario, the data were siloed. We now provide information in the text making this clearer (lines 278f, 321, 333, 351, 374, 387f).

Comment/Question 3

3. I may have missed this point but the author used model trained from each data site as the "baseline" to compare with performance of SL. While models trained from each side can be used as the "lower boundary" for performance, a upper bound of "centralized learning" where model trained on aggregated data need to be calculated. Reduction of performance of SL compared with centralized learning need to be shown.

We understand and address this point. The Reviewer suggests a central model (train on all data = aggregated data). We provide this information for the leukemia case in **Extended Data Figures 2b, 2h, 3c, 4c and 4h** and for the tuberculosis cases in **Extended Data Fig. 7b, c**. We would like to take the liberty to mention that a completely centralized model is probably not a realistic scenario considering the many constraints that preclude sharing data around the world. In other words, even if data are centralized to certain data spaces, it is unlikely that

data at different central data spaces would be all centralized at one such space. However, SL could connect even data placed at different central data spaces. Here each such space would become a node of the Swarm. Considering this scenario, and considering the performance we observed for SL, we postulate that SL of central data spaces will outperform each individual central data space.

Comment/Question 4

4. The author simulated 3 data sites for SL in many of the experiments. However, in reality, it is unlikely that SL is needed if data are only distributed in 3 silos (eg. only 2 data transfer agreements needs to be down to aggregate the data) Therefore, I would suggest the author at least increase the number to 10 or 20, to show that the strategy work in a robust manner.

We see the point the reviewer is making. We therefore have now also added scenarios with more nodes to illustrate this point. This is now added in **Extended Data Fig. 5h-i** and in the text (lines 386ff).

However, we would like to take the liberty to suggest that although we think that simple partnerships (e.g., between three collaborating academic partners) might not need Swarm, this will still open the opportunity to use data, where three partner want to work together, but cannot or do not want to share the data. Another disadvantage of the aggregated data is the need for data transfer. Also, here, we have experienced delays due to lengthy data transfer times from partners with bad connections. Again, only exchanging modeling parameters, while leaving the data locally expedites the process.

Comment/Question 5

5. In addition to point 4, one beauty of blockchain is that the records on blockchain can hardly be maliciously modified as many nodes has a cope / a part of the logs. If there are too few nodes in the blockchain , a majority attack will be easy. In this sense, three nodes is far too few.

We agree with the reviewer's comments and therefore have added scenarios with additional nodes (see text lines 386ff and **Extended Data Fig. 5h-i**)

Comment/Question 6

6. How much safer a dynamic center is compared with a fixed center need to be further clarified.

We thank the reviewer for this comment and have addressed it in the Result section (lines 242f, 249ff) and the Supplementary Information section (lines 1512ff). A dynamic center

ensures that the system is democratic and avoids any monopolies. A dynamic center also ensures that the system does not have a single point of failure. If a node fails, the Swarm Learning network chooses a leader among the available nodes as long as the minimum number of nodes criteria is met. Whereas in a fixed center system the failure of the central node will halt the learning process.

The Swarm Learning network is run on a fully decentralized control anchored to a blockchain ensuring accountability and traceability of actions. Having decentralized control ensures that a single participant cannot change the rules of engagement unilaterally. These architectural attributes provide a brokered-trust environment which is comparatively safer than a centrally controlled system. So, the members can experience an equal sense of participation.

Comment/Question 7

7. Inclusion of COVID-19 data into this study is a plus give the current urgently needs for robust COVID-19 related models. However, the explanation of motivation of using COVID-19 data is not so convincing. One big problem for most COVID-19 nucleic acid tests is the high false negative rate. The transcriptome based prediction method showed in this study could be used as a supplementary to nucleic acid test and serological tests. I think the value is not reflected in the manuscript

We thank the Reviewer for this comment and we completely agree. In the first submission, we probably have been too conservative about the value the assessment of the host's response based on robust signatures in blood transcriptomes could have to detect COVID-19 patients in comparison to viral testing - mainly by PCRs - that can be false negative. With the addition of 110 acute and 183 convalescent COVID-19 samples, we have further strengthened the value of our approach and we therefore decided to follow the Reviewers suggestion to better reflect the value. This is now added to the manuscript (line 504f, 636ff)

Comment/Question 8

8. The texts in figure 3 and 4 are hardly readable and the boxplot size need to be adjusted such that the dot does not confust the boxes.

We apologize for falling short in high quality of these figures and have revised the figures and as requested by the Editors adapted the size of the letters within all figures to the one outlined in the Nature style sheet.

Referee #4 (Remarks to the Author):

In this paper, "Swarm Learning as a privacy-preserving machine learning approach for disease classification," Warnat-Herresthal, Schultze, and Shastry et al. use swarm learning (SL), a relatively new kind of machine-learning to examine over 14,000 blood transcriptomes derived from 100 individual studies with non-uniform distribution of cases and controls and significant study biases, to show the ability of SL to develop disease classifiers based on distributed data

for COVID-19, tuberculosis, or leukemias that outperform those developed at individual sites. This method notably protects the privacy at distinct sites and is proposed as a new means by which people could apply machine learning to large-scale data. They also applied these ideas to a set of 132 COVID-19 patient blood profiles.

Overall the work is promising and could lead to broader use of these methods.

We thank the reviewer for the careful evaluation of our work and for the very positive comments.

Comment/Question 1

1) The algorithm describes cases and controls for their analysis of features and the selection of parameters, but how would this model work on more granular or continuous variable data?

This is an excellent question by the reviewer and we totally agree that our initial report will open up many new avenues of which the reviewer suggests assessing more granular or continuous data. To address this important point already in this report, we have chosen to provide more granular models, the first one being a multi-class predictor for the different leukemias (lines 379ff, Extended Data Fig. 5f-g), the second one a new data set addressing a multi-class problem in chest X-rays (lines 482ff, 1122ff, Fig. 3m-n). We can show that Swarm performs well for these additional settings. Thirdly, we provide a more diversified, more granular scenario for COVID-19, addressing the influence of very heterogeneous data at each individual Swarm node (lines 537ff, Extended Data Fig. 9). Here we include additional centers with different COVID-19 patient cohorts. Last, we assess the influence of factors such as age or sex on performance (lines 571ff, Extended Data Fig. 10), which suggests that Swarm Learning can handle such influencing parameters as well.

Comment/Question 2

2) Their figures are illustrative in concept but unclear on some details. For example, Figure 2a shows studies A-E, but the text and legend refer to A1, A2, A3, and B, and C, which is what I presume they meant. They just need to be consistent.

We apologize for not being clear enough, we have changed this according to the suggestions

by the Reviewer. The datasets used here are now numbered A-E with A split into A1, A2, A3 and E split into E1-8. Within dataset A, the data are derived from 127 individual clinical studies. Within the schemata, we now call these studies s_1 , s_2 , s_3 and so forth. See Fig. 2 and Extended Data Fig. 3a-h.

Comment/Question 3

3) They permuted 100 times for most tests, but what would happen if they did this 500 or 1,000 times? How did they pick this number? Usually there is a parameter sweep that shows how a classifier threshold was chosen, and the authors may be able to improve their methods with a feature selection threshold sweep.

We thank the Reviewer for the comment and we completely agree that a feature selection threshold sweep would be an interesting possibility to optimize the model further. However, we would like to state that the focus on our paper was not so much the model itself, but rather the Swarm Learning framework, which by design is open for any kind of model implementation. We also exemplify this in the new Extended Data Fig. 6, where we compare the current neural network with a LASSO algorithm. Optimizing the current model further for a possible clinical application would indeed be very valuable, but we feel that this is beyond the scope of this paper. Regarding the number of permutations, we agree that this is an important aspect as well. In general, our reasoning was that we permuted the dataset where we had most cases and thus suspected higher variability between permutations more often. Following the suggestion by the Reviewer we checked whether there was a difference calculating the scenario depicted in Fig. 2 100 vs. 500 times. We found no difference in any of

the reported performance measures (see chart below).

Comment/Question 4

4) The COVID-19 data seems out of place here. With so few samples, the algorithms are almost certainly overfitting, but I don't see anything in the methods that shows how this is corrected? Can the authors clarify?

We recognize the comment of this Reviewer but would like to mention that Reviewer 2 was very much in favor of the COVID-19 work (comment 7: "Inclusion of COVID-19 data into this study is a plus"). We also understand the concern about overfitting which is an issue that has to be dealt with. We would like to emphasize that the signal for COVID-19 is extremely strong. In fact, when performing biological assessment of the data, the number of genes differentially expressed, and the magnitude of changes is tremendous. Considering this information, we are not so surprised that the model is able to learn with very few training samples. Nevertheless, we cross validated training and validation loss to ensure the models are not

overfitting. We now show training and validation loss (Extended Data Fig. 8I) for the scenario

described in Fig. 4d for training node 1 (10 cases: 90 controls) over 100 epoch and validated the loss function with the results obtained at the independent test dataset.

Comment/Question 5

5) Also for the COVID-19 data, they show strong signals in the blood, but co-infection can and does happen (7-20% depending on the study) and the authors should examine if there is any impact of this on their metrics (especially if they can get this from the clinical metadata, which would be cool). Co-infection summaries are here: <https://www.ncbi.nlm.nih.gov/pmc/articles/PMC7255350>.

We thank the Reviewer for this very insightful comment. With the help of our clinical colleagues, we re-assessed co-infections and generated a scenario, where we distributed cases unequally across nodes concerning co-infections (Extended Data Fig. 10c, lines 571ff, 1063ff). As can be seen, co-infections only have a slight impact on overall performance, however, most important SL outperforms also under these conditions.

Comment/Question 6

6) For the RNA-seq data, there are many features that can alter the accuracy and quantification dynamics, as well as dramatically different kinds of library preparations, but this does not seem to be addressed. The authors themselves noted that they were surprised that the RNA-seq and array data could be better merged here, but what about other features like read length or platform? Could they make a compelling case that it also does not matter for those features as well (for example, shown <https://genomebiology.biomedcentral.com/articles/10.1186/s13059-015-0697-y>)? This is one of the most compelling cases for these methods, is that even with the noise, you can make it through and find significant clinical features.

We thank the reviewer for this insightful suggestion and agree that the specifications of the RNA-seq methods used by the different studies are an important piece of information to consider. We followed the suggestion of the reviewer and set up a scenario for the prediction of AML where we distributed samples to nodes according to their sequencing platform. Samples sequenced with Illumina HiSeq 2500 and reads of 50 bp length (protocol 2) are distributed to node 1, Illumina HiSeq 2000 and reads of 100 bp length (protocol 2) to node 2, and a mixed dataset to node 3. The global test dataset to assess classifier performance was a mixed dataset with 50% AML and control cases from protocol 1 and 50% AML and control

cases sequenced with protocol 2 (**Extended Data Fig. 4i-j**).

Swarm Learning outperformed the nodes which were trained on only one sequencing strategy indicating that there are differences introduced by different protocols. We appreciate that additional confounding factors could contribute even further noise to the setting we have

chosen. To isolate the role of library production, sequencing conditions etc. future work will need to design studies around these questions separately. Nevertheless, important is the fact that SL can cope with possible technical batch effects successfully. We provide all available information on sequencing protocol, read length, platform for those datasets for which we assess the influence on sequencing technology used in Supplementary Table 6.

Comment/Question 7

7) For the access to computational resources (e.g. XSEDE) and code base, can the authors describe how they deploy these methods and help others to use? This is a critical feature of data access, and their GitHub page (https://github.com/schultzelab/swarm_learning) only hosts a Jupyter notebook and a few scripts, but very limited annotation or descriptions of their methods. This can and should be expanded and made more explicit.

We agree with the reviewer, access to computational resources for other scientists described and explored here is an important goal. Since the Swarm Learning technology will be very widely usable for many fields including but not limited to research data in the life and medical sciences, HPE has undertaken the following steps: For interest from the private sector, a licensing model will be established, since the Swarm Learning Library cannot be made open source at this time. For interested researchers within the academic field an evaluation license for research purposes will be available. Such license will also come with the respective instructions for the use of the Swarm Learning Library so that interested researcher will be able to create the Swarm network environment, including setup of a Swarm node, the Swarm node network, onboarding of Swarm nodes and training rounds within the Swarm network.

For the Reviewers we provide a sandbox environment which hosts a 3 node Swarm network with the AML dataset distributed among the nodes and a test node. Shell scripts are provided to run the experiments using this Swarm network and the AML dataset. Reviewers can request access to the system via the Editors, who will be provided with respective access codes, if requested. Additionally, within the methods section (lines 1512-1569), we now provide further information concerning the code used, the description about the Swarm licenses and we provide parameter settings used in the cases described in the manuscript (see Supplementary Table 8). Since we mainly used standard Keras models, we provide information about the steps how we used the existing ML software.

Comment/Question 8

8) For the discussion, the authors should examine a bit more of the details of what cannot be seen with these methods. Related to #1, the authors claim to detect AML and ALL with high accuracy (96-96%), but what about the sub-classes of AML? There are some types of AML that

show hypermethylator phenotypes or much more aggressive disease based on genetics/epigenetic differences, and even entirely novel classifiers based on genetic and epigenetic heterogeneity (e.g. <https://europepmc.org/article/pmc/pmc4938719>). The do approach this with acute vs. latent Tb, but it would be good to think about other parameters in other diseases, and if/how this SL approached could apply.

We thank the Reviewer for these valuable points. We interpret the questions that the reviewer agrees with us on the fact that Swarm Learning opens many new avenues, which is exactly what the paper is intended to do. In a recent manuscript, we have partially addressed the more specialized question concerning subclassification of AML (Warnat-Herresthal et al., iScience, 2020). While we also think that this is an extremely important question, the data collected retrospectively do not allow to answer this question for all subtypes since the classification of the AMLs has changed since the time when most of the samples were collected and a re-naming according to the new classification is not possible, hence, the existing dataset cannot be used to answer this question. We know from some of our collaborators that currently generated blood transcriptome data will become available in the future to address this question also in a Swarm Learning setting. At the moment, this would require the initiation of a new trial which we think goes beyond the scope of the current manuscript. As correctly pointed out by the Reviewer, we addressed this for tuberculosis to illustrate that this is - in principle - possible. We now also added a multi-classification approach for identifying all major leukemias (**Extended Data Fig. 5f-g**) and we provide a multi-classification approach for chest X-rays (**Figure 3m-n**). Both cases are different to what the Reviewer suggested, however, we propose that these cases also represent examples of additional models as they were proposed by the reviewer. Moreover, the fact that samples from the large majority of convalescent COVID-19 cases are not recognized when only acute cases are used for training further supports that Swarm Learning will perform under more complex clinical scenarios. We strongly believe that our manuscript will exactly trigger these additional very interesting and important questions as this was already the case for our Reviewer.

Reviewer Reports on the First Revision:

Referees' comments:

Referee #1 provided comments to the editors, which are paraphrased below:

The referee states that "at present the claims in the paper remain overstated. The authors should remove any claims regarding privacy. They can expand on the discussion regarding why they are not actually providing privacy in the paper in the appropriate section. Once this is fixed, we will be happy to see the paper again, but I find it unacceptable that the authors are openly stating that their method enables privacy preserving learning when this is not the case. This is a very important point in the healthcare domain and it is vital to be precise with respect to what the method actually provides."

The reviewer thinks that the explanations provided evade the fact that they are not providing privacy in the approach by saying in the letter that it is orthogonal, which is not currently reflected on the title. This may be better reflected by saying, the referee thinks, that "they provide "confidentiality" guarantees". The reviewer thinks that the approach does not implement or evaluate any mechanism to obtain privacy as stated. The reviewer thinks that some statements are inaccurate, and there is confusion between obfuscation and anonymization as providing privacy, when it is well known for more than a decade that neither of these techniques provide any meaningful privacy guarantees.

The reviewer states that the main difference provided compared to FL is that the approach doesn't assume a central party will hold the model updates and aggregate them. Their assessment is that this is a slight improvement in the underlying trust model but not a significant gain in terms of privacy.

Overall the reviewer feels that the revision did not make substantial changes on the aspect they previously evaluated in the previous round, so their recommendation is unchanged.

Referee #2 (Remarks to the Author):

I am glad to see the authors made efforts to address all the questions. My comments are as follow:

1. The author further test the SL method in non-IID settings. Though the F1 score and accuracy appear to be better than individual controls, it is hard to tell if sensitivity and specificity are improved. However, knowing federated training on non-IID data is hard and this is applied research rather than a pure algorithm research, this kinds of results are acceptable.
2. It is good to know that data used in this study were really siloed in real-world use case, though the number of silos is small, which make SL/federated learning not absolutely necessary in this setting (eg. a continual learning/repeating training scheme can be used), I think the method proposed still holds values when larger data networks become available and results from this study is inspiring in the medical informatics community.
3. Though I agree that aggregation all data for training is very difficult in the medical community, another "control experiment" could be ensemble model. The simplest form can be just averaging the predictions from models trained in each silo
4. I am glad to see that even if the authors increased the number of nodes, SL still work well. However, I still not convinced that blockchain is necessary in this use case as the primary assumption in blockchain is that no one is trustable and there are huge number of users(like thousands if not millions)
5. The new COVID-19 data does add more strength to the conclusion.

Referee #4 (Remarks to the Author):

In this revised paper, "Swarm Learning enables privacy-preserving machine learning for disease classification," Warnat-Herresthal et al have extensively updated their paper and added a range of new authors, analyses, and data sets. This includes several key experiments to validate their original findings (across 28 pages of response), most notably:

- 1) A new multi-class predictor for the different leukemias (Extended Data Fig. 5f-g), a new data set

addressing a multi-class problem in chest X-rays (Fig. 3m-n), and a larger case set for COVID-19, with additional centers with different COVID-19 patient cohorts.

2) They assessed the influence of factors such as age or sex on performance (Extended Data Fig. 10), which show limited impact.

3) The authors have made a new Extended Data Fig. 6, comparing the current neural network with a LASSO algorithm, and show rough equivalence.

4) They tested 500 vs. 1000 permutations and showed the results were very robust to this iteration.

5) They have updated their GitHub page, which is appreciated.

Given all these updates and responses, I am satisfied with the updated manuscript.

Author Rebuttals to First Revision (author comments are in blue):

Referee #1 provided comments to the editors, which are paraphrased below:

The referee states that “at present the claims in the paper remain overstated. The authors should remove any claims regarding privacy. They can expand on the discussion regarding why they are not actually providing privacy in the paper in the appropriate section. Once this is fixed, we will be happy to see the paper again, but I find it unacceptable that the authors are openly stating that their method enables privacy preserving learning when this is not the case. This is a very important point in the healthcare domain and it is vital to be precise with respect to what the method actually provides.”

We thank the Editors for sharing with us this very important information. We recognize the trust the Editors have given us with making the decision to share this information. Consequently, we have discussed this important point in detail again among the experts within our large team.

As a result, we provide the following solution to the Editors: With introducing rather minor changes in the text, we think that we can positively follow the suggestion of the reviewer without losing the overall message of the manuscript. We provide the changes marked in yellow in a separate version of the manuscript as a supplementary for completeness, but already provide a clean version of the document with all the changes as the revised manuscript. In addition, we added a new panel to Extended Data Figure 1 (1a) further illustrating the uniqueness of the technical setup of Swarm Learning concerning consorial data governance, trust, confidentiality, and security.

Furthermore, we share with you our thought process for the suggested solution to answer the Reviewer’s concern:

Reviewer #1 raises an important point, and we agree with her/him that we need to be careful with the term privacy-preserving. Indeed, this term is sometimes - even in the recent literature

- not used precise enough. During the last weeks, we continued to study this issue within our team, because we also felt that it will stay of high importance.

Privacy-preserving is being used for many different dimensions that are necessary to be taken care of, when considering machine learning. We worked on a model that brings these dimensions into the context of Swarm Learning (which is now added as Extended Data Figure 1a). These dimensions include consortial data governance, trust, confidentiality, security and data privacy that all must be dealt with to reach complete privacy. The Reviewer is correct, the dimension data privacy (protecting the data with i.e. Differential Privacy) was not addressed experimentally by us, because we felt it to be orthogonal to Swarm Learning, since data privacy is a general dimension, which would have to be similarly addressed for any machine learning, be it central learning or federated learning with a central party/custodian. On the other hand, the combination of all the other dimensions within Swarm Learning enables a privacy-preserving environment. Strictly speaking, we are both correct, the Reviewer, because he argues that data privacy is part of the complete protection system and we when we were speaking about the Swarm Learning-specific infrastructure aspects.

To accommodate the Reviewers criticism, we have made respective changes in the document (lines 1ff, 93f, 104,111, 157, 160, 168, 215f, 226, 261, 596, 599, 674, 812, 916, 1389ff, 1476ff).

The reviewer thinks that the explanations provided evade the fact that they are not providing privacy in the approach by saying in the letter that it is orthogonal, which is not currently reflected on the title This may be better reflected by saying, the referee thinks, that “they provide “confidentiality” guarantees”. The reviewer thinks that the approach does not implement or evaluate any mechanism to obtain privacy as stated. The reviewer thinks that some statements are inaccurate, and there is confusion between obfuscation and anonymization as providing privacy, when it is well known for more than a decade that neither of these techniques provide any meaningful privacy guarantees.

As outlined above, we follow the Reviewer by making the respective changes throughout the document. Once we follow this route, the further arguments (the confusion about obfuscation and anonymization as providing privacy) of the Reviewer in this statement do not apply anymore and we therefore would not discuss or address them in the paper to avoid any confusion for the reader.

The reviewer states that the main difference provided compared to FL is that the approach doesn't assume a central party will hold the model updates and aggregate them. Their assessment is that this is a slight improvement in the underlying trust model but not a significant gain in terms of privacy.

We respectfully disagree with the Reviewer. Making a central party obsolete is a conceptual change for the trust model equally sharing results and this is certainly not a slight improvement, but a game changer. This has been stated in the manuscript for the reader to easily grasp the difference. In addition, the three other reviewers also follow our chain of argument. We have no right to question the motivations for downplaying this fact by the Reviewer, but we

respectfully ask the Editors to appreciate this critical change introduced by SL. As long as a central party is allowed, such as in current federal learning approaches, data monopolists will remain in control of the learning process and the use thereof. Swarm Learning is critically different recognizing equal rights for all members of the swarm. For medicine, but also for other sectors with private data, this is not only an important difference to FL, it is the critical difference.

In other words, while some might see the technical differences to be incremental, the change of how this will be implemented has not only technical implication, it has a major implication for how ML and AI will be implemented in the medical sector and other major sectors with private data.

Making the central party required in FL obsolete is an important step towards more independence from data monopolists, which is an important step towards more secure systems for ML and AI as well as a prerequisite for implementing privacy (following the definition by the Reviewer). A precise definition of the term privacy is currently lacking for the different domains (here medicine and computation science) when looking solely from the angle of privacy-preserving artificial intelligence. Privacy-preserving AI has the intention to keep data protected while confidentiality cares for unintended leakage and from deliberate disclosure attempts. Only the combination of both of these aforementioned dimensions in addition to the enabling trust elements, appropriate security measures and governance structures build out a comprehensive picture in terms of privacy needed. While of highest interest, we feel that it is beyond the scope of the first manuscript on SL to address the weaknesses of FL at the same time. Rather we respectfully ask the Editor to allow us to focus the manuscript on describing the characteristics and on the performance of SL. Furthermore, our findings for COVID-19, tuberculosis and leukemias are also medically of highest interest, which goes far beyond the privacy question.

Overall the reviewer feels that the revision did not make substantial changes on the aspect they previously evaluated in the previous round, so their recommendation is unchanged.

As outlined above, we hope, we now have clarified both points sufficiently: The argument that we overstate privacy-preserving is addressed as well as the argument about the conceptual difference between FL and Swarm Learning.

Referee #2 (Remarks to the Author):

I am glad to see the authors made efforts to address all the questions. My comments are as follow:

1. The author further test the SL method in non-IID settings. Though the F1 score and accuracy appear to be better than individual controls, it is hard to tell if sensitivity and specificity are improved. However, knowing federated training on non-IID data is hard and this is applied research rather than a pure algorithm research, this kinds of results are acceptable.

We thank the Reviewer for this comment. Sensitivity and specificity for all scenarios are provided within the supplements. We apologize, if this was not clear for the Reviewer. Overall, we interpret the Reviewer's comment ("results are acceptable") that the Reviewer has no further requests for this point.

2. It is good to know that data used in this study were really siloed in real-world use case, though the number of silos is small, which make SL/federated learning not absolutely necessary in this setting (eg. a continual learning/repeating training scheme can be used), I think the method proposed still holds values when larger data networks become available and results from this study is inspiring in the medical informatics community.

We appreciate the comment by the Reviewer and also think that our work is inspiring the medical informatics community. We further agree that these approaches will become even more important if the number of swarm members further increases. If we interpret the comment correctly, the Reviewer does not make any additional requests.

3. Though I agree that aggregation all data for training is very difficult in the medical community, another "control experiment" could be ensemble model. The simplest form can be just averaging the predictions from models trained in each silo

This is an interesting point raised by the Reviewer and we have now added this to the manuscript.

Ensemble learning works based on the concept of training multiple models and then combining their predictions. Averaging the predictions is the most commonly used technique of combining predictions. This makes the model generic compared to any individual model used in ensemble. However, the swarm learning and federated learning techniques combine the model parameters, i.e. weights of neural networks, instead of final prediction. Also the combining process is more granular. This helps in achieving better results.

As highlighted in the paper "Federated learning in medicine: facilitating multi-institutional collaborations without sharing patient data" (<https://www.nature.com/articles/s41598-020-69250-1>), increasing the frequency of averaging (synchronization) will improve the model accuracy. Swarm Learning provides a parameter to tune this frequency as low as a batch level. So, Swarm Learning will be able to achieve higher accuracy compared to ensemble models. To illustrate this for the AML data in our manuscript (Fig. 2d), we have averaged the prediction results as it would be done by an ensemble learning approach as suggested by the reviewer by averaging predictions. Swarm outperforms the ensemble model in accuracy, sensitivity, specificity and AUC, as shown in the plots below.

4. I am glad to see that even if the authors increased the number of nodes, SL still work well. However, I still not convinced that blockchain is necessary in this use case as the primary assumption in blockchain is that no one is trustable and there are huge number of users(like thousands if not millions)

We thank the Reviewer for this comment. The idea behind using private permissioned blockchain, even if the number of members in a swarm is low, is to provide a standardized framework of joint learning without making any exceptions at the lower boundary. Since the blockchain comes with a smart contract, we would argue that even if two independent entities learn together, the legal regulations for rights and responsibilities of each partner need to be clear and smart contracts inherited within the blockchain will guarantee this. This is certainly true in a commercial setting, but similarly true for research consortia, which also need the respective contracts between the contributing institutions prior such consortia projects are performed. We have now added this additional aspect to the Supplementary Information to make the reader aware that the blockchain contains such a smart contract and that we favor a unified model independent of swarm size (see lines 1395ff.).

5. The new COVID-19 data does add more strength to the conclusion.

We thank the Reviewer for this very positive judgement.

Referee #4 (Remarks to the Author):

In this revised paper, "Swarm Learning enables privacy-preserving machine learning for

disease classification,” Warnat-Herresthal et al have extensively updated their paper and added a range of new authors, analyses, and data sets. This includes several key experiments to validate their original findings (across 28 pages of response),

We thank the Reviewer for this very positive response

most notably:

1) A new multi-class predictor for the different leukemias (Extended Data Fig. 5f-g), a new data set addressing a multi-class problem in chest X-rays (Fig. 3m-n), and a larger case set for COVID-19, with additional centers with different COVID-19 patient cohorts.

We thank the Reviewer for recognizing our efforts to answer the raised questions.

2) They assessed the influence of factors such as age or sex on performance (Extended Data Fig. 10), which show limited impact.

We thank the Reviewer for recognizing our efforts to answer the raised questions.

3) The authors have made a new Extended Data Fig. 6, comparing the current neural network with a LASSO algorithm, and show rough equivalence.

We thank the Reviewer for recognizing our efforts to answer the raised questions.

4) They tested 500 vs. 1000 permutations and showed the results were very robust to this iteration.

We thank the Reviewer for recognizing our efforts to answer the raised questions.

5) They have updated their GitHub page, which is appreciated.

We thank the Reviewer for recognizing our efforts to answer the raised questions.

Given all these updates and responses, I am satisfied with the updated manuscript. We

thank the Reviewer for recognizing our efforts to answer the raised questions.

Reviewer Reports on the Second Revision:

Referee #1 expressed to the editors that they are satisfied with the revision.

Referee #2 (Remarks to the Author):

I appreciate the authors' efforts in answering all the questions and addressing all the concerns. Though the only thing I am still not convinced is the value of using blocking chain when there are not many nodes in the system. When there are only very few nodes, the data sharing can simply be series of paperworks, which needs to be done anyway even with the blockchain. Having said this, this concern does not question the novelty of this manuscript.

Author Rebuttals to Second Revision (author comments are in blue):

Referees' comments:

Referee #1 expressed to the editors that they are satisfied with the revision.

Answer:

We thank the reviewer for the final answer.

Referee #2 (Remarks to the Author):

I appreciate the authors' efforts in answering all the questions and addressing all the concerns. Though the only thing I am still not convinced is the value of using blocking chain when there are not many nodes in the system. When there are only very few nodes, the data sharing can simply be series of paperworks, which needs to be done anyway even with the blockchain. Having said this, this concern does not question the novelty of this manuscript.

Answer:

We thank the reviewer for his/her appreciation of our work. Considering a continuous digitalization of our world we prefer blockchain technology as part of electronic smart contracts within Swarm Learning solutions as being the preferred solution over paperwork based contracts, even if the collaborating parties are only few. Since it can be foreseen that smart contracts will replace paper-based solutions, Swarm Learning is fully prepared for these future developments.